# Vanishing Feature: Diagnosing Model Merging and Beyond

Xingyu Qu[1], Samuel Horvath[1]

[1]Mohamed bin Zayed University of Artificial Intelligence (MBZUAI),

Xingyu.Qu@mbzuai.ac.ae, Samuel.Horvath@mbzuai.ac.ae

Model merging offers an efficient way to combine pre-trained neural networks but often suffers from inconsistent performance, especially when merging models with different initializations. We identify the "vanishing feature" phenomenon, where input-induced features diminish during propagation through the merged model, degrading performance. Through theoretical and empirical analysis, we reveal that this phenomenon underpins challenges like variance collapse and explains techniques like permutation-based merging, post-merging normalization, etc. We show that existing normalization strategies can be enhanced by precisely targeting the vanishing feature issue. Leveraging these insights, we propose the "Preserve-First Merging" (PFM) strategy, which preserves early-layer features, enabling merged VGG16 models on CIFAR-10 to surpass the original models without post-training for the first time. Furthermore, we demonstrate that the vanishing feature phenomenon extends to other contexts, such as model pruning. Applying post-pruning normalization to mitigate the issue significantly improves one-shot pruning performance at high sparsity, offering a simple and effective post-pruning solution. The code is available at https://github.com/XingyuQu/VF.

## 1. Introduction and Related Work

Model merging [1–6] combines pre-trained neural networks, enabling reuse without costly retraining. By merging the parameters of multiple *edge models* into a single unified model, this technique aims to aggregate knowledge to approximate the performance of model ensembles [7] while reducing storage and computation costs. However, despite its promise, the performance of merged models is often inconsistent, particularly when merging models trained with *different* initializations [8–10]. In such cases, the merged model's performance frequently degrades significantly to a random guess. Recent research attributes this poor performance to the permutation invariance of neural networks and the variance collapse phenomenon, proposing techniques like permutation-based merging, model fusion, and post-merging normalization for partial recovery [10–17]. However, these methods remain limited in many aspects, often requiring significantly increased model width or post-training to achieve optimal results. This unsatisfactory progress stems from the field's limited understanding, as existing analyses primarily focus on idealized scenarios and lack practical guidance [10, 18].

**Contribution.** This paper revisits the standard scenario of merging models trained on the same dataset but with different initializations, where existing methods often fail to match the performance of the edge models. Our key contributions are summarized as follows:

- We identify the *"vanishing feature"* phenomenon, where input-induced features progressively diminish during propagation through the model, rendering its performance highly fragile. This phenomenon can lead to the dominance of the input-independent bias terms in the model's output, causing the variance collapse observed in prior works. Furthermore, we show that several challenges in merging deep models and techniques like permutation-based merging and post-merging normalization can be understood through the lens of the vanishing feature. It also reveals a novel finding that residual connections can facilitate merging.
- Building on these insights, we demonstrate that current post-merging normalization strategies can be refined to more effectively address the vanishing feature phenomenon, yielding significant

Second Conference on Parsimony and Learning (CPAL 2025).

improvements in specific scenarios. Additionally, we emphasize the importance of preserving features in early layers in achieving optimal merging performance and introduce the *"Preserve-First Merging"* (PFM) strategy. PFM improves current merging performance substantially while maintaining an efficiency-accuracy trade-off. Notably, it enables merged VGG16 models on CIFAR-10 to outperform the edge models without post-training for the first time.

- We show that the vanishing feature phenomenon extends beyond model merging to other scenarios, such as model pruning [19]. Moreover, applying a modified normalization technique effectively mitigates this issue, significantly enhancing the performance of one-shot pruning at high sparsity without the need for additional parameters or post-training.

**Notation.** We use boldface symbols to denote vectors ($\mathbf{x}$), matrices ($\mathbf{W}$), and vector-valued functions ($\mathbf{f}(\mathbf{x})$), while scalar variables ($x$) and scalar-valued functions ($f(x)$) are represented in plain font. Subscripts like $\alpha$ in $\mathbf{f}_\alpha$ denote the *merged model* obtained by interpolating the parameters of the *edge models* $\mathbf{f}_0$ and $\mathbf{f}_1$, i.e., $\boldsymbol{\Theta}_\alpha = (1 - \alpha) \cdot \boldsymbol{\Theta}_0 + \alpha \cdot \boldsymbol{\Theta}_1$, where $\boldsymbol{\Theta}$ represents the model's parameters. In the context of $f_\alpha$, we often use statistics weighted-averaged over edge models, denoted by an overline in the paper, e.g., $\overline{f(x)} := (1 - \alpha) \cdot f_0(x) + \alpha \cdot f_1(x)$. For an integer $N$, denotes $[N] = \{1, \ldots, N\}$.

## 2. Preliminaries

Given two pre-trained neural networks $\mathbf{f}_0$ and $\mathbf{f}_1$ with the same architecture, the goal of model merging is to construct a merged model $\mathbf{f}_\alpha$ ($0 \leq \alpha \leq 1$) that ideally outperforms both edge models.

**Permutation Invariance in Neural Networks.** Merging models trained with different initializations is challenging due to the *permutation invariance* property of neural networks [10, 11]. For an $L$-layer fully connected feed-forward network defined as $\mathbf{f}(\mathbf{x}) = \mathbf{f}^{(L)}(\mathbf{x})$, $\mathbf{f}^{(l)}(\mathbf{x}) = \sigma(\mathbf{W}^{(l)}\mathbf{f}^{(l-1)}(\mathbf{x}) + \mathbf{b}^{(l)})$, where $\mathbf{f}^{(l)}(\mathbf{x})$ is the output of the $l$-th layer, $\sigma$ is the activation function, and $\mathbf{W}^{(l)}, \mathbf{b}^{(l)}$ are the weights and biases, permutation invariance implies that permuting a layer's output, $\mathbf{f}^{(l)}(\mathbf{x}) \leftarrow \mathbf{P}\mathbf{f}^{(l)}(\mathbf{x})$, leaves the network output unchanged if the next layer's parameters are permuted correspondingly, $\mathbf{W}^{(l+1)} \leftarrow \mathbf{W}^{(l+1)}\mathbf{P}^T$, as $\mathbf{P}^T\mathbf{P} = \mathbf{I}$ for any permutation matrix $\mathbf{P}$. Note that if a layer's parameters are permuted, $\mathbf{W}^{(l)} \leftarrow \mathbf{P}\mathbf{W}^{(l)}$, $\mathbf{b}^{(l)} \leftarrow \mathbf{P}\mathbf{b}^{(l)}$, its output permute accordingly. Thus, this property can lead to significantly divergent training trajectories for models initialized differently, making the merging process more challenging.

**Permutation-Based Model Merging.** Previous works empirically show considerable improvement by eliminating this invariance through the *permutation-based model merging* [10–13, 17, 20]. Before merging, these methods apply a permutation to align the parameters between edge models. This permutation is often determined by minimizing the distance between networks in either the weight space or the activation space. Specifically, this involves solving an optimization problem $\min_\pi \|\boldsymbol{\Theta}_0 - \pi(\boldsymbol{\Theta}_1)\|$ or $\min_\pi \sum_l \|\mathbf{f}_0^{(l)}(\mathbf{x}) - \pi(\mathbf{f}_1^{(l)}(\mathbf{x}))\|$, where $\boldsymbol{\Theta}_0$ and $\boldsymbol{\Theta}_1$ represent the parameters and $\pi$ is the permutation function. Following Ainsworth et al. [11], we refer to these approaches as *weight-matching (WM)-based merging* and *activation-matching (AM)-based merging*. For merging across different datasets, Stoica et al. [13] proposed the *ZipIt!* method, introducing the insight that allowing parameter merging within a single model, rather than limiting it to inter-model merging, can be advantageous. This intra-merging can avoid the forced merging of dissimilar features across different datasets. Details are provided in Appendix A.8.2. While there exist advanced fusion-based merging strategies [14, 15, 21–23], this paper primarily focuses on permutation-based methods, as they are more computationally tractable and have demonstrated superior performance in our setting [11].

**Post-Merging Normalization.** The performance of permutation-based merging degrades on more advanced model architectures and datasets. Singh and Jaggi [14] showed that re-training the merged model can recover the performance, albeit with a computational cost. Jordan et al. [12] revealed that one critical issue behind the performance degradation is the existence of *"variance collapse"* phenomenon in the merged model, which can reduce the network to a random guess by having almost zero variance in the output layer. The authors then proposed *REPAIR* to address this issue by normalizing the merged model based on activation statistics from edge models. Formally, for edge models $\mathbf{f}_0$, $\mathbf{f}_1$, and the merged model $\mathbf{f}_\alpha$, REPAIR correct the pre-activation output of selected layers

by first normalizing it and then applying an affine transformation:

$$\textbf{REPAIR}: \; f_\alpha^{(l,i)}(\mathbf{x}) \leftarrow \frac{f_\alpha^{(l,i)}(\mathbf{x}) - \mathbb{E}_\mathbf{x}[f_\alpha^{(l,i)}(\mathbf{x})]}{\mathrm{Std}_\mathbf{x}[f_\alpha^{(l,i)}(\mathbf{x})]} \cdot \overline{\mathrm{Std}_\mathbf{x}[f^{(l,i)}(\mathbf{x})]} + \overline{\mathbb{E}_\mathbf{x}[f^{(l,i)}(\mathbf{x})]}, \tag{1}$$

where $\overline{\mathrm{Std}_\mathbf{x}[f^{(l,i)}(\mathbf{x})]} = ((1-\alpha) \cdot \mathrm{Std}_\mathbf{x}[f_0^{(l,i)}(\mathbf{x})] + \alpha \cdot \mathrm{Std}_\mathbf{x}[f_1^{(l,i)}(\mathbf{x})])$, $\overline{\mathbb{E}_\mathbf{x}[f^{(l,i)}(\mathbf{x})]} = (1-\alpha) \cdot \mathbb{E}_\mathbf{x}[f_0^{(l,i)}(\mathbf{x})] + \alpha \cdot \mathbb{E}_\mathbf{x}[f_1^{(l,i)}(\mathbf{x})])$, $f_*^{(l,i)}$ is the $i_{\mathrm{th}}$ element of the pre-activation at layer $l$ of model $\mathbf{f}_*$, and expectation is taken over the training distribution. Empirically, REPAIR mitigates variance collapse and significantly enhances model performance without requiring post-training. Technically, REPAIR involves inserting batch normalization [24] layers after selected layers. For models already incorporating batch normalization, REPAIR leverages the existing BatchNorm layers, a procedure referred to as *RESET* in this paper. Further details are provided in Appendix A.3.

## 3. Experimental Setup

We outline the experimental setup for this paper, with details provided in Appendix A.3. For merging experiments, we independently train two models with the same architecture on the same dataset but initialize them differently. Section 4 focuses on linear networks trained on MNIST [25], while later sections investigate multiple VGG16 [26] and ResNet20 [27] variants (e.g., with batch normalization or layer normalization [28]) on CIFAR-10, CIFAR-100 [29], and ImageNet [30]. The baseline merging methods include direct merging, weight/activation-matching-based merging [10, 11], ZipIt! [13], and CCA merging [23]. Appendix A.8.4 reports results on merging vision transformers [31] based on the OT-acts fusion [15]. The baseline normalization methods are REPAIR and RESET [12]. One-shot pruning experiments, including the ConvNeXt [32] model, are detailed in Section 5.5. For pruning, weights in all convolutional and linear layers are considered, except for the first convolutional layer in ResNet50 on ImageNet. All linear layers are pruned on ConvNeXt following Sun et al. [33].

## 4. Analyzing Merging in Linear Networks

This section analyzes the merging of two linear networks with different initializations. Section 4.1 establishes the theoretical setup and analyzes the variance collapse phenomenon. Section 4.2 introduces the *vanishing feature* phenomenon, identifying it as the root cause of variance collapse and degraded model performance.

### 4.1. Variance Collapse in Linear Networks

While REPAIR enhances merging for models with different initializations by addressing the variance collapse issue, a performance gap often persists. To address this, we start by revisiting variance collapse on linear networks to uncover deeper insights. Consider merging two independently trained linear networks of depth $L$: $\mathbf{f}_i(\mathbf{x}) = \mathbf{f}_i^{(L)}(\mathbf{x})$, $\mathbf{f}_i^{(l)}(\mathbf{x}) = \mathbf{W}_i^{(l)}\mathbf{f}_i^{(l-1)}(\mathbf{x}) + \mathbf{b}_i^{(l)}$, $\mathbf{f}_i^{(0)}(\mathbf{x}) = \mathbf{x}$, $i \in \{0, 1\}$, $l \in [L]$. Here, $\mathbf{W}_i^{(l)}$ and $\mathbf{b}_i^{(l)}$ represents the weight matrix and bias vector at $l_{\mathrm{th}}$ layer of $\mathbf{f}_i$. We formalize the definition of variance collapse in the following way:

**Definition 4.1** ($\varepsilon_{\mathcal{D},\mathbf{f}}$-**Variance Collapse**). For a neural network $\mathbf{f}$ of depth $L$, there is an $\varepsilon_{\mathcal{D},\mathbf{f}}$-*variance collapse* over data distribution $\mathcal{D}$ if there exists a non-increasing upper bound sequence $(\varepsilon_{\mathcal{D},\mathbf{f}}^{(l)})_{l=1}^L$ and a small constant $\varepsilon_{\mathcal{D},\mathbf{f}} > 0$ such that $\mathrm{Var}_{\mathbf{x} \sim \mathcal{D}}[\mathbf{f}^{(l)}(\mathbf{x})] \leq \varepsilon_{\mathcal{D},\mathbf{f}}^{(l)}$ for $l \in [L]$, and $\varepsilon_{\mathcal{D},\mathbf{f}}^{(L)} \leq \varepsilon_{\mathcal{D},\mathbf{f}}$.

For linear networks, the intermediate activations can indeed be written explicitly with a feature-bias decomposition (derivations in Appendix A.1):

$$\mathbf{f}_\alpha^{(l)}(\mathbf{x}) = \underbrace{\prod_{j=1}^l ((1-\alpha)\mathbf{W}_0^{(j)} + \alpha\mathbf{W}_1^{(j)})\mathbf{x}}_{\mathbf{g}_\alpha^{(l)}(\mathbf{x})} + \underbrace{\sum_{j=1}^l \prod_{k=j+1}^l ((1-\alpha)\mathbf{W}_0^{(k)} + \alpha\mathbf{W}_1^{(k)})((1-\alpha)\mathbf{b}_0^{(j)} + \alpha\mathbf{b}_1^{(j)})}_{\mathbf{h}_\alpha^{(l)} = \sum_{j=1}^l \tilde{\mathbf{h}}_\alpha^{(l)}(\mathbf{b}^{(j)})},$$

$$\tag{2}$$

where $\mathbf{g}_\alpha^{(l)}(\mathbf{x})$ represents the *input-induced latent feature* extracted by the first $l$ layers, $\mathbf{h}_\alpha^{(l)}$ is an *input-independent activation offset* determined solely by model's parameters, and $\tilde{\mathbf{h}}_\alpha^{(l)}(\mathbf{b}^{(j)})$ quantifies the contribution of biases at the $j_{\text{th}}$ layer in $\mathbf{h}_\alpha^{(l)}$. From Equation 2, it's clear that $\text{Var}[\mathbf{f}_\alpha^{(l)}(\mathbf{x})] \equiv \text{Var}[\mathbf{g}_\alpha^{(l)}(\mathbf{x})]$, which reveals that *the variance collapse of intermediate activations originates from the variance collapse of input-induced features.*

## 4.2. Vanishing Feature in Linear Networks

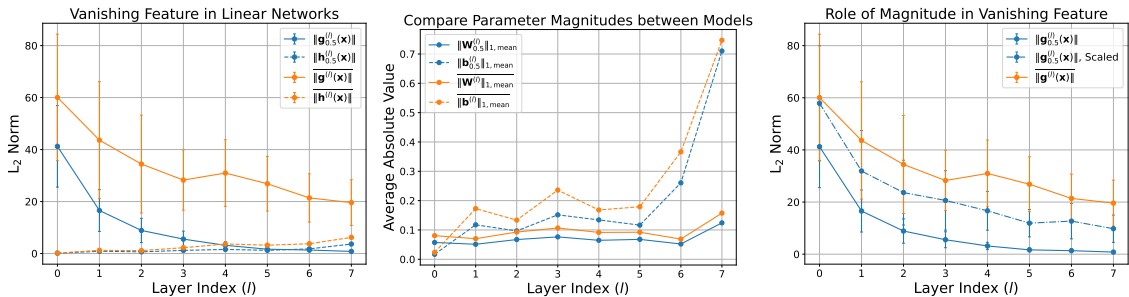

Figure 1: **Vanishing feature phenomenon in merged linear networks $\mathbf{f}_{\alpha=0.5}$.** Edge models $\mathbf{f}_0$, $\mathbf{f}_1$ are trained on MNIST with different initializations. The mean and standard deviation over the test set are reported in the left and right plots. Statistics denoted by an overline are averaged across edge models. The average absolute value of a matrix $(A_{i,j})_{i\in[m],j\in[n]}$ is given as $\|\mathbf{A}\|_{1,\text{mean}} = \sum_{i,j} |A_{i,j}|/(mn)$.

**Left:** The input-induced feature extracted by the merged model ($\mathbf{g}_{0.5}^{(l)}(\mathbf{x})$) progressively diminishes towards zero, causing the input-independent offset ($\mathbf{h}_{0.5}^{(l)}(\mathbf{x})$) dominates the activation in later layers, degrading the model performance. **Mid:** Parameter magnitudes at each layer decrease after merging, which contributes to the vanishing of features. On the other hand, bias magnitudes increase across layers, shaping the gap between $\|\mathbf{g}_{0.5}^{(l)}(\mathbf{x})\|$ and $\|\mathbf{h}_{0.5}^{(l)}\|$ as seen in the left plot. **Right:** Scaling up $\mathbf{g}_\alpha^{(l)}(\mathbf{x})$ by the ratio of its magnitude to that of the average magnitude of edge models alleviates the vanishing feature issue, suggesting that the reduced magnitude is one underlying factor.

From the last section, we know that a small $\varepsilon_{\mathcal{D},\mathbf{f}}$ in $\varepsilon_{\mathcal{D},\mathbf{f}}$-variance collapse implies that the model activation $\mathbf{f}^{(l)}(\mathbf{x})$ and input-induced feature $\mathbf{g}^{(l)}(\mathbf{x})$ converge to nearly constant vectors over the data distribution $\mathcal{D}$ in the deeper layers. Understanding these limit points is essential for diagnosing the performance degradation of merging. To investigate, we trained two 8-layer linear networks on MNIST and analyzed the merged midpoint model $\mathbf{f}_{\alpha=0.5}$. Interestingly, as shown in Figure 1 (**Left**), we observed that $\mathbf{g}_{0.5}^{(l)}(\mathbf{x})$ indeed converged to zero as $l$ increased. We term this phenomenon as *"vanishing feature"* (Definition 4.2). Intuitively, this reflects reduced model dependence on input, making the output more vulnerable to irrelevant noise.

**Definition 4.2** ($\varepsilon_{\mathcal{D},\mathbf{f}}$-**Vanishing Feature in Linear Networks**). For a linear network $\mathbf{f}$ of depth $L$, there is an $\varepsilon_{\mathcal{D},\mathbf{f}}$-*vanishing feature* over data distribution $\mathcal{D}$ if there exits a non-increasing upper bound sequence $(\varepsilon_{\mathcal{D},\mathbf{f}}^{(l)})_{l=1}^L$ and a small constant $\varepsilon_{\mathcal{D},\mathbf{f}} > 0$ such that $\mathbb{E}_{\mathbf{x}\sim\mathcal{D}}[\|\mathbf{g}^{(l)}(\mathbf{x})\|] \leq \varepsilon_{\mathcal{D},\mathbf{f}}^{(l)}$ for $l \in [L]$, and $\varepsilon_{\mathcal{D},\mathbf{f}}^{(L)} \leq \varepsilon_{\mathcal{D},\mathbf{f}}$.

**Effect of Biases.** Our experiments reveal that the performance degradation of $\mathbf{f}_{0.5}$ stems from noise introduced by the input-independent offset $\mathbf{h}_{0.5}^{(l)}$. While $\mathbf{g}_{0.5}^{(l)}(\mathbf{x})$ progressively diminishes, $\mathbf{h}_{0.5}^{(l)}$ increases in magnitude, eventually surpassing $\mathbf{g}_{0.5}^{(l)}(\mathbf{x})$ in scale within the final layers (Figure 1, **Left**). Consequently, the latent activation $\mathbf{f}_{0.5}^{(l)}(\mathbf{x})$ converges to $\mathbf{h}_{0.5}^{(l)}$, which depends on $\{\mathbf{W}_{0.5}^{(j)}, \mathbf{b}_{0.5}^{(j)}\}_{j\in[l]}$. This convergence causes variance collapse and contributes to the decline in model performance.

**Magnitude Perspective.** We further explored the contrasting behaviors between $\mathbf{g}_\alpha^{(l)}(\mathbf{x})$ (continuously shrinking) and $\mathbf{h}_\alpha^{(l)}$ (continuously growing). As shown in Figure 1 (**Mid**), the parameter

scale in the merged model decreased compared to the edge models. Generally, for a linear network $\mathbf{f}$, suppose all the parameters are scaled as $\mathbf{W}^{(l)}, \mathbf{b}^{(l)} \leftarrow \lambda\mathbf{W}^{(l)}, \mu\mathbf{b}^{(l)}$ with $\lambda, \mu < 1$[1]. This results in $\mathbf{g}^{(l)}(\mathbf{x}), \tilde{\mathbf{h}}^{(l)}(\mathbf{b}^{(j)}) \leftarrow \lambda^l \cdot \mathbf{g}^{(l)}(\mathbf{x}), \lambda^{l-j}\mu \cdot \tilde{\mathbf{h}}^{(l)}(\mathbf{b}^{(j)})$. Consider the following more fine-grained feature-bias decomposition (notation follows Equation 2):

$$\mathbf{f}^{(l)}(\mathbf{x}) = \mathbf{g}^{(l)}(\mathbf{x}) + \mathbf{h}^{(l)} = \mathbf{g}^{(l)}(\mathbf{x}) + \sum_{j=1}^{s-1} \tilde{\mathbf{h}}^{(l)}(\mathbf{b}^{(j)}) + \sum_{j=s}^{l} \tilde{\mathbf{h}}^{(l)}(\mathbf{b}^{(j)}),$$

where $s$ separates the contributions of small $j$ and large $j$. For instance, $s$ can be chosen as $l - s'$, where $s'$ is a small constant satisfying $0 \leq s' \leq l_0 - 1$ when varying $l$ from $l_0$ to $L$. As $l$ increases, the contributions from $\mathbf{g}^{(l)}(\mathbf{x})$ and $\sum_{j=1}^{s-1} \tilde{\mathbf{h}}^{(l)}(\mathbf{b}^{(j)})$ decay exponentially. Consequently, the term $\sum_{j=s}^{l} \tilde{\mathbf{h}}^{(l)}(\mathbf{b}^{(j)})$, which depends solely on the parameters in later layers, becomes increasingly dominant. This insight extends to the merged network $\mathbf{f}_\alpha$ by comparing its parameter scale to edge models, explaining the continual growth of $\mathbf{h}_\alpha^{(l)}$ by observing the increased parameter scales in the final layers of $\mathbf{f}_\alpha$ (Figure 1, **Mid**). Rescaling $\mathbf{g}_\alpha^{(l)}(\mathbf{x})$ at each layer by $\prod_{j=1}^{l} \overline{\|\mathbf{W}^{(j)}\|_{1,\text{mean}}}/\|\mathbf{W}_\alpha^{(j)}\|_{1,\text{mean}}$ alleviated the vanishing feature phenomenon (Figure 1, **Right**).

## 5. Vanishing Feature

This section builds on the analysis in the last section. Section 5.1 formalizes the generalized definition of the vanishing feature phenomenon and empirically validates its occurrence across diverse settings. Section 5.2 demonstrates how the vanishing feature phenomenon provides a unifying explanation for key strategies and challenges in model merging and uncovers the novel finding that residual connections can effectively enhance merging. Additionally, we discovered the impact of learning rate and weight decay in merging, reported in Appendix A.4. Section 5.3 empirically shows that addressing the vanishing feature can enhance post-merging strategies, leading to improved merging performance. Leveraging these insights, the *Preserve-First Merging* (PFM) strategy is proposed in Section 5.4, providing superior merging performances. Finally, Section 5.5 extends the scope by showing that the vanishing feature phenomenon also persists in one-shot pruning scenarios and that applying post-pruning normalization techniques improves performance, underscoring its broader relevance to neural network research. Some theoretical insights into the vanishing feature phenomenon are discussed in Appendix A.5.2.

### 5.1. Vanishing Feature in General Settings

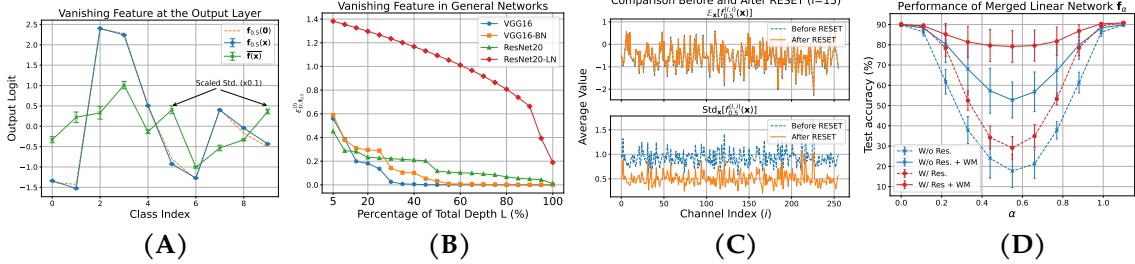

(A)  (B)  (C)  (D)

Figure 2: (**A & B**): Visualize the vanishing feature phenomenon in the merged midpoint model $\mathbf{f}_{0.5}$. Edge models are trained on CIFAR-10. (**A**) shows output logits of ResNet20s averaged over the test dataset, where the standard deviation is scaled by $0.1\times$ for better visualization. (**B**) presents the normalized upper bound sequence $(\varepsilon_{\mathcal{D},\mathbf{f}_{0.5}}^{(l)})_{l=1}^{L}$, where the normalization factor is $\overline{\mathbb{E}_\mathbf{x}[\mathbf{g}^{(l)}(\mathbf{x})]}$ (see Appendix A.5.1). (**C**): Comparison of BatchNorm running statistics before and after RESET in the merged VGG16-BN model. (**D**): Performance evaluation of merging on linear networks, with and without residual connections. Solid curves correspond to WM-based merging.

Section 4.2 revealed that the vanishing feature phenomenon in merged linear networks results in variance collapse and unstable performance. To generalize this analysis, we formalize Definition 5.1,

---
[1]For the rationale behind the "$< 1$" assumption, see Appendix A.5.2.

which extends the concept to accommodate the complexities introduced by non-linear activation functions. Notably, this definition aligns with Definition 4.2 in the case of linear networks. Using the notation from Equation 2, we define $\mathbf{g}^{(l)}(\mathbf{x}) = \mathbf{f}^{(l)}(\mathbf{x}) - \mathbf{f}^{(l)}(\mathbf{0})$, representing the input-induced features, and $\mathbf{h}^{(l)} = \mathbf{f}^{(l)}(\mathbf{0})$, capturing the input-independent offset at layer $l$. In our experiments, we observed a pronounced vanishing feature phenomenon in the merged model across diverse settings, rendering the model's output largely independent of its input, as shown in Figure 2 (**A & B**). This observation underscores the importance of understanding this phenomenon to enhance merging performance, as detailed in the subsequent sections.

**Definition 5.1** ($\varepsilon_{\mathcal{D},\mathbf{f}}$-**Vanishing Feature**). For a neural network $\mathbf{f}$ of depth $L$, there is an $\varepsilon_{\mathcal{D},\mathbf{f}}$-*vanishing feature* over data distribution $\mathcal{D}$ if there exists a non-increasing upper bound sequence $(\varepsilon_{\mathcal{D},\mathbf{f}}^{(l)})_{l=1}^{L}$ and a small constant $\varepsilon_{\mathcal{D},\mathbf{f}} > 0$ such that $\mathbb{E}_{\mathbf{x}\sim\mathcal{D}}[\|\mathbf{f}^{(l)}(\mathbf{x}) - \mathbf{f}^{(l)}(\mathbf{0})\|] \leq \varepsilon_{\mathcal{D},\mathbf{f}}^{(l)}$ for $l \in [L]$, and $\varepsilon_{\mathcal{D},\mathbf{f}}^{(L)} \leq \varepsilon_{\mathcal{D},\mathbf{f}}$.

## 5.2. Model Merging from the Lens of Vanishing Feature

Building on the analyses in Section 4, this section demonstrates how the vanishing feature phenomenon unifies the understanding of various techniques commonly employed in merging models trained from different initializations. Moreover, it leads to a novel finding that residual connections can effectively mitigate the vanishing feature.

**Permutation.** As discussed in Section 4.2, the parameter magnitude in merged models often decreases, exacerbating the vanishing feature phenomenon. For linear networks, the weights at the $l$-th layer of the midpoint model $\mathbf{f}_{0.5}$ can be expressed as: $\mathbf{W}_{0.5}^{(l)} = \frac{\mathbf{W}_0^{(l)}+\mathbf{W}_1^{(l)}}{2} = \mathbf{W}_0^{(l)} \odot \frac{\mathbf{1}_{m\times n}+\mathbf{W}_1^{(l)}\oslash\mathbf{W}_0^{(l)}}{2} = \mathbf{W}_1^{(l)} \odot \frac{\mathbf{1}_{m\times n}+\mathbf{W}_0^{(l)}\oslash\mathbf{W}_1^{(l)}}{2}$, where $\odot$ and $\oslash$ denote element-wise multiplication and division, respectively, and $m$ and $n$ are the matrix dimensions. This implies that *merging is essentially scaling the parameters of edge models by the scaling matrix*, $\mathbf{W}_0^{(l)} \oslash \mathbf{W}_1^{(l)}$. When $\mathbf{W}_0^{(l)}$ and $\mathbf{W}_1^{(l)}$ differ significantly (e.g., due to permutation invariance), the scaling matrix deviates from the identity, reducing the contribution of at least one model's parameters to the merged model. WM-based merging mitigates this issue by applying a permutation to align $\mathbf{W}_0^{(l)}$ and $\mathbf{W}_1^{(l)}$, effectively making the scaling matrix closer to the identity. This preserves parameter scales and reduces the vanishing feature phenomenon. Similar principles apply to AM-based merging.

**REPAIR.** Section 4.2 showed that vanished input-induced features allow bias terms to dominate model outputs, degrading performance. REPAIR addresses this by normalizing activations and applying an affine transformation (Equation 1), scaling activations by the standard deviation from edge models, $\overline{\mathrm{Std}_{\mathbf{x}}[f^{(l,i)}(\mathbf{x})]}$. This primarily *amplifies input-induced features* while canceling out input-independent offsets during normalization, effectively mitigating the vanishing feature phenomenon.

**Batch/Layer Normalization.** As elaborated in Section 2 and Appendix A.3, performing REPAIR on layers with batch normalization only involves recalculating the running mean $\mathbb{E}_{\mathbf{x}}[f_\alpha^{(l,i)}]$ and running standard deviation $\mathrm{Std}_{\mathbf{x}}[f_\alpha^{(l,i)}]$ in BatchNorm layers. As shown in Figure 2 (**C**), RESET significantly reduces the running standard deviation while leaving the running mean largely unchanged. Since the centered activation $f_\alpha^{(l,i)}(\mathbf{x}) - \mathbb{E}_{\mathbf{x}}[f_\alpha^{(l,i)}(\mathbf{x})]$ in Equation 1 is normalized by the running standard deviation, RESET *effectively amplifies input-induced features*. This aligns with the above interpretation of REPAIR. Layer normalization achieves a similar effect by directly normalizing activations using real-time statistics. As shown in Figure 2 (**B**), the vanishing feature is notably less pronounced in merged ResNet20-LN models.

**Depth & Residual Connection.** Section 4.2 showed that the vanishing feature phenomenon becomes more severe as model depth increases, consistent with previous findings that merging performance degrades in deeper models [10, 11]. We notice that the residual connections, however, help mitigate this issue by preserving early latent features. For example, in a linear network with equal-width layers and residual connections that add each layer's input to its output, the intermediate activation

can be expressed as (see Appendix A.1 for derivation):

$$\mathbf{f}_\alpha^{(l)}(\mathbf{x}) = \sum_{j=0}^{l} \mathbf{e}_j(\mathbf{W}_\alpha^{(1)}, \ldots, \mathbf{W}_\alpha^{(l)})\mathbf{x} + \sum_{j=1}^{l} \sum_{k=0}^{l-j} \mathbf{e}_k(\mathbf{W}_\alpha^{(j+1)}, \ldots, \mathbf{W}_\alpha^{(l)})\mathbf{b}_\alpha^{(j)}, \qquad (3)$$

where $\mathbf{e}_k(\mathbf{M}_1, \ldots, \mathbf{M}_N) = \sum_{1 \le j_1 \ldots < j_k \le N} \prod_{i=1}^{k} \mathbf{M}_{j_i}$ is the elementary symmetric polynomial of any sequence of matrices $\mathbf{M}_i$, $i \in [N]$. Setting $j = l$ in $\mathbf{g}_\alpha^{(l)}$ and $k = l - j$ in $\mathbf{h}_\alpha^{(l)}$ reduces to Equation 2. By having residual connections, the vanishing feature issue is alleviated since the elementary symmetric polynomial of low order, e.g. $\mathbf{e}_1(\mathbf{W}_\alpha^{(1)}, \ldots, \mathbf{W}_\alpha^{(l)})\mathbf{x} = \sum_{j=1}^{l} \mathbf{W}_\alpha^{(j)}\mathbf{x}$, also remain in $\mathbf{g}_\alpha^{(l)}(\mathbf{x})$. Figure 2 (**D**) shows that adding residual connections to linear networks generally improves merging.

## 5.3. Enhanced Normalization for Vanishing Feature Mitigation

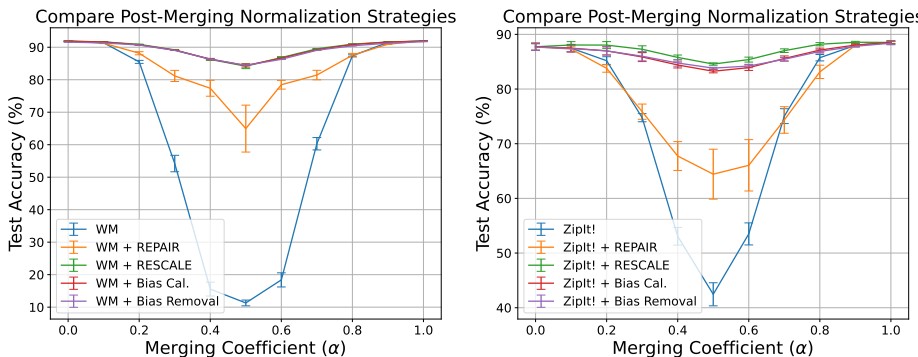

Figure 3: **Enhancing merging performance by addressing the vanishing feature issue.** We compare four normalization strategies: REPAIR, RESCALE, bias removal, and bias calibration ("Bias Cal."). The latter three are specifically designed to mitigate the vanishing feature phenomenon. Permutations derived via weight matching (**Left**) and ZipIt! (**Right**) are applied prior to merging.

The last section showed that REPAIR mitigates the vanishing feature issue by amplifying input-induced features. However, REPAIR has a potential limitation: an input-independent offset, $\overline{\mathbb{E}_\mathbf{x}[f^{(l,i)}(\mathbf{x})]}$, is reintroduced after scaling (Equation 1). This offset can affect the output, particularly when input-induced features are not sufficiently preserved, such as under insufficient scaling, misaligned permutations, or in very deep networks. We study this by designing three post-merging normalization strategies targeting vanishing feature mitigation and comparing their performances with REPAIR:

- **Bias Removal.** This baseline eliminates all bias terms by setting them to zero, providing a straightforward approach to remove input-independent offsets.
- **RESCALE.** A modification of REPAIR, "RESCALE" removes the activation shift in Equation 1 as: $f_\alpha^{(l,i)}(\mathbf{x}) \leftarrow \frac{f_\alpha^{(l,i)}(\mathbf{x})}{\mathrm{Std}_\mathbf{x}[f_\alpha^{(l,i)}(\mathbf{x})]} \cdot \overline{\mathrm{Std}_\mathbf{x}[f^{(l,i)}(\mathbf{x})]}$.
- **Bias Calibration.** As discussed earlier, bias terms dominate the outputs of deeper layers when input-induced features decay below the scale of input-independent offsets. Revisiting Equation 2, if the activation scale decays by a factor of $c$ at each layer, $\mathbf{g}_\alpha^{(l)}(\mathbf{x})$ decays by $c^l$, while $\tilde{\mathbf{h}}_\alpha^{(l)}(\mathbf{b}_j)$ decays more slowly by $c^{l-j}$. We hypothesize that directly calibrating $\tilde{\mathbf{h}}_\alpha^{(l)}(\mathbf{b}_j)$ could mitigate this imbalance. To test this, we adopt the homogeneous interpolation method from Wang et al. [34]. For the $l$-th layer, it scales the bias terms by depth as follows: $\mathbf{b}_\alpha^{(l)} \leftarrow (1 - \alpha)^l \cdot \mathbf{b}_0^{(l)} + \alpha^l \cdot \mathbf{b}_1^{(l)}$.

We present the results of merging two VGG16 models trained on CIFAR-10 in Figure 3. Permutations were applied prior to merging, using either weight matching or ZipIt!. For bias removal, removing bias terms from the last 5-6 layers yielded the best results (details in Appendix A.6.1). Notably, REPAIR underperformed compared to the three targeted strategies, especially at the midpoint. The three methods performed similarly with weight matching, but RESCALE outperformed others on

ZipIt!. The results support our hypothesis that REPAIR's incomplete handling of the vanishing feature may lead to inconsistent performance. However, we observed that REPAIR remained most effective for architectures with residual connections or on complex datasets like ImageNet, likely because the vanishing feature dynamics in these settings are more complex, requiring more tailored strategies to address them effectively. For instance, Equation 3 suggests the current bias calibration strategy is suboptimal in the presence of residual connections. *In summary, our results reveal that effectively addressing the vanishing feature issue in post-merging techniques can lead to performance improvements.*

## 5.4. Preserve-First Merging

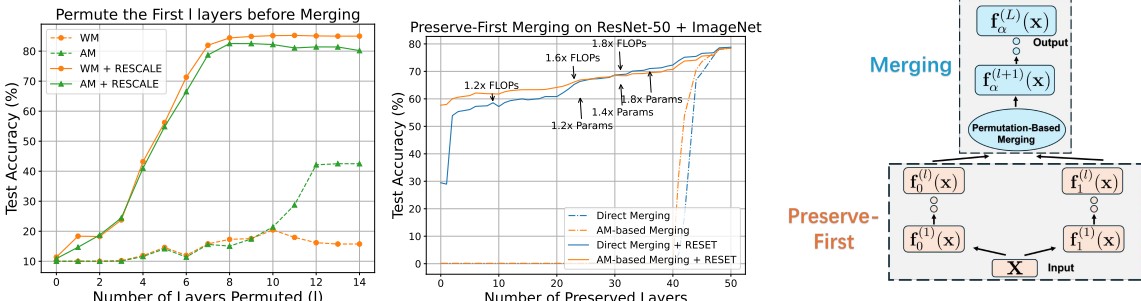

Figure 4: **Left**: Evaluation of WM/AM-based merging with only the first $l$ layers premuted before merging. Edge models are VGG16 trained on CIFAR-10. **Mid**: Evaluating PFM on merging two ResNet50s trained on ImageNet. **Right**: Illustration of PFM. The input is duplicated and processed independently through preserved layers from each edge model, retaining the latent representations. Outputs of the $l$-th layers are merged using permutations into a unified representation, which is then passed through the fully merged subsequent layers. Normalization is applied to the merged layers.

Previous sections showed that post-merging normalization could effectively mitigate the vanishing feature issue. However, its efficacy diminishes in more challenging settings, potentially due to misaligned permutations performed before merging. As analyzed in Section 5.2, permutations introduce fine-grained element-wise scaling to intermediate activations, whereas normalization applies coarser channel-wise adjustments after merging. To investigate their interplay, we measured merging performance when permutations were applied only to the first $l$ layers. As shown in Figure 4 (**Left**), performance improved with increasing $l$, peaking when $l$ was sufficiently large. This indicates that preserving features in earlier layers is crucial for effective merging. However, performance saturates beyond a certain value of $l$ and remains below that of the edge models, suggesting that feature loss persists. We hypothesize that even with permutation and normalization, some loss occurs at each layer, compounding over the network and fundamentally limiting merging performance.

While finding the optimal permutation could mitigate this issue, it is computationally intractable due to its NP-hard nature [11]. To address this, we propose the "Preserve-First Merging" (PFM) strategy. Inspired by the "partial zipping" operation from Stoica et al. [13], PFM preserves the first $l$ layers of the edge models unmerged, as shown in Figure 4 (**Right**). This design leverages well-trained features in edge models to secure an optimal upper bound for performance while merging the subsequent layers to ensure computational efficiency. Comparisons between PFM and partial zipping are provided in Appendix A.8.2. Table 1 presents the performance of PFM for merging VGG16 models trained on CIFAR-10. The baseline methods include (1) direct merging, (2) activation-matching (AM)-based merging, and (3) AM-based merging with REPAIR applied post-merging. Notably, merging the first eight layers surpasses the performance of both edge models, achieving this without requiring post-training—a first in this context. Unlike prior works that often increase model width to enhance performance, PFM achieves this with only a 20% increase in parameters [10, 11]. While PFM introduces additional FLOPs due to its multi-start design, its suitability for parallel computation maintains comparable runtime efficiency. On ImageNet (Figure 4, **Mid**), achieving performance comparable to the edge models requires a significantly larger value of $l$. We hypothesize this is due

to the limitations of existing normalization techniques in mitigating feature vanishing in merged layers, as discussed in Section 5.3. Additional results are provided in Appendix A.8, including the evaluation with the advanced CCA merging algorithm [23] and on transformers [35].

Table 1: **Performance of Preserve-First Merging (PFM) at the midpoint** ($\alpha = 0.5$) **on two VGG16 models trained on CIFAR-10.** PFM is evaluated with direct merging, permutation-based merging, and post-merging normalization techniques. Permutations are derived via activation matching. Test accuracy (mean $\pm$ standard deviation) is reported. $\text{PFM}_{l/L}$ indicates that the first $l$ out of $L$ layers are preserved (unmerged). "Bias Cal." refers to the bias calibration strategy from Section 5.3. The highest accuracy per row is in bold, and values exceeding the average accuracy of the two edge models are shaded in gray. Baseline results are boxed.

| | | Activation Matching | | | | | |
| Model | Direct | W/o Normalization | REPAIR | RESCALE | Bias Cal. | Params (M) | FLOPs (G) |
|---|---|---|---|---|---|---|---|
| Model 1 | $91.88_{\pm 0.02}\%$ | | | | | 14.72 (1×) | 0.31 (1×) |
| Model 2 | $91.67_{\pm 0.21}\%$ | | | | | 14.72 (1×) | 0.31 (1×) |
| Ensemble | $92.93_{\pm 0.29}\%$ | | | | | 29.44 (2×) | 0.63 (2×) |
| $\text{PFM}_{0/14}$ | $\boxed{10.00}_{\pm 0.00}\%$ | $\boxed{46.50}_{\pm 1.59}\%$ | $\boxed{39.09}_{\pm 4.42}\%$ | $84.68_{\pm 0.85}\%$ | $82.94_{\pm 1.41}\%$ | 14.72 (1×) | 0.31 (1×) |
| $\text{PFM}_{3/14}$ | $10.03_{\pm 0.05}\%$ | $62.40_{\pm 0.64}\%$ | $39.97_{\pm 2.55}\%$ | $\mathbf{86.77}_{\pm 0.06}\%$ | $85.53_{\pm 0.16}\%$ | 14.83 (1.008×) | 0.37 (1.19×) |
| $\text{PFM}_{6/14}$ | $10.92_{\pm 0.38}\%$ | $85.81_{\pm 0.40}\%$ | $50.61_{\pm 2.13}\%$ | $\mathbf{90.88}_{\pm 0.27}\%$ | $90.03_{\pm 0.32}\%$ | 15.87 (1.08×) | 0.47 (1.49×) |
| $\text{PFM}_{8/14}$ | $26.35_{\pm 6.54}\%$ | $91.15_{\pm 0.13}\%$ | $87.37_{\pm 1.73}\%$ | $\mathbf{92.43}_{\pm 0.23}\%$ | $91.84_{\pm 0.05}\%$ | 17.64 (1.2×) | 0.52 (1.67×) |
| $\text{PFM}_{12/14}$ | $\mathbf{92.88}_{\pm 0.23}\%$ | $\mathbf{92.88}_{\pm 0.16}\%$ | $92.85_{\pm 0.15}\%$ | $92.87_{\pm 0.21}\%$ | $92.71_{\pm 0.15}\%$ | 27.07 (1.84×) | 0.62 (1.97×) |

## 5.5. Beyond Merging: Vanishing Feature in Pruning

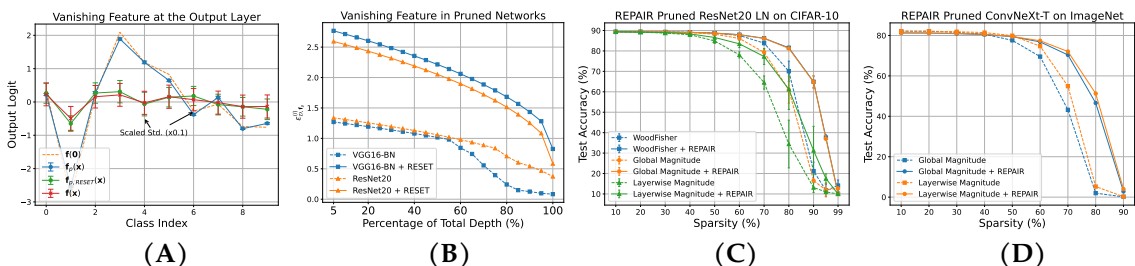

(A)  (B)  (C)  (D)

Figure 5: **Vanishing feature in pruned models.** (**A & B**): One-shot global magnitude pruning is applied with sparsity levels of 80% for VGG16-BN and 85% for ResNet20, both trained on CIFAR-10. (**A**) shows average output logits of a pruned VGG16-BN model (standard deviation scaled by $0.1\times$ for clarity). (**B**) plots the upper bound sequence $(\varepsilon_{\mathcal{D}, \mathbf{f}_p}^{(l)})_{l=1}^{L}$, normalized by $\mathbb{E}_{\mathbf{x}}[\mathbf{g}^{(l)}(\mathbf{x})]$, highlighting mitigation of the vanishing feature by RESET. (**C & D**): Similar to merging, applying REPAIR to the pruned model can significantly improve its performance, particularly at a higher sparsity. Lightweight hyperparameters were used for the WoodFisher pruner to reduce computational costs.

While the previous sections focused on vanishing features in merged models, Definition 5.1 applies more broadly to general neural networks. Building on this foundation, we extend the investigation to model pruning [19], providing a preliminary study in this context. Our experiments reveal a similar vanishing feature phenomenon in pre-trained neural networks subjected to one-shot pruning at high sparsity. As shown in Figures 5 (**A & B**), input-induced features progressively diminish across layers, rendering the model output nearly independent of the input. To address this issue, we adopt practices from model merging and propose a post-pruning normalization strategy. After pruning model $\mathbf{f}$ to $\mathbf{f}_p$, we applied REPAIR with reference statistics from $\mathbf{f}$ to normalize $\mathbf{f}_p$:

$$f_p^{(l,i)}(\mathbf{x}) \leftarrow \frac{f_p^{(l,i)}(\mathbf{x}) - \mathbb{E}_{\mathbf{x}}[f_p^{(l,i)}(\mathbf{x})]}{\text{Std}_{\mathbf{x}}[f_p^{(l,i)}(\mathbf{x})]} \cdot \text{Std}_{\mathbf{x}}[f^{(l,i)}(\mathbf{x})] + \mathbb{E}_{\mathbf{x}}[f^{(l,i)}(\mathbf{x})],$$

where the notation aligns with Equation 1. For layers followed by batch normalization, REPAIR recalculates running statistics (as in RESET) by updating BatchNorm layers—a common practice in pruning literature (e.g., Li et al. [36]), though not previously studied from the perspective of

vanishing features. For other layers, REPAIR performs affine transformation by inserting BatchNorm layers, extending post-pruning normalization to diverse architectures. We evaluated REPAIR on three one-shot pruning methods: global/layerwise magnitude pruning [19], diagonal-Fisher pruning, WoodFisher pruning [37], and Wanda [33]. Figures 5 (**C & D**) illustrate results for ResNet20-LN on CIFAR-10 and ConvNeXt-T on ImageNet. In general, we found REPAIR consistently enhanced the performance of pruned models, especially at high sparsity. Additional results and pruning details are provided in Appendix A.7. Compared to other post-pruning methods, REPAIR is efficient and lightweight: (1) it requires no post-training, (2) it adds no extra parameters, as inserted BatchNorm layers can be absorbed into preceding layers [12], and (3) it relies on minimal data for BatchNorm statistics estimation (see Appendix A.7). These qualities make it particularly suitable for resource-constrained scenarios. Additionally, REPAIR integrates seamlessly into existing pruning pipelines and can complement other post-pruning techniques.

## 6. Conclusion

This work identified and studied the vanishing feature phenomenon, providing a unified explanation for challenges in model merging such as variance collapse and insights into various merging techniques. The study demonstrates that incorporating considerations for the vanishing feature can enhance merging performance. The proposed preserve-first merging strategy effectively preserves early-layer features and improves merging performance without post-training. Additionally, the vanishing feature phenomenon is shown to extend to model pruning, where addressing it with modified normalization techniques significantly enhances one-shot pruning performance at high sparsity. These findings offer novel insights into understanding and improving model merging and underscoring the value of investigating vanishing features in broader contexts.

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

# A. Appendix

## A.1. Proofs

**Proposition A.1** (Restatement of Equation 2). *Consider two linear networks $\mathbf{f}_0$ and $\mathbf{f}_1$ of depth $L$, defined as:*

$$\mathbf{f}_i(\mathbf{x}) = \mathbf{f}_i^{(L)}(\mathbf{x}), \quad \mathbf{f}_i^{(l)}(\mathbf{x}) = \mathbf{W}_i^{(l)}\mathbf{f}_i^{(l-1)}(\mathbf{x}) + \mathbf{b}_i^{(l)}, \quad \mathbf{f}_i^{(0)}(\mathbf{x}) = \mathbf{x},$$

*where $i \in \{0, 1\}$, $l \in [L]$, $\mathbf{W}_i^{(l)}$ is the weight matrix, and $\mathbf{b}_i^{(l)}$ is the bias vector of the $l_{th}$ layer.*

*The merged network $\mathbf{f}_\alpha$ is defined as:*

$$\mathbf{f}_\alpha(\mathbf{x}) = \mathbf{f}_\alpha^{(L)}(\mathbf{x}), \quad \mathbf{f}_\alpha^{(l)}(\mathbf{x}) = ((1-\alpha)\mathbf{W}_0^{(l)} + \alpha\mathbf{W}_1^{(l)})\mathbf{f}_\alpha^{(l-1)}(\mathbf{x}) + ((1-\alpha)\mathbf{b}_0^{(l)} + \alpha\mathbf{b}_1^{(l)}),$$

*with $\mathbf{f}_\alpha^{(0)}(\mathbf{x}) = \mathbf{x}$ and $\alpha \in [0, 1]$.*

*The intermediate activations of the merged network at layer $l$ can then be expressed as:*

$$\mathbf{f}_\alpha^{(l)}(\mathbf{x}) = \prod_{j=1}^{l}((1-\alpha)\mathbf{W}_0^{(j)} + \alpha\mathbf{W}_1^{(j)})\mathbf{x} + \sum_{j=1}^{l}\prod_{k=j+1}^{l}((1-\alpha)\mathbf{W}_0^{(k)} + \alpha\mathbf{W}_1^{(k)})((1-\alpha)\mathbf{b}_0^{(j)} + \alpha\mathbf{b}_1^{(j)}).$$

*Proof.* We prove the expression by induction on the layer index $l$.

**Base Case** $(l = 1)$**:** From the definition of the merged network:

$$\mathbf{f}_\alpha^{(1)}(\mathbf{x}) = ((1-\alpha)\mathbf{W}_0^{(1)} + \alpha\mathbf{W}_1^{(1)})\mathbf{f}_\alpha^{(0)}(\mathbf{x}) + ((1-\alpha)\mathbf{b}_0^{(1)} + \alpha\mathbf{b}_1^{(1)}).$$

Since $\mathbf{f}_\alpha^{(0)}(\mathbf{x}) = \mathbf{x}$, we have:

$$\mathbf{f}_\alpha^{(1)}(\mathbf{x}) = ((1-\alpha)\mathbf{W}_0^{(1)} + \alpha\mathbf{W}_1^{(1)})\mathbf{x} + ((1-\alpha)\mathbf{b}_0^{(1)} + \alpha\mathbf{b}_1^{(1)}),$$

which matches the expression for $\mathbf{f}_\alpha^{(l)}(\mathbf{x})$ in the proposition.

**Inductive Step:** Assume the expression holds for layer $l - 1$:

$$\mathbf{f}_\alpha^{(l-1)}(\mathbf{x}) = \prod_{j=1}^{l-1}((1-\alpha)\mathbf{W}_0^{(j)} + \alpha\mathbf{W}_1^{(j)})\mathbf{x} + \sum_{j=1}^{l-1}\prod_{k=j+1}^{l-1}((1-\alpha)\mathbf{W}_0^{(k)} + \alpha\mathbf{W}_1^{(k)})((1-\alpha)\mathbf{b}_0^{(j)} + \alpha\mathbf{b}_1^{(j)}).$$

For layer $l$, from the network definition:

$$\mathbf{f}_\alpha^{(l)}(\mathbf{x}) = ((1-\alpha)\mathbf{W}_0^{(l)} + \alpha\mathbf{W}_1^{(l)})\mathbf{f}_\alpha^{(l-1)}(\mathbf{x}) + ((1-\alpha)\mathbf{b}_0^{(l)} + \alpha\mathbf{b}_1^{(l)}).$$

Substituting the inductive hypothesis:

$$\mathbf{f}_\alpha^{(l)}(\mathbf{x}) = ((1-\alpha)\mathbf{W}_0^{(l)} + \alpha\mathbf{W}_1^{(l)})\left( \prod_{j=1}^{l-1}((1-\alpha)\mathbf{W}_0^{(j)} + \alpha\mathbf{W}_1^{(j)})\mathbf{x} \right.$$
$$+ \sum_{j=1}^{l-1}\prod_{k=j+1}^{l-1}((1-\alpha)\mathbf{W}_0^{(k)} + \alpha\mathbf{W}_1^{(k)})((1-\alpha)\mathbf{b}_0^{(j)} + \alpha\mathbf{b}_1^{(j)}) \Bigg)$$
$$+ ((1-\alpha)\mathbf{b}_0^{(l)} + \alpha\mathbf{b}_1^{(l)}).$$

Expanding the terms:

$$\mathbf{f}_\alpha^{(l)}(\mathbf{x}) = \prod_{j=1}^{l}((1-\alpha)\mathbf{W}_0^{(j)} + \alpha\mathbf{W}_1^{(j)})\mathbf{x}$$
$$+ \sum_{j=1}^{l-1}\prod_{k=j+1}^{l}((1-\alpha)\mathbf{W}_0^{(k)} + \alpha\mathbf{W}_1^{(k)})((1-\alpha)\mathbf{b}_0^{(j)} + \alpha\mathbf{b}_1^{(j)})$$
$$+ ((1-\alpha)\mathbf{b}_0^{(l)} + \alpha\mathbf{b}_1^{(l)}).$$

Recognizing the last term as the contribution from biases at layer $l$, we obtain:

$$\mathbf{f}_\alpha^{(l)}(\mathbf{x}) = \prod_{j=1}^{l}((1-\alpha)\mathbf{W}_0^{(j)} + \alpha\mathbf{W}_1^{(j)})\mathbf{x} + \sum_{j=1}^{l}\prod_{k=j+1}^{l}((1-\alpha)\mathbf{W}_0^{(k)} + \alpha\mathbf{W}_1^{(k)})((1-\alpha)\mathbf{b}_0^{(j)} + \alpha\mathbf{b}_1^{(j)}).$$

Thus, the proposition holds for layer $l$.

**Conclusion:** By induction, the proposition is true for all $l \in [L]$. $\qquad\square$

**Proposition A.2** (Restatement of Equation 3). *Consider a linear network $\mathbf{f}_\alpha$ of depth $L$ with residual connections at each layer. The network is defined as:*

$$\mathbf{f}_\alpha^{(l)}(\mathbf{x}) = \mathbf{W}_\alpha^{(l)}\mathbf{f}_\alpha^{(l-1)}(\mathbf{x}) + \mathbf{b}_\alpha^{(l)} + \mathbf{f}_\alpha^{(l-1)}(\mathbf{x}),$$

*where $\mathbf{W}_\alpha^{(l)} = (1-\alpha)\mathbf{W}_0^{(l)} + \alpha\mathbf{W}_1^{(l)}$, $\mathbf{b}_\alpha^{(l)} = (1-\alpha)\mathbf{b}_0^{(l)} + \alpha\mathbf{b}_1^{(l)}$, and $\mathbf{f}_\alpha^{(0)}(\mathbf{x}) = \mathbf{x}$. Here, $\alpha \in [0,1]$ determines the merging ratio of two networks $\mathbf{f}_0$ and $\mathbf{f}_1$.*

*The intermediate activation at layer $l$ can then be expressed as:*

$$\mathbf{f}_\alpha^{(l)}(\mathbf{x}) = \sum_{j=0}^{l}\mathbf{e}_j(\mathbf{W}_\alpha^{(1)}, \ldots, \mathbf{W}_\alpha^{(l)})\mathbf{x} + \sum_{j=1}^{l}\sum_{k=0}^{l-j}\mathbf{e}_k(\mathbf{W}_\alpha^{(j+1)}, \ldots, \mathbf{W}_\alpha^{(l)})\mathbf{b}_\alpha^{(j)},$$

*where $\mathbf{e}_k(\mathbf{M}_1, \ldots, \mathbf{M}_N) = \sum_{1 \le j_1 < \cdots < j_k \le N}\prod_{i=1}^{k}\mathbf{M}_{j_i}$ represents the elementary symmetric polynomial of order $k$ for matrices $\mathbf{M}_i$, $i \in [N]$.*

*Proof.* We prove the expression by induction on the layer index $l$.

**Base Case ($l = 1$):** From the definition of the network with residual connections:

$$\mathbf{f}_\alpha^{(1)}(\mathbf{x}) = \mathbf{W}_\alpha^{(1)}\mathbf{f}_\alpha^{(0)}(\mathbf{x}) + \mathbf{b}_\alpha^{(1)} + \mathbf{f}_\alpha^{(0)}(\mathbf{x}).$$

Since $\mathbf{f}_\alpha^{(0)}(\mathbf{x}) = \mathbf{x}$, we have:

$$\mathbf{f}_\alpha^{(1)}(\mathbf{x}) = \mathbf{W}_\alpha^{(1)}\mathbf{x} + \mathbf{b}_\alpha^{(1)} + \mathbf{x}.$$

This matches the expression for $\mathbf{f}_\alpha^{(l)}(\mathbf{x})$ when $l = 1$, as:

$$\mathbf{f}_\alpha^{(1)}(\mathbf{x}) = \mathbf{e}_0(\mathbf{W}_\alpha^{(1)})\mathbf{x} + \mathbf{e}_1(\mathbf{W}_\alpha^{(1)})\mathbf{x} + \mathbf{b}_\alpha^{(1)}.$$

**Inductive Step:** Assume the expression holds for layer $l-1$:

$$\mathbf{f}_\alpha^{(l-1)}(\mathbf{x}) = \sum_{j=0}^{l-1}\mathbf{e}_j(\mathbf{W}_\alpha^{(1)}, \ldots, \mathbf{W}_\alpha^{(l-1)})\mathbf{x} + \sum_{j=1}^{l-1}\sum_{k=0}^{l-1-j}\mathbf{e}_k(\mathbf{W}_\alpha^{(j+1)}, \ldots, \mathbf{W}_\alpha^{(l-1)})\mathbf{b}_\alpha^{(j)}.$$

For layer $l$, from the network definition:

$$\mathbf{f}_\alpha^{(l)}(\mathbf{x}) = \mathbf{W}_\alpha^{(l)}\mathbf{f}_\alpha^{(l-1)}(\mathbf{x}) + \mathbf{b}_\alpha^{(l)} + \mathbf{f}_\alpha^{(l-1)}(\mathbf{x}).$$

Substituting the inductive hypothesis:

$$\mathbf{f}_\alpha^{(l)}(\mathbf{x}) = \mathbf{W}_\alpha^{(l)}\left(\sum_{j=0}^{l-1}\mathbf{e}_j(\mathbf{W}_\alpha^{(1)}, \ldots, \mathbf{W}_\alpha^{(l-1)})\mathbf{x} + \sum_{j=1}^{l-1}\sum_{k=0}^{l-1-j}\mathbf{e}_k(\mathbf{W}_\alpha^{(j+1)}, \ldots, \mathbf{W}_\alpha^{(l-1)})\mathbf{b}_\alpha^{(j)}\right)$$

$$+ \mathbf{b}_\alpha^{(l)} + \sum_{j=0}^{l-1}\mathbf{e}_j(\mathbf{W}_\alpha^{(1)}, \ldots, \mathbf{W}_\alpha^{(l-1)})\mathbf{x} + \sum_{j=1}^{l-1}\sum_{k=0}^{l-1-j}\mathbf{e}_k(\mathbf{W}_\alpha^{(j+1)}, \ldots, \mathbf{W}_\alpha^{(l-1)})\mathbf{b}_\alpha^{(j)}.$$

Expanding the terms:

$$\mathbf{f}_\alpha^{(l)}(\mathbf{x}) = \sum_{j=0}^{l}\mathbf{e}_j(\mathbf{W}_\alpha^{(1)}, \ldots, \mathbf{W}_\alpha^{(l)})\mathbf{x} + \sum_{j=1}^{l}\sum_{k=0}^{l-j}\mathbf{e}_k(\mathbf{W}_\alpha^{(j+1)}, \ldots, \mathbf{W}_\alpha^{(l)})\mathbf{b}_\alpha^{(j)},$$

where we utilize the following property of elementary symmetric polynomial: for any matrices $\mathbf{M}_i$, $i \in [N+1]$, we have $\mathbf{e}_k(\mathbf{M}_1, \dots, \mathbf{M}_{N+1}) = \mathbf{M}_{N+1}\mathbf{e}_{k-1}(\mathbf{M}_1, \dots, \mathbf{M}_N) + \mathbf{e}_{k-1}(\mathbf{M}_1, \dots, \mathbf{M}_N)$ for $1 \leq k \leq N$, and $\mathbf{e}_{N+1}(\mathbf{M}_1, \dots, \mathbf{M}_{N+1}) = \mathbf{M}_{N+1}\mathbf{e}_N(\mathbf{M}_1, \dots, \mathbf{M}_N)$. The property is straightforward to check by the definition of elementary symmetric polynomial.

**Conclusion:** By induction, the proposition holds for all $l \in [L]$. $\qquad\square$

When the first layer lacks a residual connection, the intermediate activation can be similarly derived by treating $\mathbf{f}_\alpha^{(1)}$ as the new input $\mathbf{x}$ for subsequent layers. This approach reflects practical scenarios where the input, often a high-dimensional flattened image, does not directly align with the network's internal dimensions.

**Corollary A.3** (Intermediate Activation without Residual Connection at the First Layer). *Consider a linear network $\mathbf{f}_\alpha$ of depth $L$ with residual connections at all layers except the first one. The network is defined as:*

$$\mathbf{f}_\alpha^{(l)}(\mathbf{x}) = \begin{cases} \mathbf{W}_\alpha^{(l)}\mathbf{f}_\alpha^{(l-1)}(\mathbf{x}) + \mathbf{b}_\alpha^{(l)}, & \text{if } l = 1, \\ \mathbf{W}_\alpha^{(l)}\mathbf{f}_\alpha^{(l-1)}(\mathbf{x}) + \mathbf{b}_\alpha^{(l)} + \mathbf{f}_\alpha^{(l-1)}(\mathbf{x}), & \text{if } l > 1, \end{cases}$$

*where $\mathbf{W}_\alpha^{(l)} = (1-\alpha)\mathbf{W}_0^{(l)} + \alpha\mathbf{W}_1^{(l)}$, $\mathbf{b}_\alpha^{(l)} = (1-\alpha)\mathbf{b}_0^{(l)} + \alpha\mathbf{b}_1^{(l)}$, and $\mathbf{f}_\alpha^{(0)}(\mathbf{x}) = \mathbf{x}$.*

*The intermediate activation at layer $l$ can then be expressed as:*

$$\mathbf{f}_\alpha^{(l)}(\mathbf{x}) = \begin{cases} \mathbf{W}_\alpha^{(1)}\mathbf{x} + \mathbf{b}_\alpha^{(1)}, & \text{if } l = 1, \\ \sum_{j=1}^{l} \mathbf{e}_j(\mathbf{W}_\alpha^{(2)}, \dots, \mathbf{W}_\alpha^{(l)})\mathbf{f}_\alpha^{(1)}(\mathbf{x}) + \sum_{j=2}^{l}\sum_{k=0}^{l-j} \mathbf{e}_k(\mathbf{W}_\alpha^{(j+1)}, \dots, \mathbf{W}_\alpha^{(l)})\mathbf{b}_\alpha^{(j)}, & \text{if } l > 1. \end{cases}$$

*Here, $\mathbf{e}_k(\mathbf{M}_1, \dots, \mathbf{M}_N)$ is the elementary symmetric polynomial of order $k$ for matrices $\mathbf{M}_i$, $i \in [N]$.*

## A.2. More Related Work

Current model merging techniques are most effective when merging models fine-tuned from the same pre-trained checkpoint. For merging under the pretraining-finetuning framework, Yang et al. [5], Goddard et al. [6], Tang et al. [38] summarize recently developed merging techniques and benchmarks. In this setting, the direct merging between two models is also closely related to the phenomenon called *"linear mode connectivity"* (LMC), where the linear model connectivity exists between two solutions if the test loss of the merged model remains low for any merging ratio $\alpha \in [0, 1]$ [8, 9, 39–43].

For merging models trained with different initializations, Entezari et al. [10] conducted extensive experiments and conjectured that merging could achieve optimal performance after accounting for the permutation invariance of neural networks, assuming the model is wide enough. The claim was proved for a wide enough fully connected network with a single hidden layer at initialization. Although the paper failed to empirically validate this conjecture by finding desired permutations, a successful attempt was shown in Ainsworth et al. [11], where three different matching algorithms were derived for this purpose. Notably, these algorithms succeeded in obtaining a merged midpoint model that has comparable performance to the edge models between independently trained ResNet models on CIFAR-10 for the first time, though with an increased model width. Prior and concurrent work in this direction includes Benzing et al. [17] proposing a matching algorithm to align models at initialization and Singh and Jaggi [14], Imfeld et al. [15] deriving powerful optimal-transport-based algorithms. It's also noteworthy that the matching algorithms proposed in many works have some common points, e.g., the activation matching algorithm in Ainsworth et al. [11] and Singh and Jaggi [14] and the matching algorithm in Tatro et al. [44]. While a significant improvement was seen in Ainsworth et al. [11], Jordan et al. [12] emphasized that this success highly depended on the utilization of normalization layers in models, and the matching algorithms performed poorly on plain models without normalization layers. The authors attributed this to the issue of "variance collapse" and addressed it by proposing the REPAIR method. Different from the above approaches, Crisostomi et al. [20] optimized permutations globally across all layers, which is beneficial for merging multiple models. Similar to model fusion, several works use generalized permutation matrices to increase

the similarity between edge models' parameters. Peña et al. [21] proposed a Sinkhorn-operator-based method to find matrices that estimate the permutation matrices, Navon et al. [22] utilized the DWSNets [45] to solve the weight alignment problem, and Horoi et al. [23] align the model activation by performing Canonical Correlation Analysis. Despite the extensive empirical studies, the understanding of this field remains limited. A theoretical effort was recently provided in Ferbach et al. [18] from the lens of mean field regime and generalized the results to deep networks under some assumptions. Besides merging on the same dataset, several works explored aligning models trained on different datasets [11, 12, 16]. Pittorino et al. [46] utilized the permutation-based merging approaches to remove symmetry and revealed the structure of flat regions in the loss landscape. Different from the standard permutation-based practice, scaling invariance is also considered.

Our analysis of the vanishing feature phenomenon in model merging aligns with some insights presented by Wang et al. [34]. While their work examines the merging of models between initialization and after training, focusing on the distinct impacts of linearly interpolating weights and biases on the final output, our study concentrates on merging pre-trained networks. In Wang et al. [34], a theoretical framework was introduced, highlighting the dominant role of last-layer biases in determining the performance of merged models near initialization. In contrast, our work explores the entire interpolation path, with particular emphasis on improving the midpoint model. Lastly, this work introduces the vanishing feature phenomenon as a general characteristic of neural networks, extending its relevance beyond the specific contexts of merging or linear interpolation.

## A.3. Experimental Details

Table 2: **Training Settings for Different Experiments**. We set the momentum to 0.9 for the SGD optimizer. For training VGG16 and ResNet20 on CIFAR-10 and CIFAR-100, the learning rate starts at $0.1$ and decreases by a factor of 10 at iterations $32,000$ and $48,000$ (approximately epochs 82 and 123). For ResNet20-LN on CIFAR-10, the learning rate increases linearly from $10^{-6}$ to $0.1$ during the first epoch and then follows the same step decay schedule. For ImageNet models, we utilize pre-trained checkpoints from the `Torchvision` [47] package.

| Dataset | Model | Optimizer | Learning Rate | Schedule | Batch Size | Weight Decay | Epochs |
|---------|-------|-----------|---------------|----------|------------|--------------|--------|
| MNIST | Linear Network | Adam | 0.01 | Constant | 128 | 1e-4 | 20 |
| CIFAR-10 | VGG16(-BN/LN) | SGD | 0.1 | Step Decay | 128 | 1e-4 | 160 |
| CIFAR-100 | VGG16(-BN/LN) | SGD | 0.1 | Step Decay | 128 | 1e-4 | 160 |
| CIFAR-10 | ResNet20 | SGD | 0.1 | Step Decay | 128 | 1e-4 | 160 |
| CIFAR-100 | ResNet20 | SGD | 0.1 | Step Decay | 128 | 1e-4 | 160 |
| CIFAR-10 | ResNet20-LN | SGD | 0.1 | Warm Up + Step Decay | 128 | 1e-4 | 160 |
| CIFAR-100 | ResNet20-LN | SGD | 0.1 | Warm Up + Step Decay | 128 | 1e-4 | 160 |
| ImageNet | ResNet50 | Pretrained | - | - | - | - | - |
| ImageNet | ConvNeXt-T | Pretrained | - | - | - | - | - |

**Model Architecture.** For the residual connection experiments in Section 5.2, we use 5-layer linear networks with a width of 32 per layer, where residual connections are added between the start and end of each layer except the first and last. For the linear network experiments in Section 4.2, we use 8-layer linear networks with the same width of 32. In CIFAR experiments, the VGG16 model is modified to include only a single linear classifier. We use "-BN" and "-LN" to denote models with batch normalization and layer normalization, respectively. For VGG16-LN and ResNet20-LN, all BatchNorm layers are replaced with LayerNorm layers.

**Dataset.** For the MNIST and CIFAR-10/CIFAR-100 datasets, we use the data provided by `Torchvision` [47]. For ImageNet, we use the official release from its website[2]. We perform the standard preprocessing to each dataset.

- **MNIST**: Each image is centered normalized.
- **CIFAR-10/CIFAR-100**: Each image is centered normalized. For training data, a random cropping (size $= 32 \times 32$, padding $= 4$) and random horizontal flipping are applied to each image before the re-normalization.

---

[2]https://www.image-net.org/download.php

- **ImageNet**: For test data, each image is first resized to $256 \times 256$ and then center-cropped to $224 \times 224$. For the PFM experiments reported in Figure 4, the data pre-processing is based on the implementation in Stoica et al. [13]. For pruning experiments, the pre-processing is based on the implementation in Singh and Alistarh [37].

**Training Details.** Standard training settings are summarized in Table 2. Linear networks used in Section 4.2 are trained for 5 epochs. For training VGG16 (with weight decay $> 10^{-4}$), ResNet20 (4× width), ResNet20-LN, and VGG16-LN (initialized using Kaiming uniform initialization [48]), the learning rate increases linearly during the first epoch from $10^{-6}$ to $0.1$ and subsequently follows the step decay schedule.

For the experiments in Appendix A.4, the initial learning rate is selected from $0.01, 0.03, 0.05, 0.07, 0.1$, and the weight decay factor is chosen from $10^{-4}, 2 \times 10^{-4}, 3 \times 10^{-4}, 4 \times 10^{-4}, 5 \times 10^{-4}$. In ablation experiments, models are retrained for one additional epoch. For training VGG16-LN on MNIST, we adopt the settings of Ainsworth et al. [11], using Kaiming uniform initialization and training for 25 epochs with SGD (momentum $= 0.9$). The learning rate increases linearly within the first epoch to a maximum of $0.001, 0.01$, or $0.1$, followed by a cosine annealing schedule. The batch size is 100, and weight decay is set to $5 \times 10^{-4}$.

**Measuring the Vanishing Feature.** Figures 2 (**B**), 5 (**B**), 11, 12, and 17 visualize the vanishing feature phenomenon across various settings. The non-increasing upper bound sequence defined in Definition 5.1 was used to measure the severeness of the vanishing feature. Results in Figure 17 were measured on the outputs of each layer. We first estimated the scale of the input-induced latent features in the merged model, i.e., $\mathbb{E}_{\mathbf{x} \sim \mathcal{D}} \left[ \| \mathbf{g}_\alpha^{(l)}(\mathbf{x}) \| \right] = \mathbb{E}_{\mathbf{x} \sim \mathcal{D}} [\| \mathbf{f}_\alpha^{(l)}(\mathbf{x}) - \mathbf{f}_\alpha^{(l)}(\mathbf{0}) \|]$ by feeding the entire training dataset to the model. A tight upper-bound sequence was then be recursively determined as: $\varepsilon_{\mathcal{D}, \mathbf{f}_\alpha}^{(L)} = \mathbb{E}_{\mathbf{x} \sim \mathcal{D}} \left[ \| \mathbf{g}_\alpha^{(L)}(\mathbf{x}) \| \right]$, $\varepsilon_{\mathcal{D}, \mathbf{f}_\alpha}^{(l)} = \max \left( \varepsilon_{\mathcal{D}, \mathbf{f}_\alpha}^{(l+1)}, \mathbb{E}_{\mathbf{x} \sim \mathcal{D}} \left[ \| \mathbf{g}_\alpha^{(l)}(\mathbf{x}) \| \right] \right)$, $l = L - 1, \ldots, 1$. In practice, $\mathbb{E}_{\mathbf{x} \sim \mathcal{D}} \left[ \| \mathbf{g}_\alpha^{(L)}(\mathbf{x}) \| \right]$ was first normalized by $\overline{\mathbb{E}_{\mathbf{x} \sim \mathcal{D}} \left[ \| \mathbf{g}^{(L)}(\mathbf{x}) \| \right]}$, i.e., the average scale of edge models' internal output, before estimating the upper-bound sequence. This balanced the latent feature's scale across layers. Figures 2 (**B**), 5 (**B**), 11, and 12 provide more detailed visualizations where the pre-BatchNorm outputs were also considered. While this introduces some noises, we added a smoothing factor $\gamma$ when estimating the upper-bound sequence for better visualization. Specifically, the recursive definition becomes $\varepsilon_{\mathcal{D}, \mathbf{f}_\alpha}^{(l)} = \max \left( \gamma \cdot \varepsilon_{\mathcal{D}, \mathbf{f}_\alpha}^{(l+1)} + (1 - \gamma) \cdot (1 - \frac{1}{L+1-l}) \cdot \varepsilon_{\mathcal{D}, \mathbf{f}_\alpha}^{(l+1)}, \mathbb{E}_{\mathbf{x} \sim \mathcal{D}} \left[ \| \mathbf{g}_\alpha^{(l)}(\mathbf{x}) \| \right] \right)$, $l = L-1, \ldots, 1$. The smoothing factor was chosen as $1.4$ for the visualization.

**Merging Details.** For PFM$_{l/L}$ experiments, $l$ represents the number of preserved/unmerged layers, and $L$ denotes the total number of layers in the model. Only linear and convolutional layers (excluding those in shortcut connections in ResNet models) are counted, as they account for the majority of the parameters and computational costs in the model. We implemented Preserve-First Merging (PFM) based on the partial zipping implementation from ZipIt! [13]. For the weight-matching algorithm, we adopted the implementation provided by Ainsworth et al. [11]. Similarly, the activation matching algorithm was implemented based on Ainsworth et al. [11], Jordan et al. [12], Stoica et al. [13], and for the ZipIt! framework, we used the original implementation from Stoica et al. [13].

**Normalization Details.** For merging experiments, we independently train two models with the same architecture and training hyperparameters on the same dataset but with different initializations and then perform merging. For **RESET**, we reset all BatchNorm layer running statistics to the default and recompute them by feeding the training set through the model. In other words, the target statistics $\overline{\mathrm{Std}_{\mathbf{x}}[f^{(l,i)}(\mathbf{x})]}$ and $\overline{\mathbb{E}_{\mathbf{x}}[f^{(l,i)}(\mathbf{x})]}$ in Equation 1 are directly approximated by the weights and biases in the merged BatchNorm layers. This follows the standard practices in Entezari et al. [10], Ainsworth et al. [11]. For **REPAIR**, we add a BatchNorm layer after the target layer (preceding the activation function) and initialize its weights and biases to the target statistics $\overline{\mathrm{Std}_{\mathbf{x}}[f^{(l,i)}(\mathbf{x})]}$ and $\overline{\mathbb{E}_{\mathbf{x}}[f^{(l,i)}(\mathbf{x})]}$ as defined in Equation 1. RESET is then applied to the updated model. For **RESCALE**, we introduce a custom BatchScale layer that replaces BatchNorm in the REPAIR process. Unlike BatchNorm,

BatchScale fixes the biases and running means at zero while retaining scale adjustments. For post-pruning normalization, REPAIR follows the same process while the target statistics are taken from the original unpruned model. For **bias removal**, we set all bias terms in the target layers to zero. We provide pseudo-code of RESET and REPAIR in Algorithms 1 and 2, respectively. The pre-activation (i.e., the layer output before applying the activation function) is utilized for REPAIR and RESET, following Entezari et al. [10]. Whenever a permutation was applied to the model before merging, the statistics from the *permuted* model were adopted as the reference in REPAIR and RESCALE.

---

**Algorithm 1** RESET

---

**Require:** Model $\mathbf{f}$ (to apply RESET), Dataset $D$ (for BatchNorm statistics estimation)
**Ensure:** Model $\mathbf{f}_{\text{RESET}}$ with updated BatchNorm statistics
 1:
 2: **Initialize BatchNorm statistics:**
 3: **for** each BatchNorm layer $l$ in $\mathbf{f}$ **do**
 4:    Set running mean $\mu^{(l)} \leftarrow \mathbf{0}$
 5:    Set running standard deviation $\sigma^{(l)} \leftarrow \mathbf{1}$
 6: **end for**
 7:
 8: **Recompute BatchNorm statistics:**
 9: Pass dataset $D$ through the model $\mathbf{f}$ to compute activations
10: **for** each BatchNorm layer $l$ **do**
11:    Update running mean $\mu^{(l)}$ and standard deviation $\sigma^{(l)}$ based on $D$
12: **end for**
13:
14: **return** Model $\mathbf{f}_{\text{RESET}}$ with updated BatchNorm statistics

---

**Pruning Details.** The code for all pruning experiments is adapted from Singh and Alistarh [37] with merely several new lines of code for performing REPAIR and RESET, demonstrating the convenience when merging this technique with any existing pruning framework. Consequently, the same hyper-parameters for pruning as Singh and Alistarh [37] are adopted in this work. For instance, the fisher subsample and fisher mini-batch sizes for the WoodFisher pruner are set to 400 in CIFAR-10/CIFAR-100 pruning. For the model except ResNet50, results with sparsity rates of 10%, 20%, ..., 90%, 95%, and 99% are reported. For ResNet50, we prune the model up to a sparsity of 90%. For pruning ResNet50 on ImageNet, a simple hyperparameter setting of the WoodFisher pruner is utilized, e.g., fisher subsample size = 80 and fisher mini-batch size = 100, to save computational cost and avoid interference. Although this restricts the performance of the pruner (e.g., the WoodFisher performs similarly to the global magnitude pruning), a significant improvement after RESET has already been shown in our results and should be universal after merely changing some hyperparameters. We leave the efforts towards improving the SOTA results for future work.

## A.4. Impact of Learning Rate and Weight Decay on Merging

In our experiments, we discovered a novel insight: adopting a larger learning rate and stronger weight decay during edge model training significantly enhances merging performance, as shown in Figure 6. For instance, increasing the learning rate from $0.01$ to $0.1$ reduced the barrier by over $40\%$ for VGG16-LN and ResNet20, and increasing the weight decay factor from $1e-4$ to $5e-4$ lowered the post-WM barrier for VGG16 by more than $60\%$. Figures 7 (**A**) and (**C**) provide detailed visualizations across various settings. Notably, this improvement is not attributable to better edge model performance, as their accuracy remains similar. To validate this, we performed ablation studies by retraining models initially trained with a learning rate of $0.1$ for one epoch, using a reduced learning rate of $0.01$, until their test accuracy aligned with that of models originally trained with $0.01$. As shown in Figure 7 (**B**), the barrier between models originally trained with a larger learning rate remained significantly lower. A similar trend was observed for weight decay as depicted in Figure 7 (**D**). A weight decay factor of 1e-4 was adopted for one epoch of retraining. These findings suggest

**Algorithm 2** REPAIR

---

**Require:** Model $\mathbf{f}$ (to apply REPAIR), Dataset $D$, Reference models $\{\mathbf{f}_{r_i}\}_i$, Layers $\mathcal{L}$ to process, Coefficients $[\alpha_i]_i$ for weighted averaging
**Ensure:** Model $\mathbf{f}_{\text{REPAIR}}$ with updated BatchNorm statistics
 1: **Step 1: Add BatchNorm layers to target and reference models.**
 2: **for** each model $\tilde{\mathbf{f}} \in \{\mathbf{f}\} \cup \{\mathbf{f}_{r_i}\}_i$ **do**
 3:    **for** each layer $l \in \mathcal{L}$ **do**
 4:       Insert a BatchNorm layer after $l$.
 5:    **end for**
 6: **end for**
 7:
 8: **Step 2: Compute running statistics.**
 9: **for** each model $\mathbf{f}_r \in \{\mathbf{f}_{r_i}\}_i$ **do**
10:    Feed dataset $D$ through $\mathbf{f}_r$.
11:    Record running mean $\mu_r^{(l)}$ and standard deviation $\sigma_r^{(l)}$ for all $l \in \mathcal{L}$.
12: **end for**
13:
14: **Step 3: Compute reference statistics.**
15: **for** each layer $l \in \mathcal{L}$ **do**
16:    Compute weighted mean: $\mu_{\text{ref}}^{(l)} \leftarrow \sum_i \alpha_i \cdot \mu_{r_i}^{(l)}$.
17:    Compute weighted standard deviation: $\sigma_{\text{ref}}^{(l)} \leftarrow \sum_i \alpha_i \cdot \sigma_{r_i}^{(l)}$.
18: **end for**
19:
20: **Step 4: Assign reference statistics to $\mathbf{f}$.**
21: **for** each layer $l \in \mathcal{L}$ of $\mathbf{f}$ **do**
22:    Set BatchNorm weights and biases:
23:        $\mathbf{W}^{(l)} \leftarrow \sigma_{\text{ref}}^{(l)}, \mathbf{b}^{(l)} \leftarrow \mu_{\text{ref}}^{(l)}$.
24: **end for**
25:
26: **Step 5: Apply RESET to $\mathbf{f}$.**
27: Call Algorithm 1 on $\mathbf{f}$ using dataset $D$.
28:
29: **return** Model $\mathbf{f}_{\text{REPAIR}}$

---

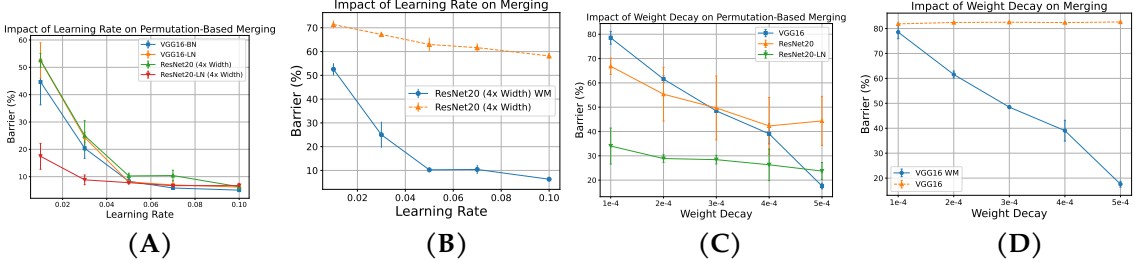

| (A) | (B) | (C) | (D) |

Figure 6: **Impact of Learning Rate (A & B) and Weight Decay (C & D) on Merging.** Edge models are trained on CIFAR-10 with different initializations. Permutations derived from the weight-matching algorithm are applied before merging in Figures (**A**) and (**C**) and the blue solid lines in Figures (**B**) and (**D**). RESET was applied for VGG16-BN and ResNet20 models. The linear barrier between the two edge models $\mathbf{f}_1$ and $\mathbf{f}_2$ is measured as $\mathcal{B}(\mathbf{f}_1, \mathbf{f}_2) := \max_{\alpha \in [0,1]} (1-\alpha) \cdot \mathrm{Acc}(\mathbf{f}_1) + \alpha \cdot \mathrm{Acc}(\mathbf{f}_2) - \mathrm{Acc}(\mathbf{f}_\alpha)$, where $\mathrm{Acc}(\mathbf{f})$ denotes the test accuracy of model $\mathbf{f}$. Results indicate that higher learning rates and stronger weight decay during edge model training improve permutation-based merging performance, with a smaller effect on direct merging.

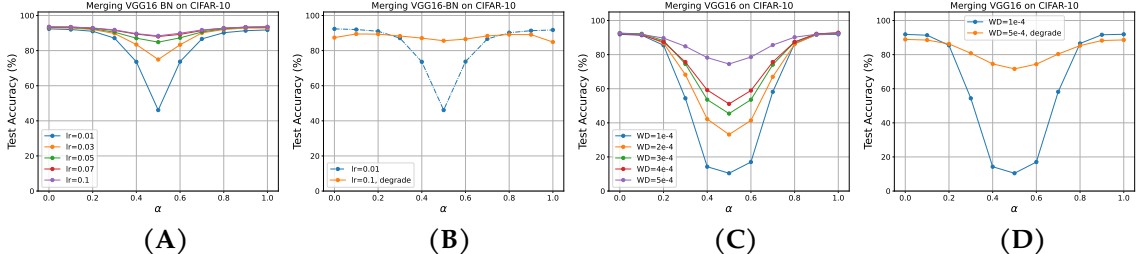

Figure 7: Figures (**A**) and (**C**) visualize the merging performances of models trained with varying learning rates and weight decay, with permutations from the weight-matching algorithm applied before merging. Figures (**B**) and (**D**) present ablation experiments, where edge model performance is intentionally degraded to compare merging outcomes. These results indicate that larger learning rates and stronger weight decay introduce implicit biases that enhance model merging effectiveness.

that larger learning rates and weight decay introduce implicit biases that are particularly beneficial for model merging.

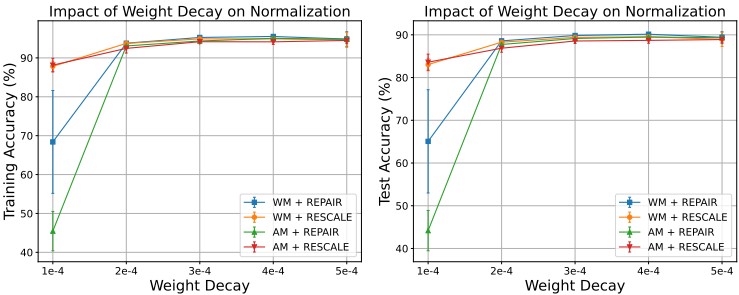

Figure 8: **Weight decay and Post-Merging Normalization.** Edge models are VGG16 trained on CIFAR-10. The merged midpoint model $\mathbf{f}_{0.5}$ is measured. Permutation-based merging is applied, followed by the REPAIR or RESCALE.

**Weight Decay and Post-Merging Normalization.** Weight decay also affects the effectiveness of post-merging normalization. As shown in Figure 8, REPAIR exhibited instability with smaller weight decay but became more reliable with stronger weight decay. In contrast, RESCALE remained robust across varying weight decay strengths. These findings align with our discussion in Section 5.3, highlighting that improved normalization techniques targeting the vanishing feature phenomenon can enhance merging performance.

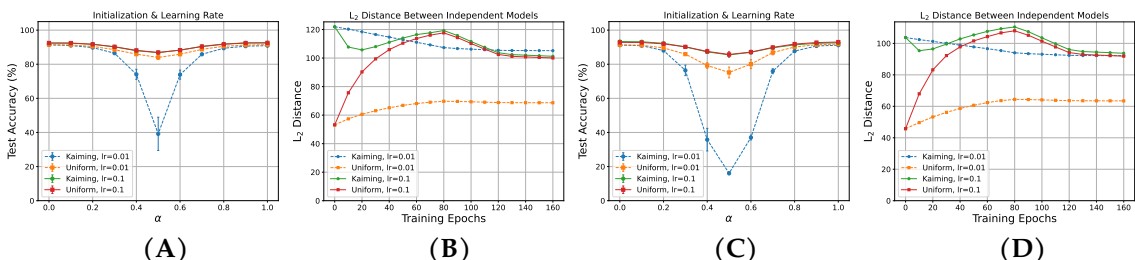

Figure 9: **Impact of initialization strategy.** Figures (**A**) and (**B**) correspond to merging VGG16-LN , while Figures (**C**) and (**D**) report results of merging merging ResNet20-LN (4x width). All models are trained on CIFAR-10. Figures (**B**) and (**D**) record the distance between the two edge models during training. Kaiming: Kaiming normal initialization. Uniform: Kaiming uniform initialization.

**Initialization Strategy.** During our study of learning rates, we noticed that initialization strategies can also impact merging effectiveness. We examined two standard initializations: Kaiming (normal)

initialization and (Kaiming) uniform initialization [48]. As shown in Figures 9 (**A**) and (**C**), while both strategies resulted in solutions with similar performance, the barrier was much larger between solutions with Kaiming initialization than those initialized uniformly. This difference disappeared when a larger learning rate was used. This phenomenon was observed in both VGG16-LN and ResNet20-LN models trained on CIFAR-10. We provide a preliminary analysis of this observation below.

Firstly, we describe the specific initialization strategies used in our experiments. For the Kaiming uniform initialization, we directly adopted the PyTorch [49] default. The Kaiming normal initialization is standard for training VGG, e.g., used in Torchvision. Specifically, for a 2D convolutional layer, the uniform initialization samples the weights and biases from the uniform distribution $U\left[\frac{-1}{\sqrt{k}}, \frac{1}{\sqrt{k}}\right]$, where $k = \text{num\_in} \times \text{kernel\_size}[0] \times \text{kernel\_size}[1]$; while the normal initialization samples the weights from the Gaussian distribution $N\left(0, \frac{2}{k}\right)$, where $k = \text{num\_out} \times \text{kernel\_size}[0] \times \text{kernel\_size}[1]$, and sets the bias to 0. For a linear layer, the uniform initialization samples weights and biases from $U\left[\frac{-1}{\sqrt{k}}, \frac{1}{\sqrt{k}}\right]$, where $k = \text{num\_in}$; while the normal initialization samples the weights from $N(0, 0.01)$ and sets bias to zero. This creates a larger variance at initialization for the normal initialization, which can be observed from the distance between two independently initialized models, as illustrated in Figures 9 (**B**) and (**D**).

After careful examination, we found that Kaiming initialization samples elements from a distribution with a larger variance, and hence, the initial distance between two independent models is larger. We report the $\ell_2$ distance between independent models during training in Figures 9 (**B**) and (**D**). We hypothesized that when trained with small learning rates, models tend to converge towards solutions close to the initialization, maintaining the large distance between the final solutions, making it challenging or even impossible to discover a permutation for effective matching. In other words, the scaling matrix $\mathbf{W}_0^{(l)} \oslash \mathbf{W}_1^{(l)}$ discussed in Section 5.2 is tricky to handle. On the other hand, the implicit bias of large learning rates mitigates the impact of initialization.

**Merging VGG16-LN on MNIST.** Moreover, we noticed that this finding also clarifies a failed attempt reported in Ainsworth et al. [11], where matching VGG16-LNs trained on MNIST yielded poor results. Indeed, the poor performance can be attributed to the learning rate adopted in the experiment being too small (0.001). As shown in Figure 10, increasing the learning rate led to immediate improvements in the merged model's performance.

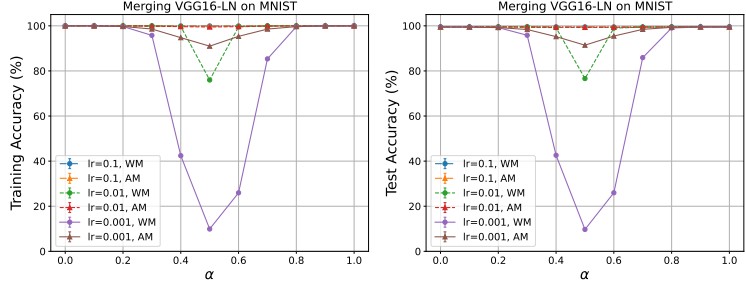

Figure 10: **Merging VGG16-LN trained on MNIST.** Report the average training and test accuracy of the merged model $\mathbf{f}_\alpha$. Permutations derived from weight matching and activation matching are applied prior to merging. Increasing the learning rate from 0.001 to 0.1 immediately improves the merging performance.

## A.5. Additional Results for Vanishing Feature and Merging

### A.5.1. Visualization of Vanishing Feature Mitigation

We provide visualizations on the effect of vanishing feature mitigation by permutation-based merging (Figure 11, **Left**), REPAIR and RESCALE (Figure 11, **Right**), and the bias removal and bias

calibration (Figure 12). Same as in Figures 2 (**B**) and 5 (**B**), the visualized upper bound sequence in Figure 11 $(\varepsilon_{\mathcal{D},\mathbf{f}_{0.5}}^{(l)})_{l=1}^{L}$ is normalized by $\overline{\mathbb{E}_{\mathbf{x}}[\mathbf{g}^{(l)}(\mathbf{x})]}$. We performed this normalization by dividing $\mathbb{E}_{\mathbf{x}\sim\mathcal{D}}[\|\mathbf{f}_{0.5}^{(l)}(\mathbf{x}) - \mathbf{f}_{0.5}^{(l)}(\mathbf{0})\|]$ by $\overline{\mathbb{E}_{\mathbf{x}\sim\mathcal{D}}[\|\mathbf{f}^{(l)}(\mathbf{x}) - \mathbf{f}^{(l)}(\mathbf{0})\|]} = \frac{\sum_{i=0}^{1}\mathbb{E}_{\mathbf{x}\sim\mathcal{D}}[\|\mathbf{f}_{i}^{(l)}(\mathbf{x}) - \mathbf{f}_{i}^{(l)}(\mathbf{0})\|]}{2}$ in Definition 5.1. Results align with our analysis in Sections 5.2 and 5.3. For the bias removal and bias calibration, we directly measure the magnitude of input-induced features and the input-independent offset.

**A.5.2. Theoretical Insights into the Vanishing Feature Phenomenon**

In this section, we provide some theoretical insights into the phenomenon of vanishing features.

**General Case.** The "Magnitude Perspective" paragraph in Section 4.2 illustrates a theoretical insight with empirical evidence. We restate the discussions for consistency. Consider a linear network of depth $L : \mathbf{f}(\mathbf{x}) = \mathbf{f}^{(L)}(\mathbf{x})$, $\mathbf{f}^{(l)}(\mathbf{x}) = \mathbf{W}^{(l)}\mathbf{f}^{(l-1)}(\mathbf{x}) + \mathbf{b}^{(l)}$, $\mathbf{f}^{(0)}(\mathbf{x}) = \mathbf{x}$. Similar to Equation 2, there is a feature-bias decomposition of the hidden output:

$$\mathbf{f}^{(l)}(\mathbf{x}) = \underbrace{\prod_{j=1}^{l}\mathbf{W}^{(j)}\mathbf{x}}_{\mathbf{g}^{(l)}(\mathbf{x})} + \underbrace{\sum_{j=1}^{l}\prod_{k=j+1}^{l}\mathbf{W}^{(k)}\mathbf{b}^{(j)}}_{\mathbf{h}^{(l)}=\sum_{j=1}^{l}\tilde{\mathbf{h}}^{(l)}(\mathbf{b}^{(j)})}, \tag{4}$$

where we use the same notation $\mathbf{g}^{(l)}(\mathbf{x})$ for the input-induced latent feature, $\mathbf{h}^{(l)}$ as the input-independent activation offset, and $\tilde{\mathbf{h}}^{(l)}(\mathbf{b}^{(j)})$ to measure the contribution of biases at the $j_{th}$ layer in $\mathbf{h}^{(l)}$.

Now, suppose there is a degradation in the parameters' magnitude: $\mathbf{W}^{(l)}, \mathbf{b}^{(l)} \leftarrow \lambda\mathbf{W}^{(l)}, \mu\mathbf{b}^{(l)}$. For simplicity, we assume the same degradation factor across layers. After the magnitude degradation, the hidden output changes in the following way: $\mathbf{g}^{(l)}(\mathbf{x}), \tilde{\mathbf{h}}^{(l)}(\mathbf{b}^{(j)}) \leftarrow \lambda^{l} \cdot \mathbf{g}^{(l)}(\mathbf{x}), \lambda^{l-j}\mu \cdot \tilde{\mathbf{h}}^{(l)}(\mathbf{b}^{(j)})$. The difference comes from the fact that $\mathbf{g}^{(l)}(\mathbf{x})$ is a successive multiplication of $l$ weight matrices and the input $\mathbf{x}$, while $\tilde{\mathbf{h}}^{(l)}(\mathbf{b}^{(j)})$ only consists of $l - j$ matrices and one bias vector. Therefore, the input-induced feature $\mathbf{g}^{(l)}(\mathbf{x})$ suffers more from this scale degradation, causing the vanishing feature phenomenon across layers. This is more clearly conveyed by the following more fine-grained feature-bias decomposition:

$$\mathbf{f}^{(l)}(\mathbf{x}) = \mathbf{g}^{(l)}(\mathbf{x}) + \mathbf{h}^{(l)} = \mathbf{g}^{(l)}(\mathbf{x}) + \sum_{j=1}^{s-1}\tilde{\mathbf{h}}^{(l)}(\mathbf{b}^{(j)}) + \sum_{j=s}^{l}\tilde{\mathbf{h}}^{(l)}(\mathbf{b}^{(j)}), \tag{5}$$

where $s$ separates the contributions of small $j$ and large $j$. For instance, $s$ can be chosen as $l - s'$, where $s'$ is a small constant satisfying $0 \leq s' \leq l_0 - 1$ when varying $l$ from $l_0$ to $L$. As $l$ increases, the contributions from $\mathbf{g}^{(l)}(\mathbf{x})$ and $\sum_{j=1}^{s-1}\tilde{\mathbf{h}}^{(l)}(\mathbf{b}^{(j)})$ decay exponentially. Consequently, the term $\sum_{j=s}^{l}\tilde{\mathbf{h}}^{(l)}(\mathbf{b}^{(j)})$, which depends solely on the parameters in later layers, becomes increasingly dominant.

This observation underpins our claim that parameters in later layers dominate $\mathbf{f}^{(l)}(\mathbf{x})$ in Section 4.2. There, by observing the increasing parameter scale in later layers (Figure 1, **Right**), we explain the contrasting behaviors of $\mathbf{g}_{\alpha}^{(l)}(\mathbf{x})$ (continuously shrinking) and $\mathbf{h}_{\alpha}^{(l)}$ (continuously growing) from the perspective of magnitude.

These analyses indicated that the vanishing feature can result from the magnitude reduction in the model subjected to a reference model. A similar style of analysis was shown by Wang et al. [34], where the linear interpolation between the start and the end of a training trajectory was studied, i.e., merging the random initialization and the final model.

**Model Merging.** Empirically, a magnitude degradation in the merged model's parameters compared to the edge models has often been observed, as shown in Figure 1 (**Mid**). This magnitude reduction contributes to the vanishing feature issue following a similar analysis as in the general case, where the edge models perform as the reference. Figure 1 (**Right**) illustrates that the issue was mitigated once all parameters were rescaled.

We also provide insights into the magnitude reduction in the merged model. Theoretically, the parameter magnitude of the merged models is initially upper bounded by the averaged magnitude of the edge models, as derived from the triangle inequality: $\left\|\frac{\mathbf{W}_0+\mathbf{W}_1}{2}\right\| \leq \frac{\|\mathbf{W}_0\|+\|\mathbf{W}_1\|}{2}$. More generally, additional bounds can be obtained by assuming specific parameter distributions. For instance, if all rows in $\mathbf{W}_0$ and $\mathbf{W}_1$ are sampled from a Gaussian distribution $\mathcal{N}(\mathbf{0}, \boldsymbol{\Sigma})$ (e.g., at initialization), straightforward calculations yield: $\mathbb{E}\left[\left\|\frac{\mathbf{W}_0+\mathbf{W}_1}{2}\right\|\right] \leq \frac{\mathbb{E}[\|\mathbf{W}_0\|]}{\sqrt{2}}$. For element-wise magnitude, as noted by Yadav et al. [50], this degradation becomes more pronounced when the edge models exhibit greater diversity. The reason is that greater diversity leads to more disagreement regarding the sign of individual parameter values across models, i.e., some parameters might have positive values in one model while being negative in the other model.

**Model Pruning.** In the context of pruning, the parameter scale of the pruned model significantly degrades compared to the original model, as most parameters are set to zero. Additionally, we observed that bias terms are often left unpruned in prior works, which could further exacerbate the vanishing feature issue.

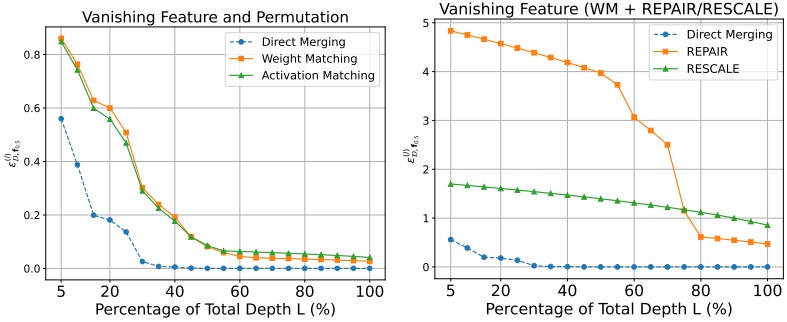

Figure 11: **Left**: Vanishing feature mitigation by **permutation-based merging**. Edge models are two VGG16s trained on CIFAR-10. Permutation derived by weight matching (orange line) and activation matching (green line) is applied before merging. **Right**: Vanishing feature mitigation by **REPAIR and RESCALE**. Edge models are two VGG16s trained on CIFAR-10. Permutation derived by weight matching is applied before merging. The upper bound sequences $(\varepsilon_{\mathcal{D},\mathbf{f}_{0.5}}^{(l)})_{l=1}^{L}$ in both plots are normalized by $\overline{\mathbb{E}_{\mathbf{x}}[\mathbf{g}^{(l)}(\mathbf{x})]}$.

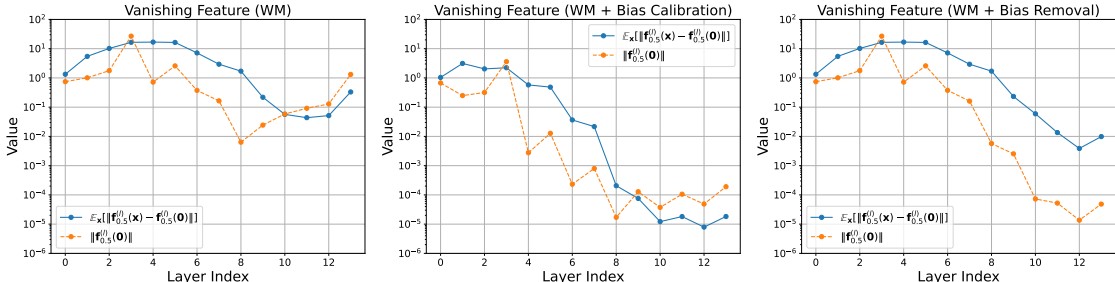

Figure 12: **Vanishing feature mitigation by bias calibration (Figure Mid) and bias removal (Figure Right).** Edge models are two VGG16s trained on CIFAR-10. Permutation derived by weight matching is applied before merging.

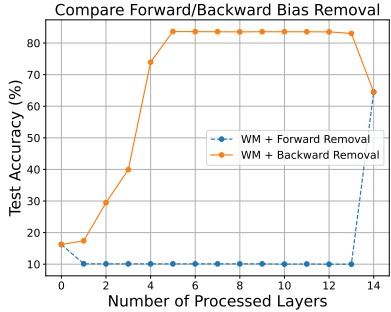

Figure 13: **Compare performance of forward and backward bias removal**. Edge models are two VGG16s trained on CIFAR-10 with different initialization. The merged midpoint model ($\alpha = 0.5$) is evaluated. Permutation derived by weight matching is applied before merging.

## A.6. Additional Results for Post-Merging Normalization

### A.6.1. Forward Bias Removal

Figure 13 illustrates the performance comparison between forward and backward bias removal, where biases in the first or last layers are sequentially set to zero. As depicted in the plot, forward bias removal degraded the model's performance, whereas backward bias removal achieved near-optimal performance by eliminating biases in only the last five layers. This aligns with the insight of performing the bias removal, as stated in Section 5.3, that it is designed to avoid the input-independent activation offset dominating the model's output due to the vanishing feature issue.

## A.7. Additional Results for Pruning

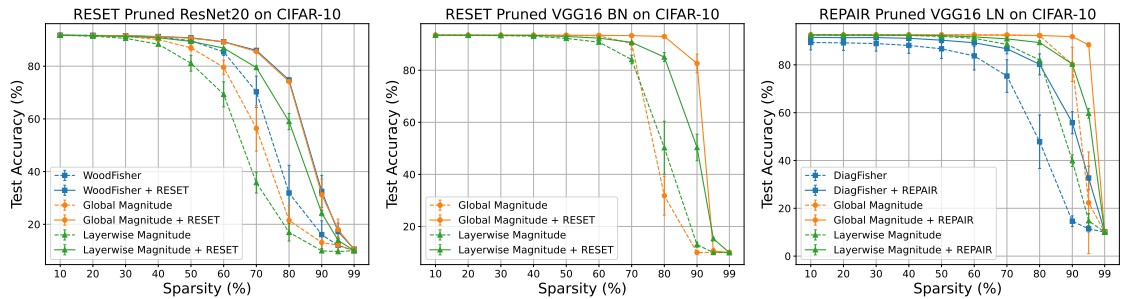

Figure 14: **Performance of REPAIR/RESET on pruned models on CIFAR-10**. The diagonal-Fisher pruning performs poorly on VGG16-BN and is hence ignored here.

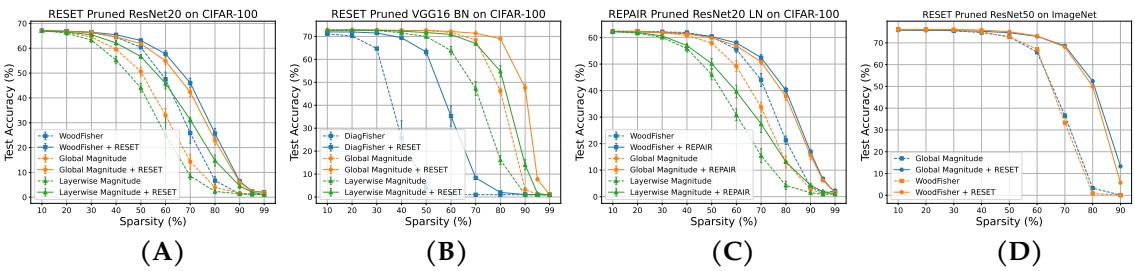

**(A)**        **(B)**        **(C)**        **(D)**

Figure 15: (**A & B & C**): Performance of REPAIR/RESET on pruned models on CIFAR-100. (**D**): Performance of RESET on pruned ResNet50 on ImageNet.

We present additional results of one-shot pruning ResNet20, ResNet20-LN, VGG16-BN, VGG16-LN on CIFAR-10 and CIFAR-100 in Figures 14 and 15 (**A & B & C**). Results of applying RESET to pruned ResNet50 on ImageNet are given in Figure 15 (**D**).

We also evaluated the performance of REPAIR when combined with a more recent advanced pruning algorithm, Wanda [33]. Specifically, we replicated their experiments on pruning the ConvNext-B network trained on the ImageNet dataset. Following their setup, only linear layers (excluding the first embedding layer and the final classification head) were pruned, as these layers account for most of the model's parameters. Layer-wise pruning was applied, and we utilized the pre-trained checkpoint provided in their repository[3]. After pruning, REPAIR was applied to correct activation statistics in all convolutional and linear layers using statistics from the original unpruned model. The results are reported in Figure 16 (**Left**). Consistent with our findings, REPAIR substantially enhances the pruned model's performance, especially at higher sparsity levels. For instance, REPAIR improves the test accuracy of Wanda and magnitude pruning by 36% and 55%, respectively, at 80% sparsity. Interestingly, we also observed that magnitude pruning, despite initially underperforming Wanda, often surpasses it after applying REPAIR. For example, at 76% sparsity, the test accuracy following Wanda pruning is 43.24% (without REPAIR) and 65.04% (with REPAIR), while the results for magnitude pruning are 35.45% (without REPAIR) and 69.88% (with REPAIR). Further investigation of this could yield valuable insights for improving existing pruning algorithms.

In Jordan et al. [12], it was reported that feeding a small portion of data ($\sim$ 5,000 examples) for statistics recording is already sufficient to guarantee the performance of REPAIR/RESET. In the context of pruning, we found that even one batch of data (64 in our pruning experiment) is often already enough. We provide one example of applying RESET with 64/128/258 samples to the pruned ResNet20 on CIFAR-10 in Figure 16. Note that this is both for the statistics recording in the unpruned model and the pruned model. For global magnitude pruning and layerwise magnitude pruning, one batch of data already suffices to produce a result almost the same as the complete RESET. This is particularly meaningful in resource-limited settings, especially considering its remarkable performance-boosting ability, which can yield comparable results to advanced pruning methods like WoodFisher, even when applied to simple magnitude pruning.

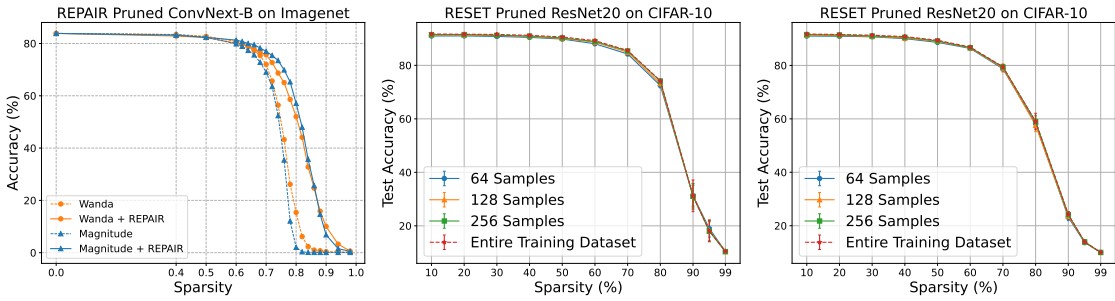

Figure 16: **Left:** REPAIR Performance on pruned ConvNext-B model on ImageNet. 1k samples were used to estimate the statistics in REPAIR. **Mid and Right:** Compare the number of samples utilized in RESET after the global magnitude pruning (**Mid**) and layerwise magnitude pruning (**Right**).

## A.8. Additional Results for Preserve-First Merging

### A.8.1. Vanishing Feature and Preserve-First Merging

This section evaluates the severity of the vanishing feature phenomenon in the merged VGG16/VGG16-BN/ResNet20 models after applying PFM. The CIFAR-10 dataset was used, and two edge models were independently trained. Visualizations are provided in Figure 17. The upper-bound sequence, normalized by the average statistics of the edge models, was measured. For $\text{PFM}_{l/L}$, the

---

[3]`https://github.com/locuslab/wanda`

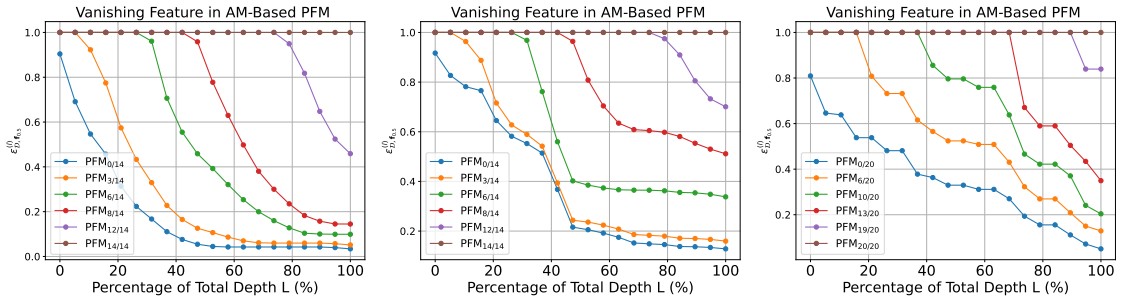

Figure 17: **Visualizing the vanishing feature phenomenon** in the merged VGG16 (**Left**), VGG16-BN (**Mid**), and ResNet20 (**Right**) models after applying AM-based PFM. The CIFAR-10 dataset was used, and two edge models were independently trained. The upper-bound sequence, normalized by the average statistics of the edge models, was measured.

intermediate activations of the preserved $l$ layers matched the averaged activations from the edge models, ensuring a tight upper bound of 1. The plots demonstrate that PFM effectively alleviates the vanishing feature issue, particularly when more early layers are preserved.

The corresponding accuracy and efficiency results are consistent with those reported in Tables 1, 7 and 6. The merged model utilized permutations obtained via the activation-matching algorithm. To avoid interference, REPAIR/RESET was not applied after merging.

### A.8.2. Compare Preserve-First Merging and Partial Zipping

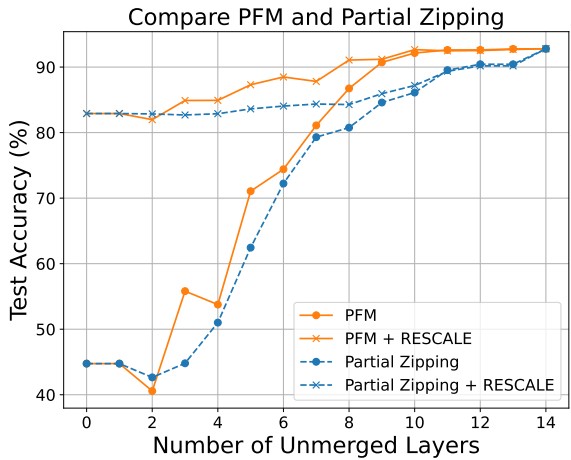

Figure 18: **Comparison of PFM and partial zipping from Stoica et al. [13] in same-dataset merging**. Edge models are VGG16s trained on CIFAR-10 with different initializations. Activation matching is used to derive permutations before merging. The x-axis denotes the number of unmerged layers: for PFM, the first $K$ layers are preserved, whereas for partial zipping, the last $K$ layers remain unmerged.

As discussed in Section 5.4, the proposed preserve-first merging strategy draws inspiration from the partial zipping operation introduced by ZipIt! [13]. ZipIt! is a specialized merging framework designed to handle models trained on different datasets and initializations. It addresses feature divergence across datasets through two key innovations: (1) "zip": enabling intra-model merging to account for channels with no counterpart in the other model, and (2) "partial zipping": preserving the last several layers unmerged, forming a multi-head structure advantageous for multi-task inference.

For activation-based-merging ($\alpha = 0.5$) at layer $l$, the ZipIt! method computes a "merge matrix" $\mathbf{M}_\alpha^{(l)}$ based on the similarity of features $\mathbf{f}_0^{(l)}$ and $\mathbf{f}_1^{(l)}$ from edge models $\mathbf{f}_1$ and $\mathbf{f}_2$. Each matched feature pair is averaged to produce merged features:

$$\mathbf{f}_\alpha^{(l)} = \mathbf{M}_\alpha^{(l)} \begin{bmatrix} \mathbf{f}_0^{(l)} \\ \mathbf{f}_1^{(l)} \end{bmatrix},$$

where $\mathbf{M}_\alpha^{(l)} \in \mathbb{R}^{d_l \times 2d_l}$ is constructed by greedily matching highly correlated features and $\mathbf{f}_0^{(l)}, \mathbf{f}_1^{(l)} \in \mathbb{R}^{d_l}$. To handle subsequent layers, ZipIt! applies an "unmerge matrix" $\mathbf{U}_\alpha^{(l)}$, which approximately inverts $\mathbf{M}_\alpha^{(l)}$, splitting the merged features back into components for $\mathbf{f}_0^{(l)}$ and $\mathbf{f}_1^{(l)}$. This ensures compatibility with the subsequent layers:

$$\mathbf{f}_0^{(l+1)} \approx \mathbf{L}_0^{(l+1)}(\mathbf{U}_0^{(l)}\mathbf{f}_\alpha^{(l)}), \quad \mathbf{f}_1^{(l+1)} \approx \mathbf{L}_1^{(l+1)}(\mathbf{U}_1^{(l)}\mathbf{f}_\alpha^{(l)}).$$

Here, $\mathbf{U}_\alpha^{(l)}$ is partitioned into $\mathbf{U}_0^{(l)}$ and $\mathbf{U}_1^{(l)}$, each applied to the respective split, and $\mathbf{L}_*^{(l+1)}$ represents the function of the $(l+1)_{\text{th}}$ layer of $\mathbf{f}_*$. For linear layers, the "zip" operation merges parameters by combining the merge and unmerge transformations, computed as

$$\mathbf{W}_\alpha^{(l)} = \mathbf{M}_0^{(l)}\mathbf{W}_0^{(l)}\mathbf{U}_0^{(l-1)} + \mathbf{M}_1^{(l)}\mathbf{W}_1^{(l)}\mathbf{U}_1^{(l-1)},$$

where $\mathbf{M}_0^{(l)}$ and $\mathbf{M}_1^{(l)}$ are column partitions of $\mathbf{M}_\alpha^{(l)}$, and the biases are similarly merged without unmerging.

While effective for cross-dataset scenarios, ZipIt! does not address the challenges of same-dataset merging. Its design inherently overlooks improvements in these settings, and the original work includes no experiments on same-dataset merging. By contrast, our PFM strategy is explicitly informed by the analysis of the vanishing feature phenomenon, as detailed in Sections 5.2, 5.3, and 5.4. PFM is specifically designed to retain early-layer features, which are crucial for maintaining performance, making it a natural fit for same-dataset merging.

Experimental results validate these insights. As shown in Figure 3, the intra-model merging approach of ZipIt! even underperforms the vanilla AM-based merging. Additionally, in Figure 18, we compare the performance of PFM with the partial zipping strategy from ZipIt!. Across both pre- and post-RESCALE scenarios, PFM consistently outperforms partial zipping, except in the case where only the first/last two layers remain unmerged.

### A.8.3. Evaluation of PFM with CCA Merging

We also evaluated the performance of PFM with the more advanced CCA merging algorithm (Horoi et al. [23]). Instead of aligning the activations from edge models in their original space as in the activation-matching algorithm, CCA merging projects the original activations to a common representation space and maximizes their correlations in the new space. Given the activation matrix at the $l$-th layer of two edge models as $\mathbf{f}_0^{(l)}(\mathbf{x})$ and $\mathbf{f}_1^{(l)}(\mathbf{x})$, CCA merging algorithm finds two projection matrices $\mathbf{P}_0^{(l)}$ and $\mathbf{P}_1^{(l)}$ to project the activations as $\tilde{\mathbf{f}}_0^{(l)}(\mathbf{x}) \leftarrow \mathbf{P}_0^{(l)}\mathbf{f}_0^{(l)}(\mathbf{x})$ and $\tilde{\mathbf{f}}_1^{(l)}(\mathbf{x}) \leftarrow \mathbf{P}_1^{(l)}\mathbf{f}_1^{(l)}(\mathbf{x})$, where the correlation between $\tilde{\mathbf{f}}_0^{(l)}(\mathbf{x})$ and $\tilde{\mathbf{f}}_1^{(l)}(\mathbf{x})$ is maximized. The parameters in the merged model are obtained by $\mathbf{W}_\alpha^{(l)} = (1-\alpha) \cdot \mathbf{W}_0^{(l)} + \alpha \cdot \mathbf{T_1}^{(l)}\mathbf{W}_1^{(l)}(\mathbf{T_1}^{(l-1)})^{-1}$, $\mathbf{b}_\alpha^{(l)} = (1-\alpha) \cdot \mathbf{b}_0^{(l)} + \alpha \cdot \mathbf{T_1}^{(l)}\mathbf{b}_1^{(l)}$, where $\mathbf{T_1}^{(k)} = (\mathbf{P}_0^{(k)})^{-1}\mathbf{P}_1^{(k)}$ for any $k \in [L]$. This method considerably improves the merging performance compared to the vanilla weight-matching and activation-matching algorithms. We report the performances of PFM + CCA merging in Tables 3 and 4. The tables show that PFM still brings notable improvement with this advanced merging backbone. Interestingly, we noticed that the merged model's performance surpassed that of the model ensemble on the VGG16-BN + CIFAR-10 setting by merging the last two layers.

Meanwhile, it is important to note that CCA merging adjusts the edge model's parameters based on the matching results from a different space. This adjustment alters the model's outputs in the original space, leading to a degradation in the individual model's performance. Consequently, post-merging

normalization is essential to ensure the effectiveness of the merging process, as highlighted in the original paper [23]. For example, as shown in Table 4, the merged model performs no better than random guessing when post-merging normalization is omitted, even with the application of PFM.

Table 3: Performance of the preserve-first merging (PFM) at the midpoint ($\alpha = 0.5$) on two **VGG16s trained on CIFAR-10** from different initializations, evaluated using the CCA merging strategy. Test accuracy with mean and standard deviation is reported. $\text{PFM}_{l/L}$ means that the first $l$ out of $L$ layers are preserved. "Bias Cal." refers to the bias calibration strategy proposed in Section 5.3. The highest accuracy in each row is bolded, and values surpassing the two end models' averages are shaded gray. Baseline values are boxed.

| Model | Direct | CCA Merging | | | | Params (M) | FLOPs (G) |
| | | W/o Normalization | REPAIR | RESCALE | Bias Cal. | | |
|---|---|---|---|---|---|---|---|
| Model 1 | $91.88_{\pm0.02}\%$ | | | | | 14.72 (1×) | 0.31 (1×) |
| Model 2 | $91.67_{\pm0.21}\%$ | | | | | 14.72 (1×) | 0.31 (1×) |
| Ensemble | $92.93_{\pm0.29}\%$ | | | | | 29.44 (2×) | 0.63 (2×) |
| $\text{PFM}_{0/14}$ | $\boxed{10.00}_{\pm0.00}\%$ | $\boxed{81.90}_{\pm3.86}\%$ | $\boxed{57.73}_{\pm4.32}\%$ | $\mathbf{83.14}_{\pm2.03}\%$ | $80.32_{\pm4.29}\%$ | 14.72 (1×) | 0.31 (1×) |
| $\text{PFM}_{3/14}$ | $10.03_{\pm0.05}\%$ | $83.74_{\pm3.67}\%$ | $65.42_{\pm1.68}\%$ | $\mathbf{85.67}_{\pm1.92}\%$ | $82.65_{\pm3.9}\%$ | 14.83 (1.008×) | 0.37 (1.19×) |
| $\text{PFM}_{6/14}$ | $10.92_{\pm0.38}\%$ | $88.77_{\pm2.13}\%$ | $73.52_{\pm3.16}\%$ | $\mathbf{89.91}_{\pm0.87}\%$ | $88.00_{\pm2.3}\%$ | 15.87 (1.08×) | 0.47 (1.49×) |
| $\text{PFM}_{8/14}$ | $26.35_{\pm6.54}\%$ | $90.68_{\pm1.59}\%$ | $79.81_{\pm3.69}\%$ | $\mathbf{91.77}_{\pm0.13}\%$ | $89.97_{\pm1.75}\%$ | 17.64 (1.2×) | 0.52 (1.67×) |
| $\text{PFM}_{12/14}$ | $92.88_{\pm0.19}\%$ | $92.91_{\pm0.28}\%$ | $92.33_{\pm0.23}\%$ | $\mathbf{92.92}_{\pm0.13}\%$ | $92.85_{\pm0.23}\%$ | 27.07 (1.84×) | 0.62 (1.97×) |

Table 4: Performance of the preserve-first merging (PFM) at the midpoint ($\alpha = 0.5$) on two **VGG16-BN models trained on CIFAR-10** from different initializations, evaluated using CCA merging strategy. Test accuracy with mean and standard deviation is reported. $\text{PFM}_{l/L}$ means that the first $l$ out of $L$ layers are preserved. "RESET" refers to the post-merging RESET technique. The highest accuracy in each row is bolded, and values surpassing the two base models' averages are shaded gray. Baseline values are boxed.

| Model | Direct | CCA Merging | | Params (M) | FLOPs (G) |
| | | W/o Normalization | RESET | | |
|---|---|---|---|---|---|
| Model 1 | $93.48_{\pm0.03}\%$ | | | 14.73 (1×) | 0.31 (1×) |
| Model 2 | $93.41_{\pm0.10}\%$ | | | 14.73 (1×) | 0.31 (1×) |
| Ensemble | $94.37_{\pm0.10}\%$ | | | 29.46 (2×) | 0.63 (2×) |
| $\text{PFM}_{0/14}$ | $\boxed{10.00}_{\pm0.00}\%$ | $\boxed{10.00}_{\pm0.00}\%$ | $\boxed{91.17}_{\pm0.14}\%$ | 14.73 (1×) | 0.31 (1×) |
| $\text{PFM}_{3/14}$ | $10.00_{\pm0.00}\%$ | $10.00_{\pm0.00}\%$ | $\mathbf{91.66}_{\pm0.16}\%$ | 14.84 (1.008×) | 0.37 (1.12×) |
| $\text{PFM}_{6/14}$ | $10.00_{\pm0.00}\%$ | $10.00_{\pm0.00}\%$ | $\mathbf{93.2}_{\pm0.02}\%$ | 15.88 (1.08×) | 0.47 (1.49×) |
| $\text{PFM}_{8/14}$ | $25.05_{\pm9.65}\%$ | $10.00_{\pm0.00}\%$ | $\mathbf{94.28}_{\pm0.14}\%$ | 17.65 (1.2×) | 0.52 (1.67×) |
| $\text{PFM}_{12/14}$ | $94.26_{\pm0.10}\%$ | $10.00_{\pm0.00}\%$ | $\mathbf{94.42}_{\pm0.08}\%$ | 27.09 (1.84×) | 0.62 (1.97×) |

#### A.8.4. Evaluation of PFM on Transformer-Based Models

In this Section, We evaluated PFM in the context of merging transformer-based models. Specifically, we merged two vision transformers (Dosovitskiy et al. [31]) (ViTs) on the CIFAR-10 dataset. These models were pre-trained on the ImageNet dataset with different initializations and subsequently fine-tuned on CIFAR-10. The merging backbone employed was OT-acts (Imfeld et al. [15]), a recently proposed state-of-the-art fusion algorithm based on optimal transport theory. We utilized the pre-trained checkpoints provided by the paper[4]. The model architecture is a simplified ViT with seven transformer blocks and fewer parameters.

As before, we applied PFM after modifying the original edge models using the OT-acts algorithm. For $\text{PFM}_{l/8}$, we preserved the first $l$ transformer blocks, while $\text{PFM}_{0/8}$ represents the original merged model. The test accuracy and efficiency profile are presented in Table 5. The results show that PFM can effectively enhance the performance of this advanced fusion algorithm when merging vision

---

[4] https://drive.google.com/file/d/1ez2VqveQSJyBJ0WlzdrsFetoIruZU4Ph/view

transformers. The performance could be further enhanced by more fine-grained handling of the activation flows within the preserved layers, as suggested by the OT-acts paper. This may have prevented PFM from fully demonstrating its potential in this experiment.

Table 5: Performance of the preserve-first merging (PFM) at the midpoint ($\alpha = 0.5$) on two **ViTs pre-trained on ImageNet and fine-tuned on CIFAR-10** from different initializations, evaluated using OT-acts. The test accuracy is reported. $\text{PFM}_{l/L}$ means the first $l$ out of $L$ transformer blocks are preserved, while $\text{PFM}_{0/8}$ represents the original merged model. The highest accuracy in each row is bolded. Baseline values are boxed.

| Model | Direct | OT-acts | Params (M) | FLOPs (M) |
|---|---|---|---|---|
| Model 1 | 93.34% | - | 12.44 (1×) | 13.59 (1×) |
| Model 2 | 93.31% | - | 12.44 (1×) | 13.59 (1×) |
| Ensemble | 93.37% | - | 24.89 (2×) | 27.19 (2×) |
| $\text{PFM}_{0/8}$ | 7.59 % | **61.48** % | 12.44 (1×) | 13.59 (1×) |
| $\text{PFM}_{3/8}$ | 28.46% | **73.79%** | 17.93 (1.44×) | 20.25 (1.49×) |
| $\text{PFM}_{4/8}$ | 42.41% | **85.02%** | 19.71 (1.58×) | 22.02 (1.62×) |
| $\text{PFM}_{5/8}$ | 71.34% | **88.46%** | 21.48 (1.73×) | 23.79 (1.75×) |
| $\text{PFM}_{6/8}$ | 89.41% | **90.84%** | 23.26 (1.87×) | 25.56 (1.88×) |

### A.8.5. Evaluation of PFM in Other Settings

We present more evaluation results in Tables 6, 7, 8, 9, 10, and 11.

Table 6: Performance of the preserve-first merging (PFM) at the midpoint ($\alpha = 0.5$) on two **VGG16s trained on CIFAR-10** from different initializations, evaluated using the ZipIt! merging strategy. Test accuracy with mean and standard deviation is reported. $\text{PFM}_{l/L}$ means that the first $l$ out of $L$ layers are preserved. "Bias Cal." refers to the bias calibration strategy proposed in Section 5.3. The highest accuracy in each row is bolded, and values surpassing the two end models' averages are shaded gray. Baseline values are boxed.

| Model | Direct | ZipIt! Merging | | | | Params (M) | FLOPs (G) |
|---|---|---|---|---|---|---|---|
| | | W/o Normalization | REPAIR | RESCALE | Bias Cal. | | |
| Model 1 | $91.88_{\pm 0.02}\%$ | | | | | 14.72 (1×) | 0.31 (1×) |
| Model 2 | $91.67_{\pm 0.21}\%$ | | | | | 14.72 (1×) | 0.31 (1×) |
| Ensemble | $92.93_{\pm 0.29}\%$ | | | | | 29.44 (2×) | 0.63 (2×) |
| $\text{PFM}_{0/14}$ | $10.00_{\pm 0.00}\%$ | $42.62_{\pm 3.04}\%$ | $67.93_{\pm 5.03}\%$ | $84.7_{\pm 0.23}\%$ | $83.24_{\pm 0.13}\%$ | 14.72 (1×) | 0.31 (1×) |
| $\text{PFM}_{3/14}$ | $10.03_{\pm 0.05}\%$ | $59.73_{\pm 3.28}\%$ | $69.89_{\pm 7.06}\%$ | $\mathbf{86.63}_{\pm 0.45}\%$ | $85.78_{\pm 0.62}\%$ | 14.83 (1.008×) | 0.37 (1.19×) |
| $\text{PFM}_{6/14}$ | $10.92_{\pm 0.38}\%$ | $85.30_{\pm 0.78}\%$ | $80.36_{\pm 4.46}\%$ | $\mathbf{90.84}_{\pm 0.25}\%$ | $90.20_{\pm 0.27}\%$ | 15.87 (1.08×) | 0.47 (1.49×) |
| $\text{PFM}_{8/14}$ | $26.35_{\pm 6.54}\%$ | $91.01_{\pm 0.27}\%$ | $89.04_{\pm 1.34}\%$ | $\mathbf{92.44}_{\pm 0.1}\%$ | $92.04_{\pm 0.24}\%$ | 17.64 (1.2×) | 0.52 (1.67×) |
| $\text{PFM}_{12/14}$ | $92.88_{\pm 0.19}\%$ | $\mathbf{92.89}_{\pm 0.25}\%$ | $92.88_{\pm 0.19}\%$ | $92.91_{\pm 0.24}\%$ | $92.74_{\pm 0.16}\%$ | 27.07 (1.84×) | 0.62 (1.97×) |

Table 7: Performance of the preserve-first merging (PFM) at the midpoint ($\alpha = 0.5$) on two **ResNet-20s trained on CIFAR-10** from different initializations, evaluated using Activation Matching and ZipIt! strategies. Test accuracy with mean and standard deviation is reported. PFM$_{l/L}$ means that the first $l$ out of $L$ layers are preserved. "RESET" refers to the post-merging RESET technique. The highest accuracy in each row is bolded, and values surpassing the two base models' averages are shaded gray. Baseline values are boxed.

| Model | Direct | Activation Matching | | ZipIt! Merging | | Params (M) | FLOPs (M) |
|---|---|---|---|---|---|---|---|
| | | W/o Normalization | RESET | W/o Normalization | RESET | | |
| Model 1 | $91.85_{\pm0.25}\%$ | | | | | 0.27 (1×) | 41.21 (1×) |
| Model 2 | $91.79_{\pm0.2}\%$ | | | | | 0.27 (1×) | 41.21 (1×) |
| Ensemble | $93.08_{\pm0.21}\%$ | | | | | 0.54 (2×) | 82.43 (2×) |
| PFM$_{0/20}$ | $\boxed{9.73}_{\pm0.47}\%$ | $\boxed{11.65}_{\pm2.36}\%$ | $\boxed{64.12}_{\pm0.33}\%$ | $13.66_{\pm3.65}\%$ | $62.20_{\pm3.66}\%$ | 0.27 (1×) | 41.21 (1×) |
| PFM$_{6/20}$ | $9.02_{\pm1.70}\%$ | $13.57_{\pm3.58}\%$ | $\mathbf{68.33}_{\pm0.52}\%$ | $15.03_{\pm5.42}\%$ | $63.01_{\pm3.76}\%$ | 0.28 (1.04×) | 53.65 (1.3×) |
| PFM$_{10/20}$ | $10.33_{\pm0.51}\%$ | $32.10_{\pm6.39}\%$ | $\mathbf{72.71}_{\pm0.73}\%$ | $37.15_{\pm5.14}\%$ | $70.98_{\pm3.50}\%$ | 0.31 (1.14×) | 62.14 (1.51×) |
| PFM$_{13/20}$ | $10.27_{\pm0.35}\%$ | $52.57_{\pm5.81}\%$ | $\mathbf{76.76}_{\pm0.99}\%$ | $43.63_{\pm6.96}\%$ | $72.29_{\pm2.66}\%$ | 0.34 (1.24×) | 69.26 (1.68×) |
| PFM$_{19/20}$ | $92.01_{\pm0.35}\%$ | $93.05_{\pm0.14}\%$ | $\mathbf{93.10}_{\pm0.16}\%$ | $87.60_{\pm1.00}\%$ | $89.75_{\pm0.75}\%$ | 0.54 (2×) | 82.43 (2×) |

Table 8: Performance of the preserve-first merging (PFM) at the midpoint ($\alpha = 0.5$) on two **VGG16-BN models trained on CIFAR-10** from different initializations, evaluated using Activation Matching and ZipIt! strategies. Test accuracy with mean and standard deviation is reported. PFM$_{l/L}$ means that the first $l$ out of $L$ layers are preserved. "RESET" refers to the post-merging RESET technique. The highest accuracy in each row is bolded, and values surpassing the two base models' averages are shaded gray. Baseline values are boxed.

| Model | Direct | Activation Matching | | ZipIt! Merging | | Params (M) | FLOPs (G) |
|---|---|---|---|---|---|---|---|
| | | W/o Normalization | RESET | W/o Normalization | RESET | | |
| Model 1 | $93.48_{\pm0.03}\%$ | | | | | 14.73 (1×) | 0.31 (1×) |
| Model 2 | $93.41_{\pm0.10}\%$ | | | | | 14.73 (1×) | 0.31 (1×) |
| Ensemble | $94.37_{\pm0.10}\%$ | | | | | 29.46 (2×) | 0.63 (2×) |
| PFM$_{0/14}$ | $\boxed{10.00}_{\pm0.00}\%$ | $\boxed{18.71}_{\pm1.74}\%$ | $\boxed{89.88}_{\pm0.12}\%$ | $\boxed{16.01}_{\pm1.58}\%$ | $\boxed{89.60}_{\pm0.18}\%$ | 14.73 (1×) | 0.31 (1×) |
| PFM$_{3/14}$ | $10.00_{\pm0.00}\%$ | $27.08_{\pm2.55}\%$ | $\mathbf{90.44}_{\pm0.22}\%$ | $23.95_{\pm2.93}\%$ | $90.22_{\pm0.29}\%$ | 14.84 (1.008×) | 0.37 (1.12×) |
| PFM$_{6/14}$ | $10.00_{\pm0.00}\%$ | $76.07_{\pm1.05}\%$ | $\mathbf{92.86}_{\pm0.06}\%$ | $73.96_{\pm1.35}\%$ | $92.58_{\pm0.29}\%$ | 15.88 (1.08×) | 0.47 (1.49×) |
| PFM$_{8/14}$ | $25.05_{\pm9.65}\%$ | $92.04_{\pm0.37}\%$ | $94.16_{\pm0.11}\%$ | $91.93_{\pm0.14}\%$ | $\mathbf{94.17}_{\pm0.13}\%$ | 17.65 (1.2×) | 0.52 (1.67×) |
| PFM$_{12/14}$ | $94.26_{\pm0.10}\%$ | $\mathbf{94.36}_{\pm0.03}\%$ | $\mathbf{94.36}_{\pm0.06}\%$ | $94.32_{\pm0.06}\%$ | $94.35_{\pm0.05}\%$ | 27.09 (1.84×) | 0.62 (1.97×) |

Table 9: Performance of the preserve-first merging (PFM) at the midpoint ($\alpha = 0.5$) on two **VGG16-BN models trained on CIFAR-100** from different initializations, evaluated using Activation Matching and ZipIt! strategies. Test accuracy with mean and standard deviation is reported. PFM$_{l/L}$ means that the first $l$ out of $L$ layers are preserved. "RESET" refers to the post-merging RESET technique. The highest accuracy in each row is bolded, and values surpassing the two base models' averages are shaded gray. Baseline values are boxed.

| Model | Direct | Activation Matching | | ZipIt! Merging | | Params (M) | FLOPs (G) |
|---|---|---|---|---|---|---|---|
| | | W/o Normalization | RESET | W/o Normalization | RESET | | |
| Model 1 | $72.76_{\pm0.17}\%$ | | | | | 14.73 (1×) | 0.31 (1×) |
| Model 2 | $72.79_{\pm0.15}\%$ | | | | | 14.73 (1×) | 0.31 (1×) |
| Ensemble | $75.21_{\pm0.30}\%$ | | | | | 29.46 (2×) | 0.63 (2×) |
| PFM$_{0/14}$ | $\boxed{1.00}_{\pm0.00}\%$ | $\boxed{1.25}_{\pm0.44}\%$ | $\boxed{59.82}_{\pm0.55}\%$ | $\boxed{1.07}_{\pm0.10}\%$ | $\boxed{59.44}_{\pm0.65}\%$ | 14.73 (1×) | 0.31 (1×) |
| PFM$_{3/14}$ | $1.00_{\pm0.01}\%$ | $1.42_{\pm0.32}\%$ | $\mathbf{60.83}_{\pm0.58}\%$ | $1.19_{\pm0.17}\%$ | $60.23_{\pm0.75}\%$ | 14.84 (1.008×) | 0.37 (1.19×) |
| PFM$_{7/14}$ | $1.02_{\pm0.03}\%$ | $4.64_{\pm1.20}\%$ | $\mathbf{65.43}_{\pm0.57}\%$ | $2.86_{\pm0.56}\%$ | $65.05_{\pm0.34}\%$ | 16.47 (1.12×) | 0.51 (1.61×) |
| PFM$_{10/14}$ | $32.45_{\pm2.04}\%$ | $69.77_{\pm0.60}\%$ | $\mathbf{73.56}_{\pm0.56}\%$ | $69.72_{\pm0.66}\%$ | $73.54_{\pm0.37}\%$ | 22.37 (1.52×) | 0.6 (1.91×) |
| PFM$_{12/14}$ | $69.31_{\pm1.11}\%$ | $74.72_{\pm0.36}\%$ | $\mathbf{74.92}_{\pm0.40}\%$ | $74.60_{\pm0.14}\%$ | $74.67_{\pm0.24}\%$ | 27.09 (1.84×) | 0.62 (1.97×) |

Table 10: Performance of the preserve-first merging (PFM) at the midpoint ($\alpha = 0.5$) on two **VGG16 models trained on CIFAR-100** from different initializations. Test accuracy with mean and standard deviation is reported. $\text{PFM}_{l/L}$ means that the first $l$ out of $L$ layers are preserved. "REPAIR" and "RESCALE" refer to post-merging REPAIR and RESCALE techniques. "Bias Cal." refers to the bias calibration strategy. The highest accuracy in each row is bolded, and values surpassing the two base models' averages are shaded gray. Baseline values are boxed.

| | | Activation Matching | | | | | |
| Model | Direct | W/o Normalization | REPAIR | RESCALE | Bias Cal. | Params (M) | FLOPs (G) |
|---|---|---|---|---|---|---|---|
| Model 1 | $63.34_{\pm0.07}\%$ | | | | | 14.72 (1×) | 0.31 (1×) |
| Model 2 | $63.39_{\pm0.11}\%$ | | | | | 14.72 (1×) | 0.31 (1×) |
| Ensemble | $66.30_{\pm0.34}\%$ | | | | | 29.44 (2×) | 0.63 (2×) |
| $\text{PFM}_{0/14}$ | $\boxed{1.02}_{\pm0.03}\%$ | $\boxed{1.99}_{\pm0.68}\%$ | $\boxed{30.63}_{\pm4.82}\%$ | $30.15_{\pm1.57}\%$ | $\mathbf{33.76}_{\pm1.53}\%$ | 14.72 (1×) | 0.31 (1×) |
| $\text{PFM}_{3/14}$ | $1.03_{\pm0.05}\%$ | $4.53_{\pm0.47}\%$ | $31.35_{\pm8.23}\%$ | $\mathbf{41.81}_{\pm1.93}\%$ | $40.99_{\pm1.12}\%$ | 14.83 (1.008×) | 0.37 (1.19×) |
| $\text{PFM}_{6/14}$ | $1.42_{\pm0.18}\%$ | $14.26_{\pm0.67}\%$ | $34.17_{\pm8.51}\%$ | $\mathbf{47.47}_{\pm1.37}\%$ | $46.39_{\pm0.58}\%$ | 15.87 (1.08×) | 0.47 (1.49×) |
| $\text{PFM}_{8/14}$ | $1.67_{\pm0.25}\%$ | $30.79_{\pm2.36}\%$ | $47.45_{\pm2.69}\%$ | $\mathbf{52.46}_{\pm0.33}\%$ | $51.20_{\pm0.32}\%$ | 17.64 (1.2×) | 0.52 (1.67×) |
| $\text{PFM}_{12/14}$ | $56.79_{\pm0.40}\%$ | $63.70_{\pm0.26}\%$ | $\mathbf{64.31}_{\pm0.35}\%$ | $63.66_{\pm0.29}\%$ | $63.40_{\pm0.06}\%$ | 27.07 (1.84×) | 0.62 (1.97×) |

Table 11: Performance of the preserve-first merging (PFM) at the midpoint ($\alpha = 0.5$) on two **VGG16 models trained on CIFAR-100** from different initializations, evaluated using ZipIt! strategies. Test accuracy with mean and standard deviation is reported. $\text{PFM}_{l/L}$ means that the first $l$ out of $L$ layers are preserved. "REPAIR" and "RESCALE" refer to post-merging REPAIR and RESCALE techniques. "Bias Cal." refers to the bias calibration strategy. The highest accuracy in each row is bolded, and values surpassing the two base models' averages are shaded gray. Baseline values are boxed.

| | | ZipIt! Merging | | | | | |
| Model | Direct | W/o Normalization | REPAIR | RESCALE | Bias Cal. | Params (M) | FLOPs (G) |
|---|---|---|---|---|---|---|---|
| Model 1 | $63.34_{\pm0.07}\%$ | | | | | 14.72 (1×) | 0.31 (1×) |
| Model 2 | $63.39_{\pm0.11}\%$ | | | | | 14.72 (1×) | 0.31 (1×) |
| Ensemble | $66.30_{\pm0.34}\%$ | | | | | 29.44 (2×) | 0.63 (2×) |
| $\text{PFM}_{0/14}$ | $\boxed{1.02}_{\pm0.03}\%$ | $\boxed{1.43}_{\pm0.09}\%$ | $\boxed{25.24}_{\pm2.03}\%$ | $28.44_{\pm4.75}\%$ | $\mathbf{31.91}_{\pm1.36}\%$ | 14.72 (1×) | 0.31 (1×) |
| $\text{PFM}_{3/14}$ | $1.03_{\pm0.05}\%$ | $3.23_{\pm0.16}\%$ | $29.76_{\pm2.45}\%$ | $\mathbf{41.33}_{\pm0.96}\%$ | $40.07_{\pm0.96}\%$ | 14.83 (1.008×) | 0.37 (1.19×) |
| $\text{PFM}_{6/14}$ | $1.42_{\pm0.18}\%$ | $12.69_{\pm0.24}\%$ | $34.05_{\pm2.61}\%$ | $\mathbf{46.93}_{\pm1.12}\%$ | $45.72_{\pm0.61}\%$ | 15.87 (1.08×) | 0.47 (1.49×) |
| $\text{PFM}_{8/14}$ | $1.67_{\pm0.25}\%$ | $29.48_{\pm0.55}\%$ | $42.38_{\pm0.28}\%$ | $\mathbf{52.59}_{\pm0.58}\%$ | $51.00_{\pm0.69}\%$ | 17.64 (1.2×) | 0.52 (1.67×) |
| $\text{PFM}_{12/14}$ | $56.79_{\pm0.40}\%$ | $63.51_{\pm0.21}\%$ | $\mathbf{64.03}_{\pm0.06}\%$ | $63.31_{\pm0.26}\%$ | $63.05_{\pm0.28}\%$ | 27.07 (1.84×) | 0.62 (1.97×) |

Table 12: Detailed breakdown of the architecture of the VGG16 network used in the PFM experiment, showing input/output shapes, parameter counts, memory usage, floating-point operations (FLOPs), and other metrics. Metrics include multiply-add operations (MAdd), floating-point operations executed (FLOPs), memory read (MemRead) and written (MemWrite) in bytes, percentage of total execution time (Duration), and total memory read and written (MemR+W). The statistics and tables are generated by the `torchstat` package [51].

| | Module Name | Input Shape | Output Shape | Params | Memory (MB) | MAdd | FLOPs | MemRead (B) | MemWrite (B) | Duration (%) | MemR+W (B) |
|---|---|---|---|---|---|---|---|---|---|---|---|
| 0 | features.0 | 3 32 32 | 64 32 32 | 1792.0 | 0.25 | 3,538,944.0 | 1,835,008.0 | 19456.0 | 262144.0 | 6.06% | 281600.0 |
| 1 | features.1 | 64 32 32 | 64 32 32 | 0.0 | 0.25 | 65,536.0 | 65,536.0 | 262144.0 | 262144.0 | 2.18% | 524288.0 |
| 2 | features.2 | 64 32 32 | 64 32 32 | 36928.0 | 0.25 | 75,497,472.0 | 37,814,272.0 | 409856.0 | 262144.0 | 5.65% | 672000.0 |
| 3 | features.3 | 64 32 32 | 64 32 32 | 0.0 | 0.25 | 65,536.0 | 65,536.0 | 262144.0 | 262144.0 | 1.93% | 524288.0 |
| 4 | features.4 | 64 32 32 | 64 16 16 | 0.0 | 0.06 | 49,152.0 | 65,536.0 | 262144.0 | 65536.0 | 2.74% | 327680.0 |
| 5 | features.5 | 64 16 16 | 128 16 16 | 73856.0 | 0.12 | 37,748,736.0 | 18,907,136.0 | 360960.0 | 131072.0 | 4.13% | 492032.0 |
| 6 | features.6 | 128 16 16 | 128 16 16 | 0.0 | 0.12 | 32,768.0 | 32,768.0 | 131072.0 | 131072.0 | 1.90% | 262144.0 |
| 7 | features.7 | 128 16 16 | 128 16 16 | 147584.0 | 0.12 | 75,497,472.0 | 37,781,504.0 | 721408.0 | 131072.0 | 4.87% | 852480.0 |
| 8 | features.8 | 128 16 16 | 128 16 16 | 0.0 | 0.12 | 32,768.0 | 32,768.0 | 131072.0 | 131072.0 | 1.89% | 262144.0 |
| 9 | features.9 | 128 16 16 | 128 8 8 | 0.0 | 0.03 | 24,576.0 | 32,768.0 | 131072.0 | 32768.0 | 2.39% | 163840.0 |
| 10 | features.10 | 128 8 8 | 256 8 8 | 295168.0 | 0.06 | 37,748,736.0 | 18,890,752.0 | 1213440.0 | 65536.0 | 3.61% | 1278976.0 |
| 11 | features.11 | 256 8 8 | 256 8 8 | 0.0 | 0.06 | 16,384.0 | 16,384.0 | 65536.0 | 65536.0 | 1.87% | 131072.0 |
| 12 | features.12 | 256 8 8 | 256 8 8 | 590080.0 | 0.06 | 75,497,472.0 | 37,765,120.0 | 2425856.0 | 65536.0 | 4.06% | 2491392.0 |
| 13 | features.13 | 256 8 8 | 256 8 8 | 0.0 | 0.06 | 16,384.0 | 16,384.0 | 65536.0 | 65536.0 | 1.86% | 131072.0 |
| 14 | features.14 | 256 8 8 | 256 8 8 | 590080.0 | 0.06 | 75,497,472.0 | 37,765,120.0 | 2425856.0 | 65536.0 | 4.32% | 2491392.0 |
| 15 | features.15 | 256 8 8 | 256 8 8 | 0.0 | 0.06 | 16,384.0 | 16,384.0 | 65536.0 | 65536.0 | 1.84% | 131072.0 |
| 16 | features.16 | 256 8 8 | 256 4 4 | 0.0 | 0.02 | 12,288.0 | 16,384.0 | 65536.0 | 16384.0 | 2.32% | 81920.0 |
| 17 | features.17 | 256 4 4 | 512 4 4 | 1180160.0 | 0.03 | 37,748,736.0 | 18,882,560.0 | 4737024.0 | 32768.0 | 3.91% | 4769792.0 |
| 18 | features.18 | 512 4 4 | 512 4 4 | 0.0 | 0.03 | 8,192.0 | 8,192.0 | 32768.0 | 32768.0 | 1.83% | 65536.0 |
| 19 | features.19 | 512 4 4 | 512 4 4 | 2359808.0 | 0.03 | 75,497,472.0 | 37,756,928.0 | 9472000.0 | 32768.0 | 4.54% | 9504768.0 |
| 20 | features.20 | 512 4 4 | 512 4 4 | 0.0 | 0.03 | 8,192.0 | 8,192.0 | 32768.0 | 32768.0 | 1.84% | 65536.0 |
| 21 | features.21 | 512 4 4 | 512 4 4 | 2359808.0 | 0.03 | 75,497,472.0 | 37,756,928.0 | 9472000.0 | 32768.0 | 4.65% | 9504768.0 |
| 22 | features.22 | 512 4 4 | 512 4 4 | 0.0 | 0.03 | 8,192.0 | 8,192.0 | 32768.0 | 32768.0 | 1.85% | 65536.0 |
| 23 | features.23 | 512 4 4 | 512 2 2 | 0.0 | 0.01 | 6,144.0 | 8,192.0 | 32768.0 | 8192.0 | 2.30% | 40960.0 |
| 24 | features.24 | 512 2 2 | 512 2 2 | 2359808.0 | 0.01 | 18,874,368.0 | 9,439,232.0 | 9447424.0 | 8192.0 | 4.84% | 9455616.0 |
| 25 | features.25 | 512 2 2 | 512 2 2 | 0.0 | 0.01 | 2,048.0 | 2,048.0 | 8192.0 | 8192.0 | 1.85% | 16384.0 |
| 26 | features.26 | 512 2 2 | 512 2 2 | 2359808.0 | 0.01 | 18,874,368.0 | 9,439,232.0 | 9447424.0 | 8192.0 | 4.88% | 9455616.0 |
| 27 | features.27 | 512 2 2 | 512 2 2 | 0.0 | 0.01 | 2,048.0 | 2,048.0 | 8192.0 | 8192.0 | 1.84% | 16384.0 |
| 28 | features.28 | 512 2 2 | 512 2 2 | 2359808.0 | 0.01 | 18,874,368.0 | 9,439,232.0 | 9447424.0 | 8192.0 | 4.92% | 9455616.0 |
| 29 | features.29 | 512 2 2 | 512 2 2 | 0.0 | 0.01 | 2,048.0 | 2,048.0 | 8192.0 | 8192.0 | 1.86% | 16384.0 |
| 30 | avgpool | 512 2 2 | 512 1 1 | 0.0 | 0.00 | 2,048.0 | 2,048.0 | 8192.0 | 2048.0 | 2.05% | 10240.0 |
| 31 | classifier.0 | 512 | 10 | 5130.0 | 0.00 | 10,230.0 | 5,120.0 | 22568.0 | 40.0 | 3.20% | 22608.0 |
| total | | | | 14719818.0 | 2.23 | 626,774,006.0 | 313,879,552.0 | 22568.0 | 40.0 | 100.00% | 63565136.0 |

Table 13: Detailed breakdown of the architecture of the VGG16-BN network used in the PFM experiment.

| | Module Name | Input Shape | Output Shape | Params | Memory (MB) | MAdd | FLOPs | MemRead (B) | MemWrite (B) | Duration (%) | MemR+W (B) |
|---|---|---|---|---|---|---|---|---|---|---|---|
| 0 | features.0 | 3 32 32 | 64 32 32 | 1792.0 | 0.25 | 3,538,944.0 | 1,835,008.0 | 19456.0 | 262144.0 | 7.01% | 281600.0 |
| 1 | features.1 | 64 32 32 | 64 32 32 | 128.0 | 0.25 | 262,144.0 | 131,072.0 | 262656.0 | 262144.0 | 2.18% | 524800.0 |
| 2 | features.2 | 64 32 32 | 64 32 32 | 0.0 | 0.25 | 65,536.0 | 65,536.0 | 262144.0 | 262144.0 | 1.41% | 524288.0 |
| 3 | features.3 | 64 32 32 | 64 32 32 | 36928.0 | 0.25 | 75,497,472.0 | 37,814,272.0 | 409856.0 | 262144.0 | 5.82% | 672000.0 |
| 4 | features.4 | 64 32 32 | 64 32 32 | 128.0 | 0.25 | 262,144.0 | 131,072.0 | 262656.0 | 262144.0 | 1.68% | 524800.0 |
| 5 | features.5 | 64 32 32 | 64 32 32 | 0.0 | 0.25 | 65,536.0 | 65,536.0 | 262144.0 | 262144.0 | 1.22% | 524288.0 |
| 6 | features.6 | 64 32 32 | 64 16 16 | 0.0 | 0.06 | 49,152.0 | 65,536.0 | 262144.0 | 65536.0 | 1.89% | 327680.0 |
| 7 | features.7 | 64 16 16 | 128 16 16 | 73856.0 | 0.12 | 37,748,736.0 | 18,907,136.0 | 360960.0 | 131072.0 | 3.30% | 492032.0 |
| 8 | features.8 | 128 16 16 | 128 16 16 | 256.0 | 0.12 | 131,072.0 | 65,536.0 | 132096.0 | 131072.0 | 1.31% | 263168.0 |
| 9 | features.9 | 128 16 16 | 128 16 16 | 0.0 | 0.12 | 32,768.0 | 32,768.0 | 131072.0 | 131072.0 | 1.15% | 262144.0 |
| 10 | features.10 | 128 16 16 | 128 16 16 | 147584.0 | 0.12 | 75,497,472.0 | 37,781,504.0 | 721408.0 | 131072.0 | 3.90% | 852480.0 |
| 11 | features.11 | 128 16 16 | 128 16 16 | 256.0 | 0.12 | 131,072.0 | 65,536.0 | 132096.0 | 131072.0 | 1.33% | 263168.0 |
| 12 | features.12 | 128 16 16 | 128 16 16 | 0.0 | 0.12 | 32,768.0 | 32,768.0 | 131072.0 | 131072.0 | 1.18% | 262144.0 |
| 13 | features.13 | 128 16 16 | 128 8 8 | 0.0 | 0.03 | 24,576.0 | 32,768.0 | 131072.0 | 32768.0 | 1.58% | 163840.0 |
| 14 | features.14 | 128 8 8 | 256 8 8 | 295168.0 | 0.06 | 37,748,736.0 | 18,890,752.0 | 1213440.0 | 65536.0 | 2.74% | 1278976.0 |
| 15 | features.15 | 256 8 8 | 256 8 8 | 512.0 | 0.06 | 65,536.0 | 32,768.0 | 67584.0 | 65536.0 | 1.27% | 133120.0 |
| 16 | features.16 | 256 8 8 | 256 8 8 | 0.0 | 0.06 | 16,384.0 | 16,384.0 | 65536.0 | 65536.0 | 1.09% | 131072.0 |
| 17 | features.17 | 256 8 8 | 256 8 8 | 590080.0 | 0.06 | 75,497,472.0 | 37,765,120.0 | 2425856.0 | 65536.0 | 3.33% | 2491392.0 |
| 18 | features.18 | 256 8 8 | 256 8 8 | 512.0 | 0.06 | 65,536.0 | 32,768.0 | 67584.0 | 65536.0 | 1.26% | 133120.0 |
| 19 | features.19 | 256 8 8 | 256 8 8 | 0.0 | 0.06 | 16,384.0 | 16,384.0 | 65536.0 | 65536.0 | 1.12% | 131072.0 |
| 20 | features.20 | 256 8 8 | 256 8 8 | 590080.0 | 0.06 | 75,497,472.0 | 37,765,120.0 | 2425856.0 | 65536.0 | 3.52% | 2491392.0 |
| 21 | features.21 | 256 8 8 | 256 8 8 | 512.0 | 0.06 | 65,536.0 | 32,768.0 | 67584.0 | 65536.0 | 1.28% | 133120.0 |
| 22 | features.22 | 256 8 8 | 256 8 8 | 0.0 | 0.06 | 16,384.0 | 16,384.0 | 65536.0 | 65536.0 | 1.07% | 131072.0 |
| 23 | features.23 | 256 8 8 | 256 4 4 | 0.0 | 0.02 | 12,288.0 | 16,384.0 | 65536.0 | 16384.0 | 1.48% | 81920.0 |
| 24 | features.24 | 256 4 4 | 512 4 4 | 1180160.0 | 0.03 | 37,748,736.0 | 18,882,560.0 | 4737024.0 | 32768.0 | 3.14% | 4769792.0 |
| 25 | features.25 | 512 4 4 | 512 4 4 | 1024.0 | 0.03 | 32,768.0 | 16,384.0 | 36864.0 | 32768.0 | 1.29% | 69632.0 |
| 26 | features.26 | 512 4 4 | 512 4 4 | 0.0 | 0.03 | 8,192.0 | 8,192.0 | 32768.0 | 32768.0 | 1.06% | 65536.0 |
| 27 | features.27 | 512 4 4 | 512 4 4 | 2359808.0 | 0.03 | 75,497,472.0 | 37,756,928.0 | 9472000.0 | 32768.0 | 4.33% | 9504768.0 |
| 28 | features.28 | 512 4 4 | 512 4 4 | 1024.0 | 0.03 | 32,768.0 | 16,384.0 | 36864.0 | 32768.0 | 1.30% | 69632.0 |
| 29 | features.29 | 512 4 4 | 512 4 4 | 0.0 | 0.03 | 8,192.0 | 8,192.0 | 32768.0 | 32768.0 | 1.13% | 65536.0 |
| 30 | features.30 | 512 4 4 | 512 4 4 | 2359808.0 | 0.03 | 75,497,472.0 | 37,756,928.0 | 9472000.0 | 32768.0 | 4.25% | 9504768.0 |
| 31 | features.31 | 512 4 4 | 512 4 4 | 1024.0 | 0.03 | 32,768.0 | 16,384.0 | 36864.0 | 32768.0 | 1.31% | 69632.0 |
| 32 | features.32 | 512 4 4 | 512 4 4 | 0.0 | 0.03 | 8,192.0 | 8,192.0 | 32768.0 | 32768.0 | 1.08% | 65536.0 |
| 33 | features.33 | 512 4 4 | 512 2 2 | 0.0 | 0.01 | 6,144.0 | 8,192.0 | 32768.0 | 8192.0 | 1.45% | 40960.0 |
| 34 | features.34 | 512 2 2 | 512 2 2 | 2359808.0 | 0.01 | 18,874,368.0 | 9,439,232.0 | 9447424.0 | 8192.0 | 4.46% | 9455616.0 |
| 35 | features.35 | 512 2 2 | 512 2 2 | 1024.0 | 0.01 | 8,192.0 | 4,096.0 | 12288.0 | 8192.0 | 1.42% | 20480.0 |
| 36 | features.36 | 512 2 2 | 512 2 2 | 0.0 | 0.01 | 2,048.0 | 2,048.0 | 8192.0 | 8192.0 | 1.08% | 16384.0 |
| 37 | features.37 | 512 2 2 | 512 2 2 | 2359808.0 | 0.01 | 18,874,368.0 | 9,439,232.0 | 9447424.0 | 8192.0 | 4.48% | 9455616.0 |
| 38 | features.38 | 512 2 2 | 512 2 2 | 1024.0 | 0.01 | 8,192.0 | 4,096.0 | 12288.0 | 8192.0 | 1.39% | 20480.0 |
| 39 | features.39 | 512 2 2 | 512 2 2 | 0.0 | 0.01 | 2,048.0 | 2,048.0 | 8192.0 | 8192.0 | 1.08% | 16384.0 |
| 40 | features.40 | 512 2 2 | 512 2 2 | 2359808.0 | 0.01 | 18,874,368.0 | 9,439,232.0 | 9447424.0 | 8192.0 | 4.51% | 9455616.0 |
| 41 | features.41 | 512 2 2 | 512 2 2 | 1024.0 | 0.01 | 8,192.0 | 4,096.0 | 12288.0 | 8192.0 | 1.40% | 20480.0 |
| 42 | features.42 | 512 2 2 | 512 2 2 | 0.0 | 0.01 | 2,048.0 | 2,048.0 | 8192.0 | 8192.0 | 1.08% | 16384.0 |
| 43 | avgpool | 512 2 2 | 512 1 1 | 0.0 | 0.00 | 2,048.0 | 2,048.0 | 8192.0 | 2048.0 | 3.56% | 10240.0 |
| 44 | classifier.0 | 512 | 10 | 5130.0 | 0.00 | 10,230.0 | 5,120.0 | 22568.0 | 40.0 | 2.10% | 22608.0 |
| total | | | | 14728266.0 | 3.28 | 627,879,926.0 | 314,432,512.0 | 22568.0 | 40.0 | 100.00% | 65810768.0 |

Table 14: Detailed breakdown of the architecture of the ResNet20 network used in the PFM experiment.

| | Module Name | Input Shape | Output Shape | Params | Memory (MB) | MAdd | FLOPs | MemRead (B) | MemWrite (B) | Duration (%) | MemR+W (B) |
|---|---|---|---|---|---|---|---|---|---|---|---|
| 0 | conv1 | 3 32 32 | 16 32 32 | 432.0 | 0.06 | 868,352.0 | 442368.0 | 14016.0 | 65536.0 | 6.61% | 79552.0 |
| 1 | norm1 | 16 32 32 | 16 32 32 | 32.0 | 0.06 | 65,536.0 | 32768.0 | 65664.0 | 65536.0 | 2.91% | 131200.0 |
| 2 | avg_pool | 64 8 8 | 64 1 1 | 0.0 | 0.00 | 0.0 | 0.0 | 0.0 | 0.0 | 0.81% | 0.0 |
| 3 | segments.0.0.conv1 | 16 32 32 | 16 32 32 | 2304.0 | 0.06 | 4,702,208.0 | 2359296.0 | 74752.0 | 65536.0 | 3.31% | 140288.0 |
| 4 | segments.0.0.norm1 | 16 32 32 | 16 32 32 | 32.0 | 0.06 | 65,536.0 | 32768.0 | 65664.0 | 65536.0 | 4.51% | 131200.0 |
| 5 | segments.0.0.conv2 | 16 32 32 | 16 32 32 | 2304.0 | 0.06 | 4,702,208.0 | 2359296.0 | 74752.0 | 65536.0 | 7.64% | 140288.0 |
| 6 | segments.0.0.norm2 | 16 32 32 | 16 32 32 | 32.0 | 0.06 | 65,536.0 | 32768.0 | 65664.0 | 65536.0 | 5.07% | 131200.0 |
| 7 | segments.0.0.shortcut | 16 32 32 | 16 32 32 | 0.0 | 0.06 | 0.0 | 0.0 | 0.0 | 0.0 | 4.06% | 0.0 |
| 8 | segments.0.1.conv1 | 16 32 32 | 16 32 32 | 2304.0 | 0.06 | 4,702,208.0 | 2359296.0 | 74752.0 | 65536.0 | 8.05% | 140288.0 |
| 9 | segments.0.1.norm1 | 16 32 32 | 16 32 32 | 32.0 | 0.06 | 65,536.0 | 32768.0 | 65664.0 | 65536.0 | 5.56% | 131200.0 |
| 10 | segments.0.1.conv2 | 16 32 32 | 16 32 32 | 2304.0 | 0.06 | 4,702,208.0 | 2359296.0 | 74752.0 | 65536.0 | 6.33% | 140288.0 |
| 11 | segments.0.1.norm2 | 16 32 32 | 16 32 32 | 32.0 | 0.06 | 65,536.0 | 32768.0 | 65664.0 | 65536.0 | 1.07% | 131200.0 |
| 12 | segments.0.1.shortcut | 16 32 32 | 16 32 32 | 0.0 | 0.06 | 0.0 | 0.0 | 0.0 | 0.0 | 0.25% | 0.0 |
| 13 | segments.0.2.conv1 | 16 32 32 | 16 32 32 | 2304.0 | 0.06 | 4,702,208.0 | 2359296.0 | 74752.0 | 65536.0 | 1.82% | 140288.0 |
| 14 | segments.0.2.norm1 | 16 32 32 | 16 32 32 | 32.0 | 0.06 | 65,536.0 | 32768.0 | 65664.0 | 65536.0 | 1.08% | 131200.0 |
| 15 | segments.0.2.conv2 | 16 32 32 | 16 32 32 | 2304.0 | 0.06 | 4,702,208.0 | 2359296.0 | 74752.0 | 65536.0 | 2.02% | 140288.0 |
| 16 | segments.0.2.norm2 | 16 32 32 | 16 32 32 | 32.0 | 0.06 | 65,536.0 | 32768.0 | 65664.0 | 65536.0 | 1.13% | 131200.0 |
| 17 | segments.0.2.shortcut | 16 32 32 | 16 32 32 | 0.0 | 0.06 | 0.0 | 0.0 | 0.0 | 0.0 | 0.28% | 0.0 |
| 18 | segments.1.0.conv1 | 16 32 32 | 32 16 16 | 4608.0 | 0.03 | 2,351,104.0 | 1179648.0 | 83968.0 | 32768.0 | 1.54% | 116736.0 |
| 19 | segments.1.0.norm1 | 32 16 16 | 32 16 16 | 64.0 | 0.03 | 32,768.0 | 16384.0 | 33024.0 | 32768.0 | 1.09% | 65792.0 |
| 20 | segments.1.0.conv2 | 32 16 16 | 32 16 16 | 9216.0 | 0.03 | 4,710,400.0 | 2359296.0 | 69632.0 | 32768.0 | 1.56% | 102400.0 |
| 21 | segments.1.0.norm2 | 32 16 16 | 32 16 16 | 64.0 | 0.03 | 32,768.0 | 16384.0 | 33024.0 | 32768.0 | 1.00% | 65792.0 |
| 22 | segments.1.0.shortcut.0 | 16 32 32 | 32 16 16 | 512.0 | 0.03 | 253,952.0 | 131072.0 | 67584.0 | 32768.0 | 1.41% | 100352.0 |
| 23 | segments.1.0.shortcut.1 | 32 16 16 | 32 16 16 | 64.0 | 0.03 | 32,768.0 | 16384.0 | 33024.0 | 32768.0 | 1.02% | 65792.0 |
| 24 | segments.1.1.conv1 | 32 16 16 | 32 16 16 | 9216.0 | 0.03 | 4,710,400.0 | 2359296.0 | 69632.0 | 32768.0 | 1.53% | 102400.0 |
| 25 | segments.1.1.norm1 | 32 16 16 | 32 16 16 | 64.0 | 0.03 | 32,768.0 | 16384.0 | 33024.0 | 32768.0 | 1.02% | 65792.0 |
| 26 | segments.1.1.conv2 | 32 16 16 | 32 16 16 | 9216.0 | 0.03 | 4,710,400.0 | 2359296.0 | 69632.0 | 32768.0 | 1.55% | 102400.0 |
| 27 | segments.1.1.norm2 | 32 16 16 | 32 16 16 | 64.0 | 0.03 | 32,768.0 | 16384.0 | 33024.0 | 32768.0 | 1.02% | 65792.0 |
| 28 | segments.1.1.shortcut | 32 16 16 | 32 16 16 | 0.0 | 0.03 | 0.0 | 0.0 | 0.0 | 0.0 | 0.23% | 0.0 |
| 29 | segments.1.2.conv1 | 32 16 16 | 32 16 16 | 9216.0 | 0.03 | 4,710,400.0 | 2359296.0 | 69632.0 | 32768.0 | 1.49% | 102400.0 |
| 30 | segments.1.2.norm1 | 32 16 16 | 32 16 16 | 64.0 | 0.03 | 32,768.0 | 16384.0 | 33024.0 | 32768.0 | 1.00% | 65792.0 |
| 31 | segments.1.2.conv2 | 32 16 16 | 32 16 16 | 9216.0 | 0.03 | 4,710,400.0 | 2359296.0 | 69632.0 | 32768.0 | 1.51% | 102400.0 |
| 32 | segments.1.2.norm2 | 32 16 16 | 32 16 16 | 64.0 | 0.03 | 32,768.0 | 16384.0 | 33024.0 | 32768.0 | 1.01% | 65792.0 |
| 33 | segments.1.2.shortcut | 32 16 16 | 32 16 16 | 0.0 | 0.03 | 0.0 | 0.0 | 0.0 | 0.0 | 0.23% | 0.0 |
| 34 | segments.2.0.conv1 | 32 16 16 | 64 8 8 | 18432.0 | 0.02 | 2,355,200.0 | 1179648.0 | 106496.0 | 16384.0 | 1.44% | 122880.0 |
| 35 | segments.2.0.norm1 | 64 8 8 | 64 8 8 | 128.0 | 0.02 | 16,384.0 | 8192.0 | 16896.0 | 16384.0 | 1.01% | 33280.0 |
| 36 | segments.2.0.conv2 | 64 8 8 | 64 8 8 | 36864.0 | 0.02 | 4,714,496.0 | 2359296.0 | 163840.0 | 16384.0 | 1.45% | 180224.0 |
| 37 | segments.2.0.norm2 | 64 8 8 | 64 8 8 | 128.0 | 0.02 | 16,384.0 | 8192.0 | 16896.0 | 16384.0 | 1.00% | 33280.0 |
| 38 | segments.2.0.shortcut.0 | 32 16 16 | 64 8 8 | 2048.0 | 0.02 | 258,048.0 | 131072.0 | 40960.0 | 16384.0 | 1.60% | 57344.0 |
| 39 | segments.2.0.shortcut.1 | 64 8 8 | 64 8 8 | 128.0 | 0.02 | 16,384.0 | 8192.0 | 16896.0 | 16384.0 | 1.02% | 33280.0 |
| 40 | segments.2.1.conv1 | 64 8 8 | 64 8 8 | 36864.0 | 0.02 | 4,714,496.0 | 2359296.0 | 163840.0 | 16384.0 | 1.47% | 180224.0 |
| 41 | segments.2.1.norm1 | 64 8 8 | 64 8 8 | 128.0 | 0.02 | 16,384.0 | 8192.0 | 16896.0 | 16384.0 | 1.01% | 33280.0 |
| 42 | segments.2.1.conv2 | 64 8 8 | 64 8 8 | 36864.0 | 0.02 | 4,714,496.0 | 2359296.0 | 163840.0 | 16384.0 | 1.45% | 180224.0 |
| 43 | segments.2.1.norm2 | 64 8 8 | 64 8 8 | 128.0 | 0.02 | 16,384.0 | 8192.0 | 16896.0 | 16384.0 | 0.99% | 33280.0 |
| 44 | segments.2.1.shortcut | 64 8 8 | 64 8 8 | 0.0 | 0.02 | 0.0 | 0.0 | 0.0 | 0.0 | 0.24% | 0.0 |
| 45 | segments.2.2.conv1 | 64 8 8 | 64 8 8 | 36864.0 | 0.02 | 4,714,496.0 | 2359296.0 | 163840.0 | 16384.0 | 1.44% | 180224.0 |
| 46 | segments.2.2.norm1 | 64 8 8 | 64 8 8 | 128.0 | 0.02 | 16,384.0 | 8192.0 | 16896.0 | 16384.0 | 1.00% | 33280.0 |
| 47 | segments.2.2.conv2 | 64 8 8 | 64 8 8 | 36864.0 | 0.02 | 4,714,496.0 | 2359296.0 | 163840.0 | 16384.0 | 1.45% | 180224.0 |
| 48 | segments.2.2.norm2 | 64 8 8 | 64 8 8 | 128.0 | 0.02 | 16,384.0 | 8192.0 | 16896.0 | 16384.0 | 1.02% | 33280.0 |
| 49 | segments.2.2.shortcut | 64 8 8 | 64 8 8 | 0.0 | 0.02 | 0.0 | 0.0 | 0.0 | 0.0 | 0.22% | 0.0 |
| 50 | fc | 64 | 10 | 650.0 | 0.00 | 1,270.0 | 640.0 | 2856.0 | 40.0 | 1.48% | 2896.0 |
| total | | | | 272474.0 | 1.81 | 82,228,470.0 | 41214592.0 | 2856.0 | 40.0 | 100.00% | 4346512.0 |

Table 15: Detailed breakdown of the architecture of the ResNet50 network in the PFM experiment (part 1).

| | Module Name | Input Shape | Output Shape | Params | Memory (MB) | MAdd | FLOPs | MemRead (B) | MemWrite (B) | Duration (%) | MemR+W (B) |
|---|---|---|---|---|---|---|---|---|---|---|---|
| 0 | conv1 | 3 32 32 | 64 16 16 | 9408.0 | 0.06 | 4,800,512.0 | 2408448.0 | 49920.0 | 65536.0 | 2.59% | 115456.0 |
| 1 | bn1 | 64 16 16 | 64 16 16 | 128.0 | 0.06 | 65,536.0 | 32768.0 | 66048.0 | 65536.0 | 0.92% | 131584.0 |
| 2 | relu | 64 16 16 | 64 16 16 | 0.0 | 0.06 | 16,384.0 | 16384.0 | 65536.0 | 65536.0 | 0.58% | 131072.0 |
| 3 | maxpool | 64 16 16 | 64 8 8 | 0.0 | 0.02 | 32,768.0 | 16384.0 | 65536.0 | 16384.0 | 0.78% | 81920.0 |
| 4 | layer1.0.conv1 | 64 8 8 | 64 8 8 | 4096.0 | 0.02 | 520,192.0 | 262144.0 | 32768.0 | 16384.0 | 0.96% | 49152.0 |
| 5 | layer1.0.bn1 | 64 8 8 | 64 8 8 | 128.0 | 0.02 | 16,384.0 | 8192.0 | 16896.0 | 16384.0 | 0.70% | 33280.0 |
| 6 | layer1.0.conv2 | 64 8 8 | 64 8 8 | 36864.0 | 0.02 | 4,714,496.0 | 2359296.0 | 163840.0 | 16384.0 | 1.01% | 180224.0 |
| 7 | layer1.0.bn2 | 64 8 8 | 64 8 8 | 128.0 | 0.02 | 16,384.0 | 8192.0 | 16896.0 | 16384.0 | 0.68% | 33280.0 |
| 8 | layer1.0.conv3 | 64 8 8 | 256 8 8 | 16384.0 | 0.06 | 2,080,768.0 | 1048576.0 | 81920.0 | 65536.0 | 0.86% | 147456.0 |
| 9 | layer1.0.bn3 | 256 8 8 | 256 8 8 | 512.0 | 0.06 | 65,536.0 | 32768.0 | 67584.0 | 65536.0 | 0.68% | 133120.0 |
| 10 | layer1.0.relu | 256 8 8 | 256 8 8 | 0.0 | 0.06 | 16,384.0 | 16384.0 | 65536.0 | 65536.0 | 0.55% | 131072.0 |
| 11 | layer1.0.downsample.0 | 64 8 8 | 256 8 8 | 16384.0 | 0.06 | 2,080,768.0 | 1048576.0 | 81920.0 | 65536.0 | 0.87% | 147456.0 |
| 12 | layer1.0.downsample.1 | 256 8 8 | 256 8 8 | 512.0 | 0.06 | 65,536.0 | 32768.0 | 67584.0 | 65536.0 | 0.69% | 133120.0 |
| 13 | layer1.1.conv1 | 256 8 8 | 64 8 8 | 16384.0 | 0.02 | 2,093,056.0 | 1048576.0 | 131072.0 | 16384.0 | 0.94% | 147456.0 |
| 14 | layer1.1.bn1 | 64 8 8 | 64 8 8 | 128.0 | 0.02 | 16,384.0 | 8192.0 | 16896.0 | 16384.0 | 0.74% | 33280.0 |
| 15 | layer1.1.conv2 | 64 8 8 | 64 8 8 | 36864.0 | 0.02 | 4,714,496.0 | 2359296.0 | 163840.0 | 16384.0 | 0.97% | 180224.0 |
| 16 | layer1.1.bn2 | 64 8 8 | 64 8 8 | 128.0 | 0.02 | 16,384.0 | 8192.0 | 16896.0 | 16384.0 | 0.68% | 33280.0 |
| 17 | layer1.1.conv3 | 64 8 8 | 256 8 8 | 16384.0 | 0.06 | 2,080,768.0 | 1048576.0 | 81920.0 | 65536.0 | 0.84% | 147456.0 |
| 18 | layer1.1.bn3 | 256 8 8 | 256 8 8 | 512.0 | 0.06 | 65,536.0 | 32768.0 | 67584.0 | 65536.0 | 0.66% | 133120.0 |
| 19 | layer1.1.relu | 256 8 8 | 256 8 8 | 0.0 | 0.06 | 16,384.0 | 16384.0 | 65536.0 | 65536.0 | 0.45% | 131072.0 |
| 20 | layer1.2.conv1 | 256 8 8 | 64 8 8 | 16384.0 | 0.02 | 2,093,056.0 | 1048576.0 | 131072.0 | 16384.0 | 0.82% | 147456.0 |
| 21 | layer1.2.bn1 | 64 8 8 | 64 8 8 | 128.0 | 0.02 | 16,384.0 | 8192.0 | 16896.0 | 16384.0 | 0.66% | 33280.0 |
| 22 | layer1.2.conv2 | 64 8 8 | 64 8 8 | 36864.0 | 0.02 | 4,714,496.0 | 2359296.0 | 163840.0 | 16384.0 | 0.92% | 180224.0 |
| 23 | layer1.2.bn2 | 64 8 8 | 64 8 8 | 128.0 | 0.02 | 16,384.0 | 8192.0 | 16896.0 | 16384.0 | 0.65% | 33280.0 |
| 24 | layer1.2.conv3 | 64 8 8 | 256 8 8 | 16384.0 | 0.06 | 2,080,768.0 | 1048576.0 | 81920.0 | 65536.0 | 0.81% | 147456.0 |
| 25 | layer1.2.bn3 | 256 8 8 | 256 8 8 | 512.0 | 0.06 | 65,536.0 | 32768.0 | 67584.0 | 65536.0 | 0.67% | 133120.0 |
| 26 | layer1.2.relu | 256 8 8 | 256 8 8 | 0.0 | 0.06 | 16,384.0 | 16384.0 | 65536.0 | 65536.0 | 0.46% | 131072.0 |
| 27 | layer2.0.conv1 | 256 8 8 | 128 8 8 | 32768.0 | 0.03 | 4,186,112.0 | 2097152.0 | 196608.0 | 32768.0 | 0.83% | 229376.0 |
| 28 | layer2.0.bn1 | 128 8 8 | 128 8 8 | 256.0 | 0.03 | 32,768.0 | 16384.0 | 33792.0 | 32768.0 | 0.69% | 66560.0 |
| 29 | layer2.0.conv2 | 128 8 8 | 128 4 4 | 147456.0 | 0.01 | 4,716,544.0 | 2359296.0 | 622592.0 | 8192.0 | 0.92% | 630784.0 |
| 30 | layer2.0.bn2 | 128 4 4 | 128 4 4 | 256.0 | 0.01 | 8,192.0 | 4096.0 | 9216.0 | 8192.0 | 0.66% | 17408.0 |
| 31 | layer2.0.conv3 | 128 4 4 | 512 4 4 | 65536.0 | 0.03 | 2,088,960.0 | 1048576.0 | 270336.0 | 32768.0 | 0.82% | 303104.0 |
| 32 | layer2.0.bn3 | 512 4 4 | 512 4 4 | 1024.0 | 0.03 | 32,768.0 | 16384.0 | 36864.0 | 32768.0 | 0.65% | 69632.0 |
| 33 | layer2.0.relu | 512 4 4 | 512 4 4 | 0.0 | 0.03 | 8,192.0 | 8192.0 | 32768.0 | 32768.0 | 0.45% | 65536.0 |
| 34 | layer2.0.downsample.0 | 256 8 8 | 512 4 4 | 131072.0 | 0.03 | 4,186,112.0 | 2097152.0 | 589824.0 | 32768.0 | 0.88% | 622592.0 |
| 35 | layer2.0.downsample.1 | 512 4 4 | 512 4 4 | 1024.0 | 0.03 | 32,768.0 | 16384.0 | 36864.0 | 32768.0 | 0.70% | 69632.0 |
| 36 | layer2.1.conv1 | 512 4 4 | 128 4 4 | 65536.0 | 0.01 | 2,095,104.0 | 1048576.0 | 294912.0 | 8192.0 | 0.82% | 303104.0 |
| 37 | layer2.1.bn1 | 128 4 4 | 128 4 4 | 256.0 | 0.01 | 8,192.0 | 4096.0 | 9216.0 | 8192.0 | 0.63% | 17408.0 |
| 38 | layer2.1.conv2 | 128 4 4 | 128 4 4 | 147456.0 | 0.01 | 4,716,544.0 | 2359296.0 | 598016.0 | 8192.0 | 0.89% | 606208.0 |
| 39 | layer2.1.bn2 | 128 4 4 | 128 4 4 | 256.0 | 0.01 | 8,192.0 | 4096.0 | 9216.0 | 8192.0 | 0.66% | 17408.0 |
| 40 | layer2.1.conv3 | 128 4 4 | 512 4 4 | 65536.0 | 0.03 | 2,088,960.0 | 1048576.0 | 270336.0 | 32768.0 | 0.81% | 303104.0 |
| 41 | layer2.1.bn3 | 512 4 4 | 512 4 4 | 1024.0 | 0.03 | 32,768.0 | 16384.0 | 36864.0 | 32768.0 | 0.67% | 69632.0 |
| 42 | layer2.1.relu | 512 4 4 | 512 4 4 | 0.0 | 0.03 | 8,192.0 | 8192.0 | 32768.0 | 32768.0 | 0.45% | 65536.0 |
| 43 | layer2.2.conv1 | 512 4 4 | 128 4 4 | 65536.0 | 0.01 | 2,095,104.0 | 1048576.0 | 294912.0 | 8192.0 | 0.81% | 303104.0 |
| 44 | layer2.2.bn1 | 128 4 4 | 128 4 4 | 256.0 | 0.01 | 8,192.0 | 4096.0 | 9216.0 | 8192.0 | 0.65% | 17408.0 |
| 45 | layer2.2.conv2 | 128 4 4 | 128 4 4 | 147456.0 | 0.01 | 4,716,544.0 | 2359296.0 | 598016.0 | 8192.0 | 0.92% | 606208.0 |
| 46 | layer2.2.bn2 | 128 4 4 | 128 4 4 | 256.0 | 0.01 | 8,192.0 | 4096.0 | 9216.0 | 8192.0 | 0.64% | 17408.0 |
| 47 | layer2.2.conv3 | 128 4 4 | 512 4 4 | 65536.0 | 0.03 | 2,088,960.0 | 1048576.0 | 270336.0 | 32768.0 | 0.81% | 303104.0 |
| 48 | layer2.2.bn3 | 512 4 4 | 512 4 4 | 1024.0 | 0.03 | 32,768.0 | 16384.0 | 36864.0 | 32768.0 | 0.68% | 69632.0 |
| 49 | layer2.2.relu | 512 4 4 | 512 4 4 | 0.0 | 0.03 | 8,192.0 | 8192.0 | 32768.0 | 32768.0 | 0.45% | 65536.0 |
| 50 | layer2.3.conv1 | 512 4 4 | 128 4 4 | 65536.0 | 0.01 | 2,095,104.0 | 1048576.0 | 294912.0 | 8192.0 | 0.82% | 303104.0 |
| 51 | layer2.3.bn1 | 128 4 4 | 128 4 4 | 256.0 | 0.01 | 8,192.0 | 4096.0 | 9216.0 | 8192.0 | 0.65% | 17408.0 |
| 52 | layer2.3.conv2 | 128 4 4 | 128 4 4 | 147456.0 | 0.01 | 4,716,544.0 | 2359296.0 | 598016.0 | 8192.0 | 0.93% | 606208.0 |
| 53 | layer2.3.bn2 | 128 4 4 | 128 4 4 | 256.0 | 0.01 | 8,192.0 | 4096.0 | 9216.0 | 8192.0 | 0.65% | 17408.0 |
| 54 | layer2.3.conv3 | 128 4 4 | 512 4 4 | 65536.0 | 0.03 | 2,088,960.0 | 1048576.0 | 270336.0 | 32768.0 | 0.81% | 303104.0 |
| 55 | layer2.3.bn3 | 512 4 4 | 512 4 4 | 1024.0 | 0.03 | 32,768.0 | 16384.0 | 36864.0 | 32768.0 | 0.70% | 69632.0 |
| 56 | layer2.3.relu | 512 4 4 | 512 4 4 | 0.0 | 0.03 | 8,192.0 | 8192.0 | 32768.0 | 32768.0 | 0.44% | 65536.0 |

Table 16: Detailed breakdown of the architecture of the ResNet50 network in the PFM experiment (part 2).

| | Module Name | Input Shape | Output Shape | Params | Memory (MB) | MAdd | FLOPs | MemRead (B) | MemWrite (B) | Duration (%) | MemR+W (B) |
|---|---|---|---|---|---|---|---|---|---|---|---|
| 57 | layer3.0.conv1 | 512 4 4 | 256 4 4 | 131072.0 | 0.02 | 4,190,208.0 | 2097152.0 | 557056.0 | 16384.0 | 0.85% | 573440.0 |
| 58 | layer3.0.bn1 | 256 4 4 | 256 4 4 | 512.0 | 0.02 | 16,384.0 | 8192.0 | 18432.0 | 16384.0 | 0.67% | 34816.0 |
| 59 | layer3.0.conv2 | 256 4 4 | 256 2 2 | 589824.0 | 0.00 | 4,717,568.0 | 2359296.0 | 2375680.0 | 4096.0 | 1.07% | 2379776.0 |
| 60 | layer3.0.bn2 | 256 2 2 | 256 2 2 | 512.0 | 0.00 | 4,096.0 | 2048.0 | 6144.0 | 4096.0 | 0.64% | 10240.0 |
| 61 | layer3.0.conv3 | 256 2 2 | 1024 2 2 | 262144.0 | 0.02 | 2,093,056.0 | 1048576.0 | 1052672.0 | 16384.0 | 0.87% | 1069056.0 |
| 62 | layer3.0.bn3 | 1024 2 2 | 1024 2 2 | 2048.0 | 0.02 | 16,384.0 | 8192.0 | 24576.0 | 16384.0 | 0.67% | 40960.0 |
| 63 | layer3.0.relu | 1024 2 2 | 1024 2 2 | 0.0 | 0.02 | 4,096.0 | 4096.0 | 16384.0 | 16384.0 | 0.46% | 32768.0 |
| 64 | layer3.0.downsample.0 | 512 4 4 | 1024 2 2 | 524288.0 | 0.02 | 4,190,208.0 | 2097152.0 | 2129920.0 | 16384.0 | 0.98% | 2146304.0 |
| 65 | layer3.0.downsample.1 | 1024 2 2 | 1024 2 2 | 2048.0 | 0.02 | 16,384.0 | 8192.0 | 24576.0 | 16384.0 | 0.67% | 40960.0 |
| 66 | layer3.1.conv1 | 1024 2 2 | 256 2 2 | 262144.0 | 0.00 | 2,096,128.0 | 1048576.0 | 1064960.0 | 4096.0 | 0.91% | 1069056.0 |
| 67 | layer3.1.bn1 | 256 2 2 | 256 2 2 | 512.0 | 0.00 | 4,096.0 | 2048.0 | 6144.0 | 4096.0 | 0.66% | 10240.0 |
| 68 | layer3.1.conv2 | 256 2 2 | 256 2 2 | 589824.0 | 0.00 | 4,717,568.0 | 2359296.0 | 2363392.0 | 4096.0 | 1.08% | 2367488.0 |
| 69 | layer3.1.bn2 | 256 2 2 | 256 2 2 | 512.0 | 0.00 | 4,096.0 | 2048.0 | 6144.0 | 4096.0 | 0.66% | 10240.0 |
| 70 | layer3.1.conv3 | 256 2 2 | 1024 2 2 | 262144.0 | 0.02 | 2,093,056.0 | 1048576.0 | 1052672.0 | 16384.0 | 0.86% | 1069056.0 |
| 71 | layer3.1.bn3 | 1024 2 2 | 1024 2 2 | 2048.0 | 0.02 | 16,384.0 | 8192.0 | 24576.0 | 16384.0 | 0.67% | 40960.0 |
| 72 | layer3.1.relu | 1024 2 2 | 1024 2 2 | 0.0 | 0.02 | 4,096.0 | 4096.0 | 16384.0 | 16384.0 | 0.44% | 32768.0 |
| 73 | layer3.2.conv1 | 1024 2 2 | 256 2 2 | 262144.0 | 0.00 | 2,096,128.0 | 1048576.0 | 1064960.0 | 4096.0 | 0.91% | 1069056.0 |
| 74 | layer3.2.bn1 | 256 2 2 | 256 2 2 | 512.0 | 0.00 | 4,096.0 | 2048.0 | 6144.0 | 4096.0 | 0.66% | 10240.0 |
| 75 | layer3.2.conv2 | 256 2 2 | 256 2 2 | 589824.0 | 0.00 | 4,717,568.0 | 2359296.0 | 2363392.0 | 4096.0 | 1.11% | 2367488.0 |
| 76 | layer3.2.bn2 | 256 2 2 | 256 2 2 | 512.0 | 0.00 | 4,096.0 | 2048.0 | 6144.0 | 4096.0 | 0.64% | 10240.0 |
| 77 | layer3.2.conv3 | 256 2 2 | 1024 2 2 | 262144.0 | 0.02 | 2,093,056.0 | 1048576.0 | 1052672.0 | 16384.0 | 0.89% | 1069056.0 |
| 78 | layer3.2.bn3 | 1024 2 2 | 1024 2 2 | 2048.0 | 0.02 | 16,384.0 | 8192.0 | 24576.0 | 16384.0 | 0.68% | 40960.0 |
| 79 | layer3.2.relu | 1024 2 2 | 1024 2 2 | 0.0 | 0.02 | 4,096.0 | 4096.0 | 16384.0 | 16384.0 | 0.45% | 32768.0 |
| 80 | layer3.3.conv1 | 1024 2 2 | 256 2 2 | 262144.0 | 0.00 | 2,096,128.0 | 1048576.0 | 1064960.0 | 4096.0 | 0.90% | 1069056.0 |
| 81 | layer3.3.bn1 | 256 2 2 | 256 2 2 | 512.0 | 0.00 | 4,096.0 | 2048.0 | 6144.0 | 4096.0 | 0.64% | 10240.0 |
| 82 | layer3.3.conv2 | 256 2 2 | 256 2 2 | 589824.0 | 0.00 | 4,717,568.0 | 2359296.0 | 2363392.0 | 4096.0 | 1.11% | 2367488.0 |
| 83 | layer3.3.bn2 | 256 2 2 | 256 2 2 | 512.0 | 0.00 | 4,096.0 | 2048.0 | 6144.0 | 4096.0 | 0.65% | 10240.0 |
| 84 | layer3.3.conv3 | 256 2 2 | 1024 2 2 | 262144.0 | 0.02 | 2,093,056.0 | 1048576.0 | 1052672.0 | 16384.0 | 0.88% | 1069056.0 |
| 85 | layer3.3.bn3 | 1024 2 2 | 1024 2 2 | 2048.0 | 0.02 | 16,384.0 | 8192.0 | 24576.0 | 16384.0 | 0.67% | 40960.0 |
| 86 | layer3.3.relu | 1024 2 2 | 1024 2 2 | 0.0 | 0.02 | 4,096.0 | 4096.0 | 16384.0 | 16384.0 | 0.45% | 32768.0 |
| 87 | layer3.4.conv1 | 1024 2 2 | 256 2 2 | 262144.0 | 0.00 | 2,096,128.0 | 1048576.0 | 1064960.0 | 4096.0 | 0.91% | 1069056.0 |
| 88 | layer3.4.bn1 | 256 2 2 | 256 2 2 | 512.0 | 0.00 | 4,096.0 | 2048.0 | 6144.0 | 4096.0 | 0.65% | 10240.0 |
| 89 | layer3.4.conv2 | 256 2 2 | 256 2 2 | 589824.0 | 0.00 | 4,717,568.0 | 2359296.0 | 2363392.0 | 4096.0 | 1.09% | 2367488.0 |
| 90 | layer3.4.bn2 | 256 2 2 | 256 2 2 | 512.0 | 0.00 | 4,096.0 | 2048.0 | 6144.0 | 4096.0 | 0.65% | 10240.0 |
| 91 | layer3.4.conv3 | 256 2 2 | 1024 2 2 | 262144.0 | 0.02 | 2,093,056.0 | 1048576.0 | 1052672.0 | 16384.0 | 0.89% | 1069056.0 |
| 92 | layer3.4.bn3 | 1024 2 2 | 1024 2 2 | 2048.0 | 0.02 | 16,384.0 | 8192.0 | 24576.0 | 16384.0 | 0.69% | 40960.0 |
| 93 | layer3.4.relu | 1024 2 2 | 1024 2 2 | 0.0 | 0.02 | 4,096.0 | 4096.0 | 16384.0 | 16384.0 | 0.44% | 32768.0 |
| 94 | layer3.5.conv1 | 1024 2 2 | 256 2 2 | 262144.0 | 0.00 | 2,096,128.0 | 1048576.0 | 1064960.0 | 4096.0 | 0.91% | 1069056.0 |
| 95 | layer3.5.bn1 | 256 2 2 | 256 2 2 | 512.0 | 0.00 | 4,096.0 | 2048.0 | 6144.0 | 4096.0 | 0.66% | 10240.0 |
| 96 | layer3.5.conv2 | 256 2 2 | 256 2 2 | 589824.0 | 0.00 | 4,717,568.0 | 2359296.0 | 2363392.0 | 4096.0 | 1.11% | 2367488.0 |
| 97 | layer3.5.bn2 | 256 2 2 | 256 2 2 | 512.0 | 0.00 | 4,096.0 | 2048.0 | 6144.0 | 4096.0 | 0.64% | 10240.0 |
| 98 | layer3.5.conv3 | 256 2 2 | 1024 2 2 | 262144.0 | 0.02 | 2,093,056.0 | 1048576.0 | 1052672.0 | 16384.0 | 0.89% | 1069056.0 |
| 99 | layer3.5.bn3 | 1024 2 2 | 1024 2 2 | 2048.0 | 0.02 | 16,384.0 | 8192.0 | 24576.0 | 16384.0 | 0.69% | 40960.0 |
| 100 | layer3.5.relu | 1024 2 2 | 1024 2 2 | 0.0 | 0.02 | 4,096.0 | 4096.0 | 16384.0 | 16384.0 | 0.45% | 32768.0 |
| 101 | layer4.0.conv1 | 1024 2 2 | 512 2 2 | 524288.0 | 0.01 | 4,192,256.0 | 2097152.0 | 2113536.0 | 8192.0 | 1.03% | 2121728.0 |
| 102 | layer4.0.bn1 | 512 2 2 | 512 2 2 | 1024.0 | 0.01 | 8,192.0 | 4096.0 | 12288.0 | 8192.0 | 0.67% | 20480.0 |
| 103 | layer4.0.conv2 | 512 2 2 | 512 1 1 | 2359296.0 | 0.00 | 4,718,080.0 | 2359296.0 | 9445376.0 | 2048.0 | 1.69% | 9447424.0 |
| 104 | layer4.0.bn2 | 512 1 1 | 512 1 1 | 1024.0 | 0.00 | 2,048.0 | 1024.0 | 6144.0 | 2048.0 | 0.64% | 8192.0 |
| 105 | layer4.0.conv3 | 512 1 1 | 2048 1 1 | 1048576.0 | 0.01 | 2,095,104.0 | 1048576.0 | 4196352.0 | 8192.0 | 1.16% | 4204544.0 |
| 106 | layer4.0.bn3 | 2048 1 1 | 2048 1 1 | 4096.0 | 0.01 | 8,192.0 | 4096.0 | 24576.0 | 8192.0 | 0.68% | 32768.0 |
| 107 | layer4.0.relu | 2048 1 1 | 2048 1 1 | 0.0 | 0.01 | 2,048.0 | 2048.0 | 8192.0 | 8192.0 | 0.46% | 16384.0 |
| 108 | layer4.0.downsample.0 | 1024 2 2 | 2048 1 1 | 2097152.0 | 0.01 | 4,192,256.0 | 2097152.0 | 8404992.0 | 8192.0 | 1.51% | 8413184.0 |
| 109 | layer4.0.downsample.1 | 2048 1 1 | 2048 1 1 | 4096.0 | 0.01 | 8,192.0 | 4096.0 | 24576.0 | 8192.0 | 0.68% | 32768.0 |
| 110 | layer4.1.conv1 | 2048 1 1 | 512 1 1 | 1048576.0 | 0.00 | 2,096,640.0 | 1048576.0 | 4202496.0 | 2048.0 | 1.14% | 4204544.0 |
| 111 | layer4.1.bn1 | 512 1 1 | 512 1 1 | 1024.0 | 0.00 | 2,048.0 | 1024.0 | 6144.0 | 2048.0 | 0.66% | 8192.0 |
| 112 | layer4.1.conv2 | 512 1 1 | 512 1 1 | 2359296.0 | 0.00 | 4,718,080.0 | 2359296.0 | 9439232.0 | 2048.0 | 1.57% | 9441280.0 |
| 113 | layer4.1.bn2 | 512 1 1 | 512 1 1 | 1024.0 | 0.00 | 2,048.0 | 1024.0 | 6144.0 | 2048.0 | 0.65% | 8192.0 |
| 114 | layer4.1.conv3 | 512 1 1 | 2048 1 1 | 1048576.0 | 0.01 | 2,095,104.0 | 1048576.0 | 4196352.0 | 8192.0 | 1.10% | 4204544.0 |
| 115 | layer4.1.bn3 | 2048 1 1 | 2048 1 1 | 4096.0 | 0.01 | 8,192.0 | 4096.0 | 24576.0 | 8192.0 | 0.68% | 32768.0 |
| 116 | layer4.1.relu | 2048 1 1 | 2048 1 1 | 0.0 | 0.01 | 2,048.0 | 2048.0 | 8192.0 | 8192.0 | 0.44% | 16384.0 |
| 117 | layer4.2.conv1 | 2048 1 1 | 512 1 1 | 1048576.0 | 0.00 | 2,096,640.0 | 1048576.0 | 4202496.0 | 2048.0 | 1.11% | 4204544.0 |
| 118 | layer4.2.bn1 | 512 1 1 | 512 1 1 | 1024.0 | 0.00 | 2,048.0 | 1024.0 | 6144.0 | 2048.0 | 0.65% | 8192.0 |
| 119 | layer4.2.conv2 | 512 1 1 | 512 1 1 | 2359296.0 | 0.00 | 4,718,080.0 | 2359296.0 | 9439232.0 | 2048.0 | 1.51% | 9441280.0 |
| 120 | layer4.2.bn2 | 512 1 1 | 512 1 1 | 1024.0 | 0.00 | 2,048.0 | 1024.0 | 6144.0 | 2048.0 | 0.64% | 8192.0 |
| 121 | layer4.2.conv3 | 512 1 1 | 2048 1 1 | 1048576.0 | 0.01 | 2,095,104.0 | 1048576.0 | 4196352.0 | 8192.0 | 1.09% | 4204544.0 |
| 122 | layer4.2.bn3 | 2048 1 1 | 2048 1 1 | 4096.0 | 0.01 | 8,192.0 | 4096.0 | 24576.0 | 8192.0 | 0.68% | 32768.0 |
| 123 | layer4.2.relu | 2048 1 1 | 2048 1 1 | 0.0 | 0.01 | 2,048.0 | 2048.0 | 8192.0 | 8192.0 | 0.45% | 16384.0 |
| 124 | avgpool | 2048 1 1 | 2048 1 1 | 0.0 | 0.00 | 0.0 | 0.0 | 0.0 | 0.0 | 0.59% | 0.0 |
| 125 | fc | 2048 | 1000 | 2049000.0 | 0.00 | 4,095,000.0 | 2048000.0 | 8204192.0 | 4000.0 | 1.45% | 8208192.0 |
| total | | | | 25557032.0 | 2.25 | 171,759,128.0 | 86057984.0 | 8204192.0 | 4000.0 | 100.00% | 106946624.0 |

