# OpenReview forum: "Vanishing Feature: Diagnosing Model Merging and Beyond"
_CPAL.cc/2025/Proceedings_Track — CPAL 2025 (Proceedings Track) Oral_

### Official Review · Reviewer_VMLQ · 2025-01-02
**Review of Submission 45**

**Rating:** 8
**Confidence:** 3

**Review:**

**Strengths**
1. This paper was well motivated and written. The experimental results were clearly explained.
2. The theoretical results on linear networks were helpful for grounding the problem and were, to my knowledge, novel.
3. The connection of the vanishing feature to other work (particularly the permutation merging - Sec. 4.2) was nice and enabled a common thread to a number of works. This provides new insight and I believe makes this work a valuable reference for other work in model merging.
4. The results, on several architectures and data sets, compared to other methods for model merging made, were convincing.
5. The exploration of the vanishing feature in pruning was interesting and made the paper feel more substantial.

**Weaknesses**
1. I was confused by how "As $l$ increases, the scales of $g$ and $\tilde{h}$ decay exponentially, allowing parameters in the later layers to dominate f" (lines 156-157). As $g^l(x) \leftarrow \lambda^l \cdot g^l(x)$, shouldn't the parameters $g$ either be decaying exponentially (and thus, become increasingly less important), or grow exponentially (and thus, become increasingly important), depending on if $\lambda < 0$ of $\lambda > 0$? Additionally, looking at Fig 1 (center), it is not clear that the weight norms really grow exponentially, as a function of depth. Not being able to fully understand this made me slightly wary of the later discussion on what these results implied.
2. (Bordering on minor comment) I think Sec. 3.1 should be placed elsewhere. It was confusing to have discussion about VGG16 and ImageNet in a section where the focus was on Linear Networks.
3. Figures 2, 3, 5 have subpanels referred to as "A", "B", ... but there are no such labels on the actual subpanels.

**Minor comments**
1. "to approximate the performance of model ensembles Sagi and Rokach [7] while reducing storage..." (lines 21-22) should be just "[7]"
2. I was confused by what was meant by $\overline{f(x)}$ (line 56). The authors define it as an "average over edge models", but then have two model summed together by a weighting $\alpha$. Saying "$\alpha$ can be inferred from context" (line 57) was not clear to me.
3. In the Sec. 2 the models being merged are referred to as "$f_0$ and $f_1$, but then also $f_1$ and $f_2$ (line 92).
4. $fp$ should be $f_p$ (line 297).

---

### Official Review · Reviewer_46bh · 2025-01-10
**Review for Vanishing Feature**

**Rating:** 7
**Confidence:** 4

**Review:**

**Summary:**

This paper introduces the "vanishing feature" phenomenon in model merging and proposes a "Preserve-First Merging" strategy to improve performance. The work also shows how addressing this issue can enhance model pruning through post-pruning normalization.

**Strength:**

The paper is exceptionally well-written, with clear explanations and methodical presentation of ideas throughout. The experimental validation is comprehensive and thorough, featuring detailed ablation studies and convincing empirical results that strongly support the paper's claims.

**Weakness**

1. Is there any theoretical analysis of the Vanishing Feature phenomenon? Sections 3.2 and 3.3 demonstrate the vanishing feature phenomenon experimentally in linear networks and general settings. However, a theoretical explanation for why this occurs would be valuable—even if limited to linear networks, as nonlinear networks are more challenging to analyze.

2. The connection between the vanishing feature and the proposed Preserve-First Merging (PFM) method needs stronger support. While the experiments show that PFM improves test accuracy as the number of preserved layers increases, this improvement could simply result from the increased parameter count. To demonstrate that PFM specifically addresses the vanishing feature issue, it would be better to include a figure showing how $\||f_{\alpha}(x) - f_{\alpha}(0)\||$ changes with the number of preserved layers.

---

### Official Review · Reviewer_3evf · 2025-01-14
**Review for Submission45**

**Rating:** 7
**Confidence:** 3

**Review:**

**Summary**:
This paper investigates the challenges of model merging in neural networks, focusing on performance inconsistencies when combining pre-trained models with different initializations. It introduces the "vanishing feature" phenomenon, where input-induced features diminish during forward propagation in merged models, leading to performance degradation. Through theoretical analysis and empirical validation, the paper links this phenomenon to issues like variance collapse and explores techniques such as permutation-based merging and normalization strategies. The authors propose a "Preserve-First Merging" (PFM) strategy, which preserves early-layer features and enhances the performance of merged models. The vanishing feature phenomenon is also extended to the domain of model pruning, where normalization techniques improve high-sparsity pruning results.

**Strengths**:
1. **Novel Contribution**: The identification of the vanishing feature phenomenon is a significant advancement in understanding challenges related to model merging and pruning. The theoretical insights provided a unified explanation for related phenomena, including variance collapse and the effectiveness of existing merging techniques.
2. **Practical Solutions**: The proposed PFM strategy demonstrates a meaningful improvement in merging performance without the need for post-training, making it computationally efficient.Extending the vanishing feature analysis to model pruning highlights the broader applicability of the research.
3. **Strong Theoretical Foundation**:The paper provides rigorous theoretical analysis of linear networks to explain the vanishing feature phenomenon.It effectively connects theoretical findings with practical techniques like normalization and residual connections.
4. **Clear Presentation**: Visualizations and detailed discussions clarify complex concepts like the impact of layer-wise parameter scaling and the interplay between merging techniques and normalization.

**Weaknesses**:
1. **Simplified Settings**:The theoretical analysis is grounded in linear networks, which may limit its direct applicability to complex non-linear architectures commonly used in modern deep learning.
2. **Limited Scalability Analysis**: The study primarily focuses on relatively standard architectures and small-to-medium-scale datasets. Its applicability to larger architectures, such as transformers, and real-world datasets remains unexplored. Moreover, The paper does not comprehensively benchmark PFM against other state-of-the-art merging or pruning techniques, which would provide a clearer picture of its relative advantages.
3. **Optimization Overhead**:While PFM reduces post-training costs, its reliance on preserving specific layers and fine-grained parameter adjustments may introduce computational overhead, particularly for deeper or wider networks.

---

### Meta-Review · Area_Chair_WX6a · 2025-02-05

**Recommendation:** Accept (Oral)
**Confidence:** 4

**Metareview:**

This paper studies the "vanishing feature" phenomenon in model merging and proposes a "Preserve-First Merging" strategy to improve performance. The authors conduct substantial efforts in rebuttal and have addressed most of the concerns. All reviewers acknowledge the significance of this research.

---

### Decision · Program_Chairs · 2025-02-11

Accept (Oral)